# Deletion of an enhancer that controls Wnt gene expression following tissue injury produces increased adipogenesis in regenerated muscle

Catriona Y. Logan*, Xinhong Lim, Matt P. Fish, Makiko Mizutani, Brooke Swain and Roel Nusse*

## ABSTRACT

The capacity to detect and respond to injury is crucial for the recovery and long-term survival of many organisms. Wnts are commonly induced by tissue damage but how they become activated transcriptionally is not well understood. Here, we report that mouse *Wnt1* and *Wnt10b* are induced following injury in both lung and muscle. These Wnts occupy the same chromosome and are transcribed in opposite directions with 12 kb between them. We identified a highly conserved cis-acting regulatory region (enhancer) residing between *Wnt1* and *Wnt10b* that, when fused to a *lacZ* reporter, is activated post-injury. This enhancer harbors putative AP-1-binding sites that are required for reporter activity, a feature observed in other injury-responsive enhancers. Injured muscles in mice carrying a germline deletion of the enhancer region display reduced *Wnt1* and *Wnt10b* expression and show elevated intramuscular adipogenesis, which can be a hallmark of impaired muscle regeneration or tissue maintenance. Enhancer redundancy is common in development, but our *in vivo* analysis shows that loss of a single injury-responsive regulatory region in adult tissues can produce a detectable phenotype.

KEY WORDS: Wnt, Injury, Regeneration, Enhancer, Muscle, Mouse

## INTRODUCTION

The ability of an organism to regenerate or repair tissues following injury requires that a tissue can detect damage and deploy mechanisms that facilitate the restoration of tissue architecture. Central to this process is the transcriptional activation of genes, mediated by regulatory sequences termed 'enhancers.' Many cell signals that regulate embryonic development are induced following tissue injury and re-used in adults to drive regeneration or repair (Maddaluno et al., 2017; Aros et al., 2021; Fazilaty and Basler, 2023). Regulatory sequences that promote developmental expression of some of these signals have been characterized, but how tissue damage is sensed and integrated at the level of enhancers following injury, particularly for cell-to-cell signaling molecules, is not well understood.

Wnts are a family of developmental signals that also function in both normal adult tissue homeostasis and disease (Nusse and Clevers, 2017). Additionally, Wnt genes are commonly induced following injury in a wide variety of organisms to facilitate repair, regeneration or re-patterning (Whyte et al., 2012) and may even confer regenerative capacity (Campos et al., 2025 preprint). The induction of Wnts post-injury is often rapid (McClure et al., 2008; Petersen and Reddien, 2009; Fernandez-Martos et al., 2011; Vizcaya-Molina et al., 2018; Cazet et al., 2021), suggesting that Wnts are well-positioned as cell–cell signaling factors to integrate injury signals and transition cells from a homeostatic to regenerative state. Only a few studies have explored the transcriptional regulation of Wnts, with most focused on developing embryos (Echelard et al., 1994; Danielian et al., 1997; Rowitch et al., 1998; Park et al., 2012; O'Brien et al., 2018; Lekven et al., 2019; van de Grift et al., 2025 preprint). In *Drosophila*, an injury-responsive Wnt enhancer that drives *wingless* and *Wnt6* in early larval wing disks has been described (Harris et al., 2016), but how Wnts are activated following injury in adult tissues and particularly in vertebrates is unknown.

Multiple types of injury-responsive enhancers have now been identified. Some display a dual function, utilized during both development and regeneration (Huang et al., 2012), while others are largely dedicated to damage responses (Guenther et al., 2015; Heller et al., 2022). Other enhancers can be separated into different tissue-specific domains or modules (Kang et al., 2016), display temporal regulation, becoming silenced as tissues mature (Harris et al., 2016, 2020), or possess distinct initial injury-sensing and regeneration-deploying activities, with important functional consequences that may predict whether a species can regenerate or not (Wang et al., 2020). Recently, an enhancer that drives one gene under normal conditions that can be re-purposed to drive another gene upon injury has been described (Rao et al., 2025). These examples reflect the diversity and complexity of regulatory sequences that control injury-responsive gene activation and highlight the need for deeper characterization of their dynamics, function and logic.

Additionally, the degree to which injury-responsive enhancers are unique versus redundant, and whether mutations in injury-responsive enhancers negatively affect tissue restoration are not well understood. A requirement for injury-responsive enhancers has been demonstrated in several examples (Hewitt et al., 2017; Soukup et al., 2019; Wang et al., 2020; Sun et al., 2022; Gracia-Latorre et al., 2022; Zlatanova et al., 2023), but defects can be hard to detect (Sun et al., 2022). Understanding how impaired enhancer function or enhancer sequence variations might predispose one towards poor tissue healing and function over time (Zaugg et al., 2022) has important implications for human health.

Howard Hughes Medical Institute, Department of Developmental Biology, Institute for Stem Cell Biology and Regenerative Medicine, Stanford University School of Medicine, Stanford, CA 94305, USA.

*Authors for correspondence (rnusse@stanford.edu; cylogan@stanford.edu)

C.Y.L., 0000-0002-1701-3989; X.L., 0000-0002-4725-5161; M.P.F., 0009-0001-8831-0892; M.M., 0009-0008-2598-8808; B.S., 0009-0003-8585-3386; R.N., 0000-0001-7082-3748

In this study, we investigated how Wnts are transcriptionally activated in adult tissues following damage. We report identification of a regulatory region residing between *Wnt1* and *Wnt10b* that *in vivo* responds to tissue injury and when deleted results in elevated adipogenesis in regenerated muscle.

## RESULTS
### *Wnt1* and *Wnt10b* are upregulated in response to tissue injury

To examine how injury is sensed by Wnt genes and to identify those that are induced by injury, we performed an mRNA *in situ* hybridization screen in four different injury models for all 19 Wnts (Fig. S1A-D, Table S1). Treatments included naphthalene administration (lung) (Hong et al., 2001), barium chloride (BaCl$_2$) injection (muscle) (Wosczyna et al., 2019), carbon tetrachloride (CCl$_4$) exposure (liver) and streptozotocin injury (pancreas) (Furman, 2021).

All tissue injuries triggered expression of multiple Wnts (Fig. 1A-D, Fig. S1A-D, Table S1). Among those induced, *Wnt1* and *Wnt10b* were observed in two specific contexts: following BaCl$_2$ injection in the tibialis anterior (TA) muscle (Fig. 1A-G, Fig. S1B) and following naphthalene injection in the lung (Figs S1A, S2A,B). BaCl$_2$ injection is an acute chemical injury model in which muscle fibers degenerate and then regenerate over 2 weeks (Hardy et al., 2016). Naphthalene injection is an acute airway injury model in which damaged conducting airway rapidly regenerates over a similar time frame (Mahvi et al., 1977; Stripp et al., 1995; Van Winkle et al., 1995). In both lung and muscle, uninjured tissue initially displayed low *Wnt1* and *Wnt10b* transcripts, but injury induced *Wnt1* and *Wnt10b* mRNA signals, with expression continuing for several days (Fig. 1A,B, Fig. S2A,B). Quantification of *Wnt1*- and *Wnt10b*-positive nuclei in muscle confirmed elevation of mRNA over 1-3 days post-injury, although the temporal dynamics differed between the Wnts (Fig. 1C,D). These data demonstrate that Wnt genes are induced by injury in a variety of mouse tissues (Fig. 1, Figs S1A-D, S2, S3), and that *Wnt1* and *Wnt10b* are two specific Wnts that become expressed in more than one injury context.

For comparison, we examined *Wnt5a* post-BaCl$_2$ injury, because it is found in muscle and is induced by tissue damage (Polesskaya et al., 2003; Reggio et al., 2020). *Wnt5a* displayed transcripts at similar abundance and intensity to *Wnt1* and *Wnt10b*, with *Wnt5a* expression gradually increasing between 1-3 days post-injury (Fig, S3A,B). These data show that *Wnt1* and *Wnt10b* staining levels overall are typical for expression of Wnts detected in injured muscle tissue (Fig. 1A,B, Fig. S3A).

We next examined which cells express *Wnt1* and *Wnt10b* in muscle. Previous studies show that Wnts are predominantly found in fibro-adipogenic progenitors (FAPs) (Joe et al., 2010; McKellar et al., 2021), one of the stromal cell populations that regulates muscle regeneration (Joe et al., 2010; Wosczyna et al., 2019), with low-level expression in other cell types (Reggio et al., 2020). Using 2-plex *in situ* hybridization, we observed that a subset of *Wnt1*- (Fig. 1E) and *Wnt10b*- (Fig. 1F) positive cells are FAPs, marked by *Pdgfra*. A subset of Wnt-positive cells also expresses the satellite cell marker *Pax7*, and *Adgre1*, a marker predominantly expressed in macrophages (Fig. 1E,F). Double-positive nuclei expressing *Wnt/ Pdgfra* or *Wnt/Pax7* were rare, comprising less than 15% of the *Wnt1*- and *Wnt10b*-expressing cells (Fig. S4A,B). Similarly, *Wnt/Adgre1* double-positive cells never constituted more than 25% of Wnt-positive cells. (Fig. S4A,B). Because not all FAPs express *Pdgfra*, and FAPs can shuttle between dynamic cell states (Oprescu et al., 2020), we also examined co-expression of *Wnt1* and *Wnt10b* with

other FAP markers [*Osr1*, *Wisp1* (*Ccn4*), *Dpp4*, *Dlk1* and *Cxcl14*]. *Wnt1* and *Wnt10b* co-expression with these FAP markers was also rare but detectable, showing that both Wnts are found in multiple FAP subpopulations (Fig. S4A,B,D). We may have under-counted double-positive cells because a strongly expressed transcript masks a more weakly expressed one when stained by non-fluorescent dual *in situ* hybridization. We also counted cells as double positive only if their nuclei could clearly be distinguished from neighboring cells. We do not know whether *Wnt1/Wnt10b* expression post-injury in multiple cell types reflects a general role for Wnt signals in response to tissue damage, or if each Wnt plays distinct functions within each cell population.

We also examined whether *Wnt1* and *Wnt10b* were co-expressed in the same cells. *Wnt1* and *Wnt10b* double-positive nuclei were observed but their numbers were low at 1, 3 and 5 days post-injury (Fig. S4C). Most nuclei displayed one or the other signal, but not both (Fig. 1G). Since the first few days post-injury are highly dynamic, non-overlapping Wnt staining could be due to differences in the dynamics of promoter engagement with enhancers, mRNA perdurance, and expression of these genes in distinct cell subpopulations. Despite low abundance, we estimated that at least ~10% of cells were *Wnt1* and *Wnt10b* double positive at any given time point (Fig. S4C), suggesting that a common regulatory element might drive expression of these two Wnts.

### Injury-responsive regulatory elements reside between *Wnt1* and *Wnt10b*

In the zebrafish, *Wnt1* and *Wnt10b* are coordinately expressed through shared developmental enhancers (Lekven et al., 2019). Given *Wnt1* and *Wnt10b* expression in damaged adult lung and muscle, we investigated whether these genes might share regulatory elements post-injury in the mouse. *Wnt1* and *Wnt10b* reside on chromosome 15, are transcribed in opposite directions, and are separated by 12 kb, which provided a small, tractable interval in which to search for putative injury-responsive regulatory sequences. We identified three peaks of high conservation in the intergenic region between *Wnt1* and *Wnt10b* (Fig. 2A) and hypothesized that they might regulate the injury-responsive expression of these Wnts.

We selected a 6 kb region that encompassed the three most prominent conserved peaks for further analysis. This 6 kb sequence and sequences representing each individual peak were fused to a *lacZ* reporter construct containing a basal *Hsp68* (*Hspa1a/b*) promoter (Kothary et al., 1989) (Fig. 2A; see Fig. S5A for all constructs). Multiple adult transgenic lines carrying independent insertions for each construct were produced.

Initially, all lines were tested in the lung. Reporter activity was assessed at 5 days post-naphthalene injection, as airway restoration is incomplete at this time point and we could verify that injury had occurred (Zemke et al., 2009; Volckaert et al., 2011). The 6 kb region (*n*=3/3 independent lines), as well as Enhancers 2 (Enh2, *n*=2/2 independent lines), 3 (Enh3, *n*=5/5 independent lines) and 2+3 (Enh2+3, *n*=3/4 independent lines) induced *lacZ* activity in damaged airways. We could not ascertain the injury-responsiveness of Enhancer 1, as only two adult lines were obtained, one that responded to injury and one that did not (*n*=1/2 independent lines) (Fig. S5B, Table 1). Vehicle-injected animals did not display airway reporter expression (Fig. S5B). X-gal staining (representing *lacZ* activity) was occasionally observed in non-airway cell types, but patterns were inconsistent between independently generated lines, suggesting that expression was due to reporter insertion into a transcriptionally active genomic locus but not reflective of enhancer-specific activity. Non-transgenic animals lacked staining

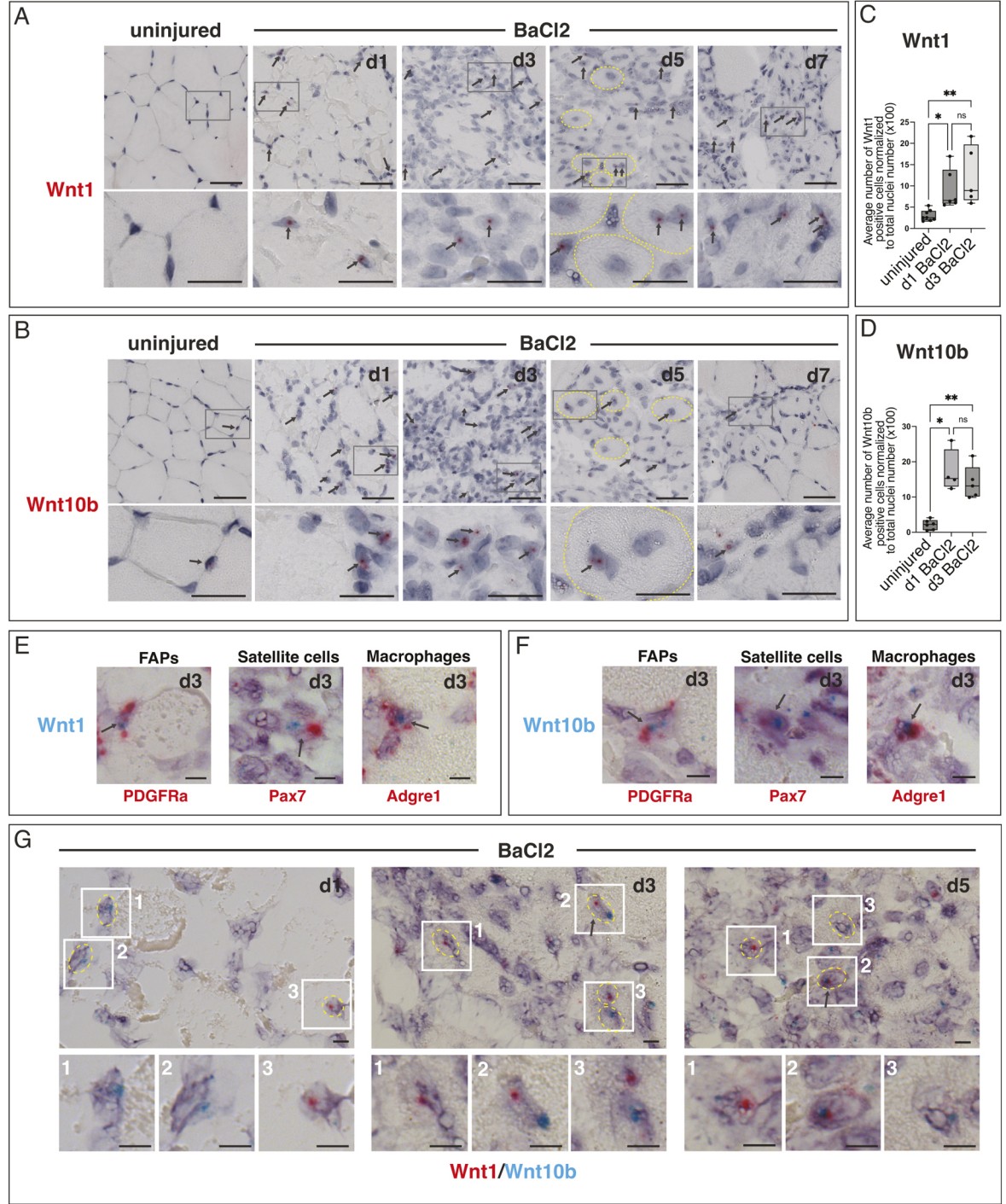

**Fig. 1. *Wnt1* and *Wnt10b* are induced by tissue injury following BaCl₂ injection.** (A,B) Wnt expression post-BaCl₂-induced injury in skeletal muscle, assessed by RNAscope mRNA *in situ* hybridization. *Wnt1* (A) and *Wnt10b* (B) expression in TA muscle at 1, 3, 5 and 7 days following BaCl₂ injection are shown. Uninjured contralateral muscle at 1 day post-injury is shown for comparison. *Wnt1*- and *Wnt10b*-positive cells are infrequently observed in uninjured muscle. Transcripts appear in single cells post-injury and then in nascent myofibers, which display centrally located nuclei. Boxes mark regions enlarged under each panel. Arrows mark red mRNA signals. Yellow dashed lines outline myofibers. (C,D) Quantification of *Wnt1* and *Wnt10b* expression at days 1 and 3 post-injury. (C) The number of *Wnt1*-positive nuclei increases significantly at 1 and 3 days post-injury compared to uninjured contralateral muscles. *$P$=0.0155, **$P$=0.0026 (one-way ANOVA, Kruskal–Wallis test). $n$=8 uninjured represents $n$=4 d1 uninjured, $n$=4 d3 uninjured; $n$=6 d1 BaCl₂, $n$=5 d3 BaCl₂. (D) Number of *Wnt10b*-positive nuclei increases significantly at 1 and 3 days post-injury compared to uninjured contralateral muscles. *$P$=0.0378, **$P$=0.0076 (Brown–Forsythe and Welch one-way ANOVA tests). $n$=6 uninjured consists of $n$=3 d1 uninjured, $n$=3 d3 uninjured; $n$=4 d1 BaCl₂, $n$=5 d3 BaCl₂. (E,F) Duplex mRNA *in situ* hybridization showing *Wnt1* (E) and *Wnt10b* (F) signals in *Pdgfra*-, *Pax7*- and *Adgre*-positive cells, which mark the FAPs, satellite cells and macrophages, respectively, at 3 days post-injury. Arrows point to examples of double-positive nuclei. $n$=2 animals. (G) Duplex mRNA *in situ* hybridization showing that *Wnt1* and *Wnt10b* can be colocalized. Representative examples from 1, 3 and 5 days post-injury are shown. White numbered boxes mark regions magnified below each image. Yellow dashed lines mark individual nuclei. Arrows mark *Wnt* double-positive cells. Box limits mark the 25th and 75th percentiles, whiskers mark minimum to maximum values, and horizontal line marks the median. d, days post-injury; ns, not significant. Scale bars: 20 μm (A,B); 5 μm (E-G).

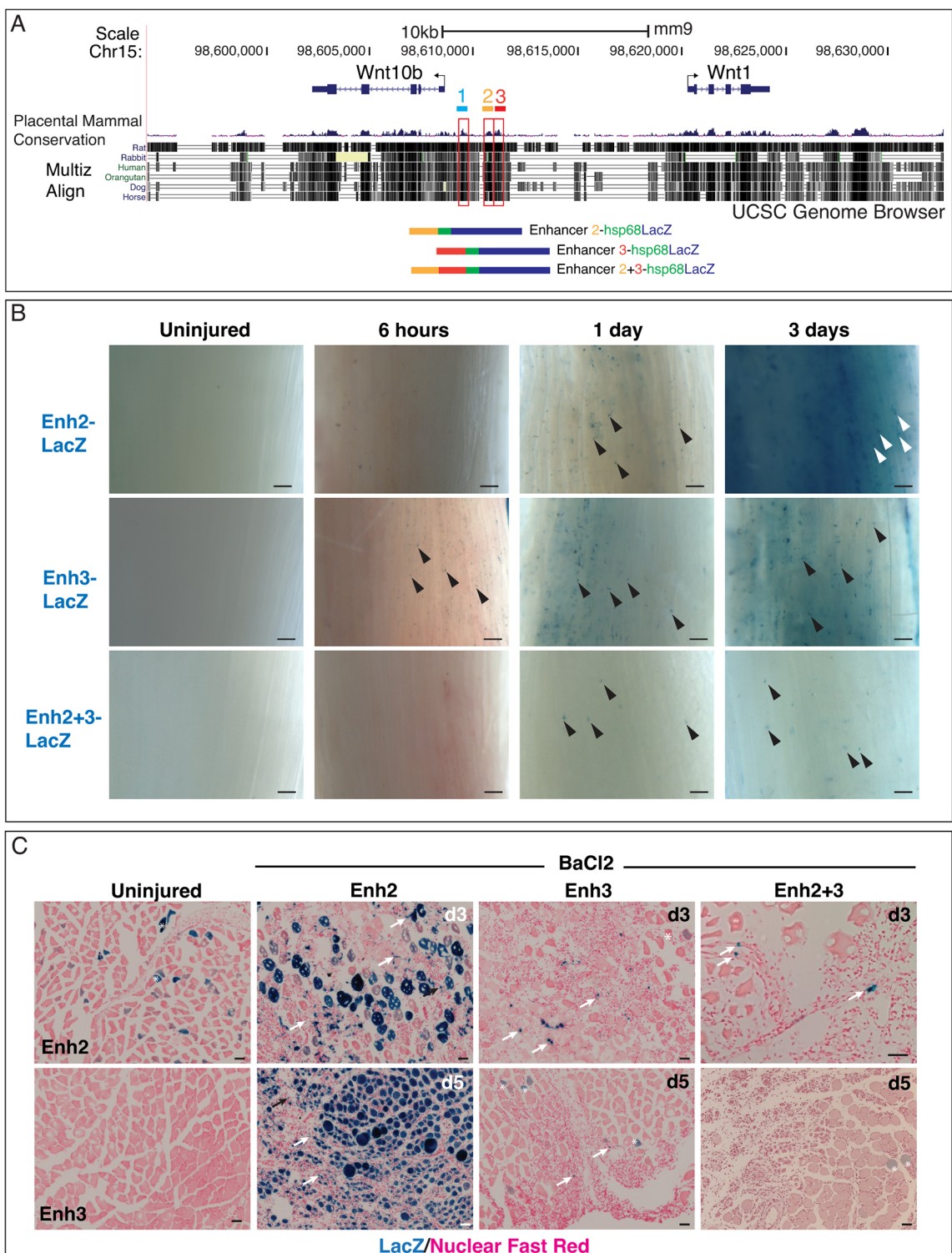

**Fig. 2. Regulatory sequences residing between *Wnt1* and *Wnt10b* are injury responsive.** (A) UCSC Genome Browser data showing that *Wnt1* and *Wnt10b* reside on the same chromosome and are transcribed in opposite directions (arrows). Three highly conserved sequences between *Wnt1* and *Wnt10b* are labeled: 1 (light blue), 2 (yellow), 3 (red). Multiple genome alignment tracks (Multiz Align) and conserved peaks from placental mammal alignments are shown. Below, schematics of constructs containing putative Enhancers 2, 3, and 2+3 fused to the *hsp68-LacZ* gene are provided [minimal *hsp68* promoter (green), *lacZ* reporter (blue)]. (B) BaCl$_2$ injury of adult Enh2-LacZ, Enh3-LacZ and Enh2+3-LacZ mice reveal that all three sequences drive reporter expression following injury. Whole-mount muscles are shown. Uninjured contralateral muscles at 6 h are provided for comparison. Arrowheads mark examples of single cells that express the reporter. (C) Sectioned tissues from X-gal-stained, BaCl$_2$-injured TA muscles from Enhancer 2, 3 and 2+3 reporter mice at 3-5 days. Uninjured muscles from Enhancer 2 and 3 reporters are shown. Arrows mark examples of single cells that express the reporter. Some variable and sporadic background staining is observed (asterisks). d, days post-injury. Scale bars: 20 µm.

**Table 1. Reporter activity in mouse lungs at 5 days post-naphthalene injury**

| Construct name | Injury response in airways (number of lines showing reporter activity) |
|---|---|
| 6 kb-LacZ | 3/3 |
| Enhancer 2+3-LacZ | 3/4 |
| Enhancer1-LacZ | 1/2 (inconclusive) |
| Enhancer2-LacZ | 2/2 |
| Enhancer3-LacZ | 5/5 |
| Enhancer3-GFP | 3/3 |
| Hsp68-LacZ | 0/4 (basal expression) |
| Enh3$^{(AP-1mutant)}$-LacZ | 1/8 |
| 5.5 kb 3′ Wnt1 Enhancer-LacZ | 0/3 |

(not shown) and the basal promoter alone fused to *lacZ* ('No Enhancer') produced sparse reporter activity (Fig. S5B, Table 1).

Finally, a Wnt1-LacZ mouse carrying a 5.5 kb 3′ regulatory region that drives embryonic *Wnt1* expression in the CNS (Echelard et al., 1994) was examined in injured lung (Fig. S5B, Table 1). This reporter failed to display *lacZ* activity (n=0/3 embryos), showing that this well-known embryonic *Wnt1* regulatory sequence is not activated by injury.

From these data, we conclude that the 6 kb region residing between *Wnt1* and *Wnt10b* harbors injury-responsive elements, with Enhancers 2 and 3 showing injury-responsive activity.

## Injury-responsive Enhancers 2 and 3 become expressed in damaged muscle

Wnts and Wnt signaling have been well-studied in developing muscle (von Maltzahn et al., 2012; Girardi and Le Grand, 2018) and play roles in muscle regeneration (Polesskaya et al., 2003; Brack et al., 2008; Murphy et al., 2014; Huraskin et al., 2016; Reggio et al., 2020; Gurriaran-Rodriguez et al., 2024; Kamizaki et al., 2024). Because *Wnt1* and *Wnt10b* transcripts are induced by muscle injury (Fig. 1) and *Wnt10b* mutants display a regeneration phenotype in injured muscles (Vertino et al., 2005), we reasoned that Enhancers 2 and/or 3 might regulate *Wnt1* and *Wnt10b* in muscle tissue. One representative mouse line of Enhancer 2-LacZ (Enh2-LacZ), Enhancer 3-LacZ (Enh3-LacZ) and Enhancer 2+3-LacZ (Enh2+3-LacZ) was selected for all subsequent experiments.

Following BaCl$_2$ injury, Enh2-LacZ and Enh3-LacZ were activated. As early as 6 h, Enh3-LacZ staining was observed as small, scattered cells that were visible as speckles in whole-mount muscle tissue, followed by speckled Enh2-LacZ staining at 24 h (Fig. 2B). Enh3-LacZ was similarly rapidly induced before Enh2-LacZ in injured lung (Fig. S6). By 3 days, BaCl$_2$-injured Enh2 reporter tissues displayed very dark X-gal staining in whole mounts (Fig. 2B), which in section appeared as blue regenerating myofibers as well as small individual cells (Fig. 2C, arrows). Enh3 reporter mice exhibited only small cells near injured myofibers at days 3-5 (Fig. 2C, arrows) and staining in nascent myofibers was not observed.

We examined the Enh2+3-LacZ reporter to assess the combined activity of Enhancers 2 and 3 in injured muscle. Speckled staining appeared in Enh2+3-LacZ mice at 1 day post-injury in whole-mounts and continued for 3 days (Fig. 2B, arrowheads), although expression appeared weaker compared to Enhancers 2 or 3 reporters alone (Fig. 2B). In sections by day 3, single cells next to myofibers were observed, but no expression in regenerating myofibers was seen (Fig. 2C). By day 5, virtually no reporter expression was detected (Fig. 2C).

Uninjured muscles displayed no reporter activity (Fig. 2B,C), although sporadic myofiber staining could sometimes be observed

(Fig. 2C, asterisks). This staining has been seen in other contexts (Guenther et al., 2015), possibly resulting from basal activity of the Hsp68-LacZ construct, and is considered background. Non-transgenic muscles displayed no X-gal staining (data not shown). Table S2 summarizes reporter expression observed in Enh2-LacZ, Enh3-LacZ and Enh2+3-LacZ muscles. Together, these data suggest that Enhancers 2 and 3 likely integrate different transcription factor inputs as cells detect and respond to injury over time. However, the main function of the Enh2+3 region may be to promote upregulation of Wnt gene expression in single cells that reside between damaged myofibers.

We next examined whether our reporters recapitulate *Wnt1* and *Wnt10b* gene expression. We were unable to determine the extent of overlap when double *in situ* hybridization for Wnt and *lacZ* was performed, because all cells were positive for *lacZ* signal (data not shown), possibly reflecting hybridization of the *in situ* probe to DNA from the multimerized transgene insertions (Bishop, 1996). As an alternative approach, we compared the overall expression patterns of *Wnt1* and *Wnt10b* with X-gal staining, and we observed that at early phases post-injury Wnt and reporter expression patterns are broadly similar. Specifically, X-gal staining from the Enh2- and Enh3-LacZ reporters was first found in single cells that reside between myofibers at day 1 (Fig. 2B,C), and *Wnt1* and *Wnt10b* transcripts were also visible in single cells located between injured myofibers at this time point (Fig. 1A,B). By days 3-5 post-injury, Enh2-LacZ was expressed in regenerating myofibers, and, similarly, Wnt expression was also found in regenerating myofiber nuclei (Fig. 1A,B). Given that Enh2+3 is not expressed in myofibers during regeneration (Fig. 2B,C), we do not know whether Enh2 regulates Wnt expression in myofibers perhaps with other enhancers, or if myofiber Wnt expression is controlled independently by other regulatory elements. Nonetheless, the appearance of *Wnt1* and *Wnt10b* in isolated single cells post-injury, and the similar appearance of single *lacZ*-positive cells in injured muscles, suggests that these enhancers could drive the initial phase of injury-responsive Wnt expression.

We asked whether publicly available data support the idea that Enh2 and Enh3 function as enhancers. UCSC Genome Browser data shows both sequences displaying distal enhancer-like signatures [DNAse1 hypersensitivity (Fig. S7A) in uninjured mouse lung and muscle, and H3K27 acetylation marks (Fig. S7B)]. A Gene Expression Omnibus dataset in which ATAC-seq was performed on muscle stem cells following BaCl$_2$ injury at 1, 16, 32 and 60 h (GSE189044; Dong et al., 2022), also shows peaks at Enhancers 2 and 3 (Fig. S7C). Additionally, *Wnt1* and *Wnt10b* may reside within a topologically associated domain, with possible contact points between their promoters and Enhancers 2 and 3 (Fig. S7D). Although these data do not match our experimental conditions exactly, these findings suggest that Enhancers 2 and 3 can function as regulatory sequences.

## Putative AP-1-binding sites are required for the injury response by Enhancer 3

The rapid induction of Enh3-LacZ in injured muscle (Fig. 2B) and lung (Fig. S6) suggests that this regulatory sequence might be particularly responsive to the initial stress of tissue damage. To determine which transcription factors activate Enh3-LacZ, we performed a bioinformatic analysis that identified two putative AP-1 sites (mm9 mouse genome alignment, sequence TGAGTCA; AP-1 consensus sequence TGAG/CTCA). One site was identified using GREAT (McLean et al., 2010). The other was identified by TFSEARCH (http://diyhpl.us/~bryan/irc/protocol-online/protocol-cache/TFSEARCH.html).

If AP-1 binding regulates Enh3, we hypothesized that it might be highly responsive to many injuries. We observed reporter activation in multiple injury contexts, including airway and tracheal damage induced by naphthalene (Hsu et al., 2014), liver injury by $CCl_4$ (Weber et al., 2003), skin injury by biopsy punch (Ansell et al., 2014) and heart tissue subjected to myocardial infarction (Kolk et al., 2009) (Fig. 3B). Uninjured tissues (Fig. 3B) and injured non-transgenic tissues (Fig. S8) displayed no β-galactosidase activity. In muscle, in situ hybridization revealed several AP-1 family members, Fosl1, Fosl2, c-fos (Fos) and fos-b (Fosb), that were dynamically expressed between 4 and 24 h post-injury (Fig. S9). These data show that Enh3 is highly injury responsive and suggest that immediate-early genes such as AP-1 may induce early and rapid activation of Enh3 post-injury.

To test whether predicted AP-1-binding sites are required for Enh3 injury responsiveness, we performed site-directed mutagenesis to alter the AP-1 consensus TGAG/CTCA sequences to GTCAGTC (Fig. 3A). In multiple genome alignments, only the house mouse carries two putative AP-1-binding sequences (Fig. S10A-C) and only one of the two predicted sites is highly conserved. We proceeded to mutate both sites to assess a requirement for AP-1 sequences in mouse Enh3, but only the conserved site could be involved in injury sensing.

Multiple stable adult Enh3 reporter lines carrying the mutant AP-1 sites fused to Hsp68-LacZ were generated. In muscle, injured tissues from wild-type (wt) Enh3 reporter mice displayed strong X-gal staining in single cells residing between damaged myofibers, but AP-1 mutant reporters failed to show X-gal staining (Fig. 3C). Occasionally, non-specifically stained single myofibers were observed (Fig. 3C, Line 44, asterisk).

Similarly, following injury in the lung, liver and skin, wt Enh3-LacZ displayed strong staining in damaged tissues (Fig. 3D), but mutant AP-1 reporter lines lacked robust injury-responsive reporter activity, resembling staining more reminiscent of uninjured wt Enh3-LacZ (Fig. 3D). Fig. 3D shows one representative example out of $n=8/8$ AP-1 mutant lines displaying reduced staining (see Fig. S11 for more lines; Table S3). Uninjured mutant AP-1 lines displayed no reporter activity (see Fig. S11 for a representative example). These data show that the putative AP-1 site(s) in Enhancer 3 are required for injury-responsive reporter activation.

### Deletion of Enhancers 2 and 3 results in loss of Wnt1 and Wnt10b expression

To determine whether Enhancers 2 and 3 are required in vivo, we deleted the Enh2+3 regulatory region using CRISPR-Cas9. Two separate mouse lines, named Δ1 and Δ2, were generated, in which the Δ1 deletion (1209 bp) was 206 bp larger than Δ2 (1003 bp) (Fig. 4A,B). Δ1/+ and Δ2/+ mice were crossed to each other to produce Δ1/Δ2 mice (Fig. 4B). This crossing scheme was used to minimize potential deleterious off-target effects of CRISPR-Cas9.

Δ1/Δ2 animals were viable, but Mendelian ratios were slightly lower for this class than predicted when assessed at weaning age (Table 2; 25% expected; 19.44% observed). The reduced percentage of Δ1/Δ2 animals also predicted that the remaining classes, if all equally healthy, each should produce ratios of ~26.85, but Δ1/+ and Δ2/+ animals were obtained at slightly lower frequencies than expected (Δ1/+ 25.66%; Δ2/+ 24.34%; Table 2). Loss of the enhancer region may be mildly detrimental in both homozygotes and heterozygotes. We did not assess why Δ1/Δ2 animals are lost during gestation or soon after birth, but Enh2-LacZ, Enh3-LacZ and Enh2+3-LacZ reporters were all expressed during development in various patterns (Fig. S12). We did not test whether sites of X-gal

staining overlap with Wnt1 or Wnt10b expression. lacZ expression in the dorsal CNS reminiscent of the 3′ 5.5 kb regulatory element that drives Wnt1 expression during embryogenesis (Echelard et al., 1994; Danielian et al., 1997; Reddy et al., 2001; Veltmaat et al., 2004) was not observed. Further studies are needed to understand how low levels of lethality are caused by Δ1/Δ2 deletion, and the possible role of Enh2 and Enh3 in embryonic Wnt1 and Wnt10b expression.

During the postnatal period, we observed no obvious lethality among pups, and, of the Δ1/Δ2 mice that were born, body weights appeared normal (Fig. 4C). We detected no obvious behavioral or morphological defects, suggesting that mice lacking Enhancers 2 and 3, once born, are mostly unaffected.

We performed mRNA in situ hybridization for Wnt1 and Wnt10b transcripts to look for loss of Wnt expression following deletion of Enhancers 2 and 3 at 3 days (Fig. 4D,F,G) and 5 days (Fig. 4E-G) post-injury. Visual examination of stained tissue did not reveal a dramatic decrease in transcripts, but quantification of Wnt1- and Wnt10b-positive nuclei showed a gradual decline in gene expression over 3-5 days (Fig. 4F,G). Comparison of Wnt1 and Wnt10b levels in wt versus Δ1/Δ2 tissue showed a slight decrease at day 3, most obvious in violin plots when all quantified fields were plotted individually. By day 5, there was a ~50% reduction in the number of Wnt-positive nuclei. Given that Enhancer 2+3-LacZ promotes transcriptional activity in single cells during earlier time points in response to $BaCl_2$-induced tissue damage, the gradual reduction in Wnt expression may reflect an inability of the Wnts to maintain sustained expression when Enhancers 2 and 3 are lost.

In contrast to Wnt1 and Wnt10b, Wnt5a expression appeared unchanged in injured wt versus Δ1/Δ2 tissues, as assessed by visual examination of transcript staining in sections (Fig. 4D,E), and when the number of Wnt5a positive nuclei was quantified (Fig. 4H). These data are consistent with the hypothesis that Enh2 and Enh3 specifically drive Wnt1 and Wnt10b expression, and Wnt5a expression in the muscle is regulated independently of Wnt1 and Wnt10b.

To investigate whether we could show by a different method that Enh2 and Enh3 control Wnt1 and Wnt10b expression levels, we performed pyrosequencing at 3 days post-injury. By crossing two different mouse strains (FVB and CAST/EIJ) carrying different single nucleotide polymorphisms (SNPs) that allowed us to distinguish Wnt1 and Wnt10b gene expression driven by the wt versus Δ1 or Δ2 alleles, we could measure how well Wnts were expressed when Enh2 and Enh3 were lost in the same muscle, under identical injury conditions. Wnt1 and Wnt10b genes were similarly expressed by both alleles in FVB$^{wt}$/CAST$^{wt}$ mice, but expression was reduced from the Δ1 or /Δ2 alleles in FVB$^{Δ1}$/CAST$^{wt}$ or FVB$^{Δ2}$/CAST$^{wt}$ mice (Fig. S13). Together, these data show that Enhancers 2 and 3 are injury-responsive regulatory sequences that likely regulate Wnt1 and Wnt10b in injured muscle; not only are the enhancers sufficient to drive injury-responsive reporter activity, but they are also partially necessary for Wnt1 and Wnt10b expression in damaged tissues.

### Deletion of Enhancers 2 and 3 produces a mild but significant increase in fat accumulation in injured muscle

When we examined whether Enh2+3 deletion might generate a phenotype, we noticed that Δ1/Δ2 TA muscles contained elevated fat at 14 days post-$BaCl_2$ injection. Fat was detected with an antibody to perilipin 1, which marks adipocytes (Blanchette-Mackie et al., 1995) (Fig. 5A-C, Fig. S14G-I). Uninjured muscles showed low perilipin staining (Fig. 5A, Fig. S14G) but both wt and Δ1/Δ2 muscles displayed perilipin signals at 14 days post-injury (Fig. 5B,C,

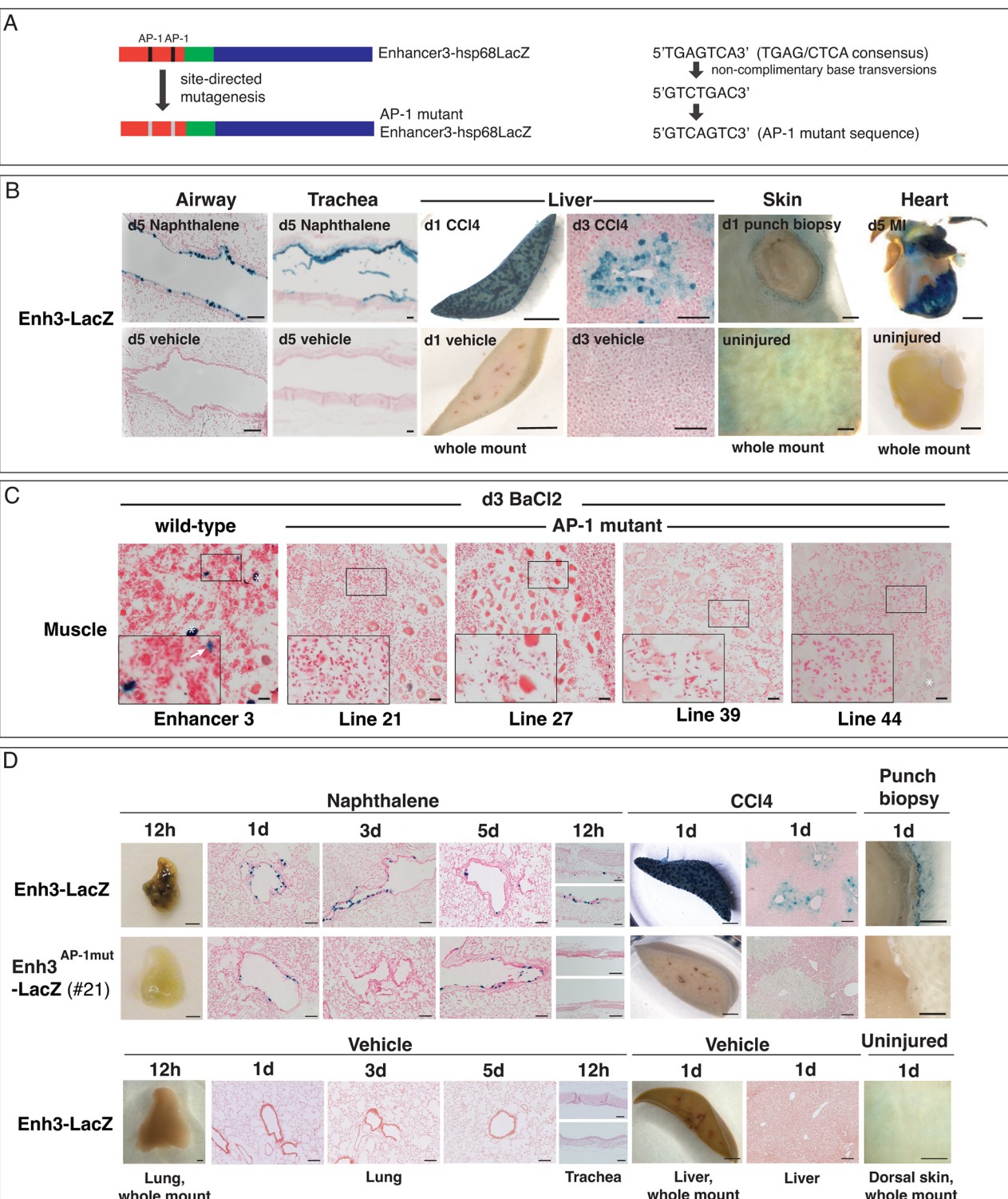

**Fig. 3. AP-1-binding sites are required for Enhancer 3 reporter activity in response to injury.** (A) Schematic of Enh3-LacZ showing two putative AP-1-binding sites and a mutated version with altered AP-1-binding sites. Predicted AP-1 binding sequences found in Enhancer 3 (TGAGTCA), the predicted sequence from non-complimentary base transversions (GTCTGAC), and the final mutant sequence (GTCAGTC) are shown. (B) X-gal staining in multiple injured tissues from Enhancer 3 reporter mice. Uninjured tissues are shown below. Naphthalene injury in airways and trachea at 5 days, CCl₄ injection in the liver at 1 day (whole mount) and 3 days (tissue section), punch biopsy of the skin at 1 day post-injury, and myocardial infarction (MI) of the heart at 5 days all produce robust Enh3-LacZ activation. Uninjured Enh3-LacZ tissues do not stain with X-gal. (C) Injured muscle from wt Enh3-LacZ compared to mutant AP-1 Enh3-LacZ. Four different mutant lines are shown. Boxes outline areas magnified in the lower left of each image. Arrow marks positive Enh3-LacZ signal. Asterisks mark background myofiber staining. (D) X-gal staining in lung, liver and skin to compare wt Enhancer 3 reporter activity versus a representative AP-1 mutant line (line #21 is shown). The wild-type Enhancer 3-LacZ displays X-gal staining in all injury contexts. The mutant reporter displays low expression that more closely resembles uninjured wt Enh3-LacZ (see Fig. S11 for more AP-1 mutant lines). d, days post-injury. Scale bars: 100 µm (B,D; lung, trachea, liver sections); 2 mm (B,D; lung, liver, skin whole mounts); 1 mm (B; heart); 20 µm (C).

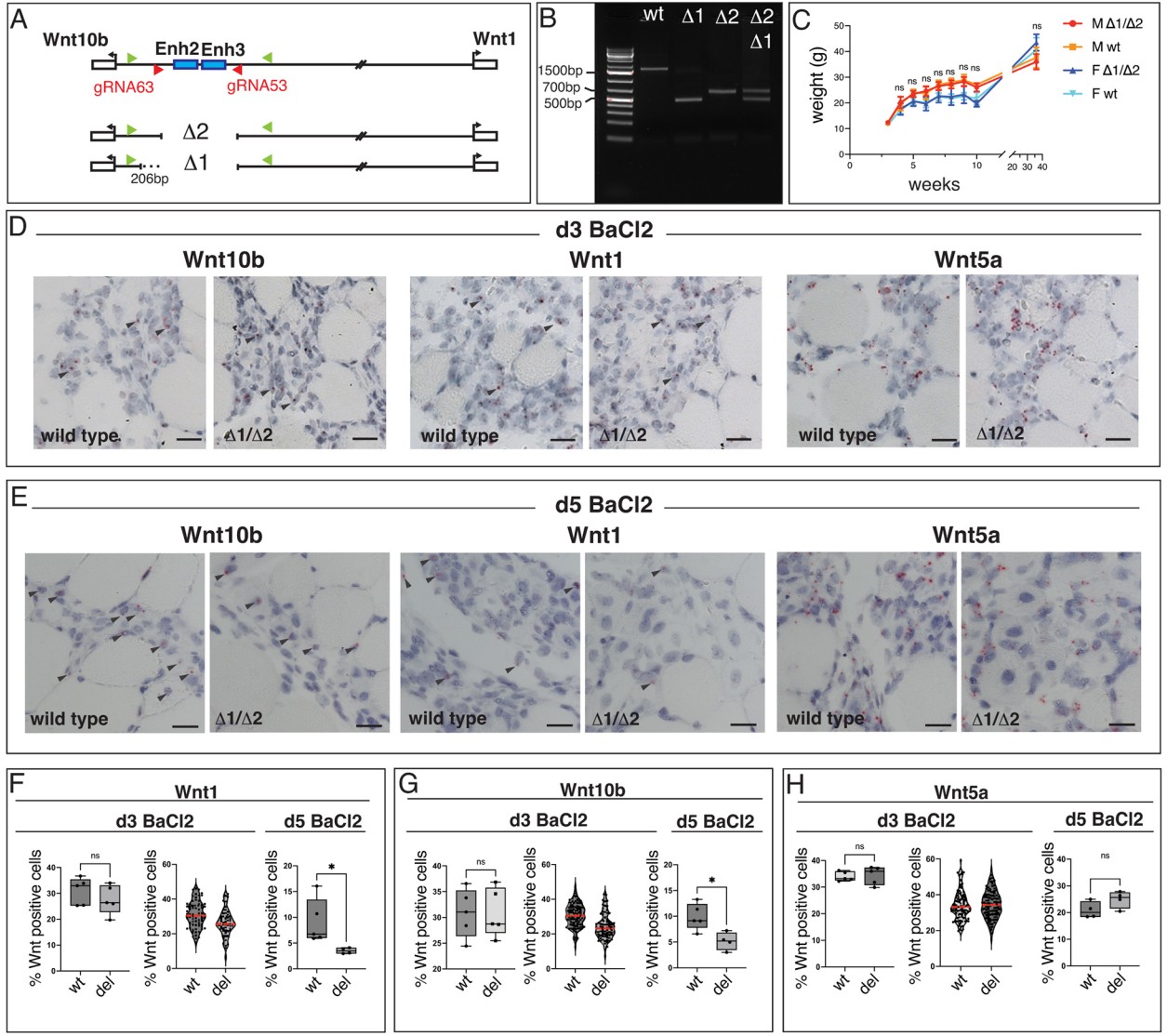

**Fig. 4. Loss of Enhancers 2 and 3 produces viable mice, but expression of Wnt1 and Wnt10b is reduced post-injury.** (A) Schematic showing CRISPR-Cas9-mediated deletion of Enhancers 2 and 3. Guide RNAs (red arrowheads) and PCR genotyping primers (green arrowheads) are shown. Two independent mouse lines carrying Enhancer 2+3 deletions are labeled Δ1 and Δ2. Δ1 is 206 bp larger than Δ2. (B) PCR reactions from genotyping primers flanking Enh2+3. Product sizes are 1.6 kb (wt), 500 bp (Δ1) and 700 bp (Δ2). Δ1/Δ2 mice produce PCR bands that run as a doublet. (C) Δ1/Δ2 and wt mice display similar overall weights. Males (M) and females (F) were followed between 2.5 and 35 weeks. ($n$=9 wt M, $n$=9 Δ1/Δ2 M; $n$=11 wt F, $n$=10 Δ1/Δ2 F) (D,E) *Wnt1*, *Wnt10b* and *Wnt5a* expression assessed by mRNA *in situ* hybridization in adult TA muscles. Arrowheads mark examples of transcripts. (D) Staining at 3 days post-injury. $n$>3 mice. (E) Staining at 5 days post-injury. $n$>3 mice. (F-H) Quantification of Wnt-expressing nuclei at 3 and 5 days post-injury. Each dot on the box plots represents mean percentage of Wnt-positive nuclei normalized to total number of nuclei counted per field for each mouse. Violin plots display all quantified fields. (F) A slight decrease in *Wnt1*-positive cells is detected at 3 days followed by significant reduction at d5. (G) A slight decrease in *Wnt10b*-positive cells is detected at d3, followed by significant reduction at d5. (H) Wnt5a expression shows no decline over time in Δ1/Δ2 muscles compared to wt. Box limits mark the 25th and 75th percentiles, whiskers mark minimum to maximum values, and horizontal line marks the median; red lines mark the median in violin plots. $n$=5 wt, $n$=5 del for *Wnt1*, *Wnt10b* and *Wnt5a* at d3; $n$=5 wt, $n$=4 del for *Wnt1*, *Wnt10b* at d5; $n$=4 wt, $n$=4 del for *Wnt5a* at d5. *Wnt1* d3, Mann–Whitney test; *Wnt1* d5, *$P$=0.0159 Mann–Whitney test; *Wnt10b* d3, unpaired *t*-test; *Wnt10b* d5, *$P$=0.0157 unpaired *t*-test; *Wnt5a* d3, Welch's *t*-test; *Wnt5a* d5, unpaired *t*-test. d, days post-injury; del, Δ1/Δ2; ns, not significant. Scale bars: 20 μm.

Fig. S14H,I). The highest levels of perilipin staining observed in wt muscles was reduced compared to that of Δ1/Δ2 muscles (Fig. 5D,E, Fig. S14C,H,I). Representative muscles spanning the range of fatty infiltration observed for each genotype are shown in Fig. S14H,I.

We quantified adipogenesis in injured muscle of wt and Δ1/Δ2 mice at 14 days post-injury. We measured adipogenesis levels over total section area and generated 'percentage adipogenesis' values over 30-50 sections per muscle, because fatty infiltration levels can fluctuate over injured muscle length (Fig. S14A). Injury can

also vary between BaCl₂-injected muscles, so we calculated a 'percentage injury' value for each sample by quantifying areas displaying centrally located nuclei (Folker and Baylies, 2013; Hardy et al., 2016; Meyer, 2018) and normalizing the values to the total section area. We further generated an 'adipogenesis index' by normalizing the percentage adipogenesis value to the percentage injury value. The adipogenesis index value was significantly higher in Δ1/Δ2 muscles compared to wt muscles (Fig. 5D).

When we plotted mean percentage adipogenesis values against percentage injury values, it was evident that, although increased

**Table 2. Mendelian ratios of mice carrying deletions of Enhancers 2 and 3 (cross: Δ1/+×Δ2/+)**

| Genotype | Expected [n (%)] | Observed [n (%)] |
|---|---|---|
| +/+ (wild type) | 189 (25%) | 231 (30.56%) |
| Δ1/+ (heterozygote, line 1) | 189 (25%) | 194 (25.66%) |
| Δ2/+ (heterozygote, line 2) | 189 (25%) | 184 (24.34%) |
| Δ1/Δ2 (homozygote) | 189 (25%) | 147 (19.44) |
| | 756 (100%) total | 756 (100%) total |

Chi-square=18.93; P=0.0003.

injury positively correlated with increased fat accumulation, Δ1/Δ2 muscles displayed higher and more variable levels of fat (Fig. 5E). Simple linear regression analysis of wt versus Δ1/Δ2 samples did not show significantly different slopes between the two genotypes but showed a significant difference in elevation (intercept) of the lines. This suggests that loss of Enhancers 2 and 3 and the accompanying decrease in Wnt expression results in more fatty infiltration in Δ1/Δ2 muscle compared to wt. Additional visualization of the data by individual mouse or by mean percentage adipogenesis values also supports this observation (Fig. S14B-D).

We further quantified adipogenesis using Hematoxylin and Eosin (H&E)-stained sections in which adipocytes appear as round 'holes' wherever fat is present (Biltz and Meyer, 2017). This method, in which adipocyte area in wt versus Δ1/Δ2 muscles was compared showed elevated fat accumulation in injured Δ1/Δ2 muscles compared to wt (Fig. S14E). We additionally measured expression levels of well-known adipogenic regulators, *Cebpa* and *Pparg*, at 7 days post-injury by RT-qPCR. Both genes were elevated in Δ1/Δ2 samples compared to wt (Fig. 5F,G), although we did not detect statistical significance.

To determine whether increased adipogenesis in Δ1/Δ2 muscles stems from reduction in Wnt signaling, we measured *Axin2* expression, a downstream target of the Wnt/β-catenin pathway (Jho et al., 2002; Lustig et al., 2002). We observed reduced *Axin2* expression at 5 days post-injury in Δ1/Δ2 muscles compared to wt, but the results were not statistically significant (Fig. S14F). *Axin2* expression was still abundant within wt and Δ1/Δ2 muscles (Ct values=~26-27), suggesting that other Wnts likely maintain signaling levels even when *Wnt1* and *Wnt10b* are lost.

Adipogenesis in the muscle is thought to be a transient process that resolves quickly after injury within a few days (Wagatsuma, 2007; Lukjanenko et al., 2013), so we assessed adipogenesis after a longer recovery. We injured the TA muscle of wt and Δ1/Δ2 mice with BaCl$_2$ and waited for 45 days. We observed more adipocytes in wt muscles post-injury compared to uninjured muscles (Fig. 5H,I) indicating that adipogenesis does not necessarily completely resolve by 45 days. Moreover, Δ1/Δ2 muscles continued to display elevated fatty infiltration compared to wt (Fig. 5I,J, Fig. S15A-D), although this effect was most pronounced in two out of the eight mice analyzed (Fig. S15C,D). These data show that, even at 45 days, there is lingering fatty infiltration that can be detected at higher levels in Δ1/Δ2 muscles compared to wt.

Since muscle regeneration involves reciprocal cell–cell interactions between satellite cells, FAPs and other cell types (Joe et al., 2010; Uezumi et al., 2010; Lemos et al., 2015; Wosczyna et al., 2019), we examined whether changes in FAP functions due to enhancer deletion might manifest as additional defects. We examined expression of *Pdgfra*, *Pax7* and *Cebpb* using RT-qPCR. Both *Pdgfra* and *Pax7* increased post-injury as satellite cells and FAPs proliferate (Joe et al., 2010) but no differences were observed between wt and Δ1/Δ2 tissues (Fig. S16A,B). *Cebpb* expression

normally decreases following injury and upon satellite cell exit from quiescence (Lala-Tabbert et al., 2021); wt and Δ1/Δ2 muscles displayed similar *Cebpb* expression (Fig. S16C). We examined myofiber cross-sectional area (CSA) across whole muscle sections (Fig. S16D,E) and also quantified CSAs only in injured myofibers, defined as those exhibiting centrally located nuclei (Fig.S16F), but no differences were observed. Finally, collagen deposition within injured muscles at 14 days post-injury was also similar between wt and Δ1/Δ2 muscles (Fig. S16G,N).

Since CRISPR-Cas9-mediated deletion of Enhancers 2 and 3 is not a conditional allele and produces animals constitutively lacking the enhancer region from birth, we also investigated whether muscle defects were detectable prior to injury. We surveyed basal levels of *Pdgfra* (Fig. S16H), *Pax7* (Fig. S16I), myofiber CSAs (Fig. S16J,K), myosin heavy chain expression (*Myh1*, *Myh2*, *Myh4*) (Fig. S16L) and collagen deposition (Fig. S16M,N), but we detected no significant differences between wt and Δ1/Δ2 muscles.

Finally, we examined whether loss of Enhancers 2 and 3 only affects muscle, or whether defects are found in other tissues. We saw no obvious phenotype when airways were injured using naphthalene (data not shown). Airway cells were restored and regeneration appeared normal, but it is possible that we missed subtle defects. In the skin, Enhancer 2 and 3 reporters were expressed following punch biopsy (Fig. S17A), but, unlike the muscle and lung, *Wnt1* and *Wnt10b* were not detected in areas surrounding the injury (Fig. S17B). Moreover, Enhancer 2+3-LacZ reporter mice failed to stain with X-gal (Fig. S17A). Enhancers 2 and 3 may each bind transcription factors induced by injury and drive reporter expression, but together they do not drive gene expression. Skin punch biopsy experiments also produced no detectable wound closure defects (Fig. S17C,D). These results demonstrate that activation of individual enhancers may not always necessarily translate into functional target gene expression, underlining the importance of tissue context.

Together, these data show that loss of Enhancers 2 and 3 produces a phenotype in which elevated fatty infiltration is observed following injury in muscle. The presence of fat in muscle is often observed in aged, disused or diseased muscle, or in experimental contexts in which muscle regeneration is perturbed (Sciorati et al., 2015; Pagano et al., 2018; Wang et al., 2024). Our results suggest that the Enh2/3 regulatory region, a ~1 kb sequence that regulates *Wnt1* and *Wnt10b* in response to injury, influences levels of adipogenesis produced as the muscles recover from tissue damage.

## DISCUSSION

We have identified an enhancer region residing between *Wnt1* and *Wnt10b* genes in the mouse that responds to tissue injury and functions during muscle regeneration. Using an *in vivo* approach to study a single injury-responsive enhancer locus, we describe its spatial and temporal activities, examine its requirement for Wnt expression and muscle recovery post-injury, and extend our understanding of how Wnts activated in response to injury function during tissue regeneration.

Evidence that *Wnt1* and *Wnt10b* share a single injury-responsive regulatory region is supported by the observation that expression of both Wnts is greatly reduced when Enhancers 2 and 3 are deleted (Fig. 4D-G, Fig. S13). Co-expression of *Wnt1* and *Wnt10b* in the same cells is detectable but infrequent (Fig. 1G), perhaps resulting from differences in mRNA stability or the duration and frequency of enhancer contacts with each Wnt promoter. Understanding how Enhancers 2 and 3 engage with promoters and other regulatory elements to drive *Wnt1* and *Wnt10b* expression post-injury in its full genomic context will require further studies.

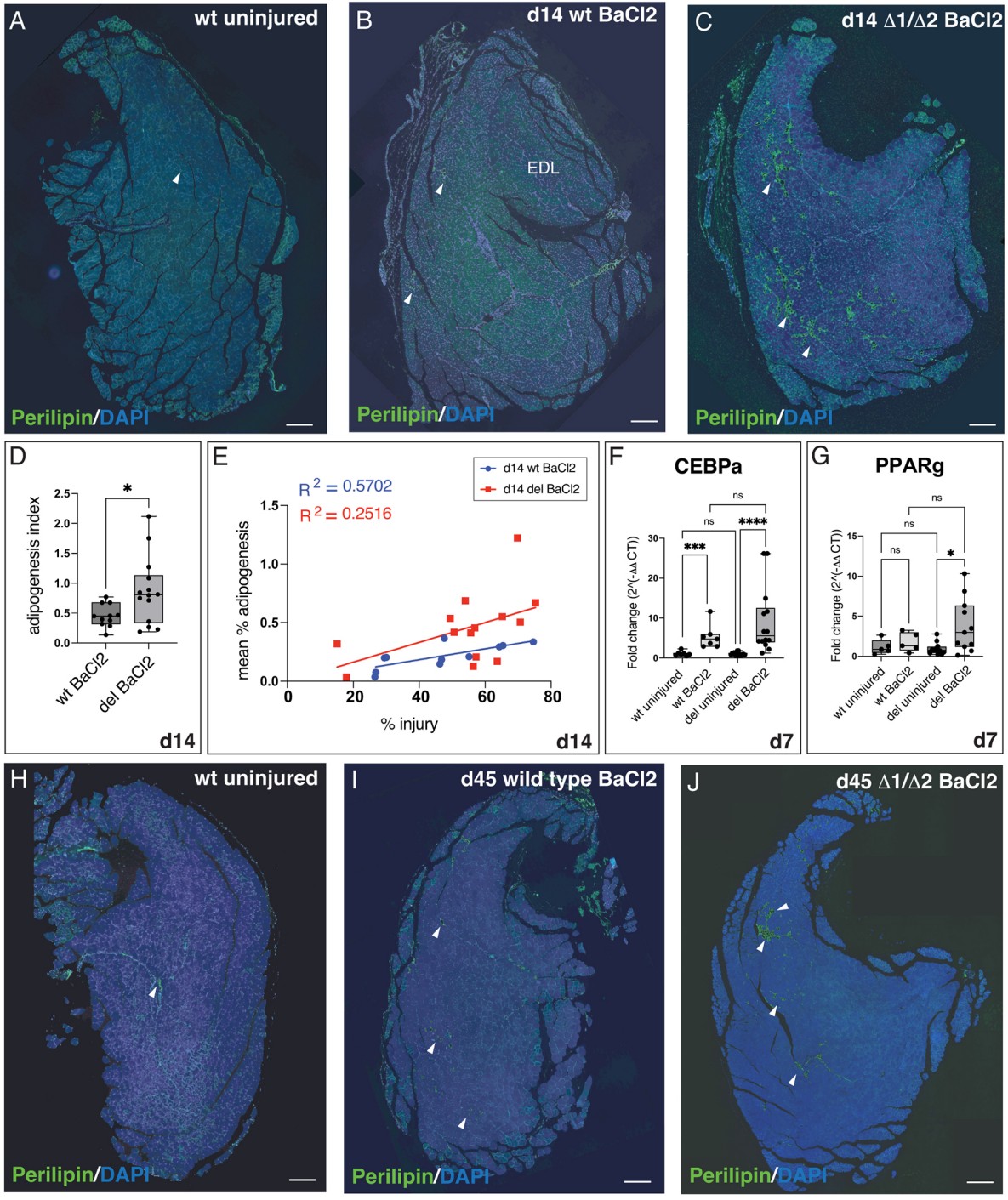

**Fig. 5. Mice lacking Enhancers 2 and 3 display elevated intramuscular adipogenesis.** (A-C) Perilipin-stained TA muscle sections. Arrowheads mark perilipin-positive areas. (A) Uninjured wt muscle shows low adipogenesis. (B) Wild-type muscles display small patches of adipocytes. (C) Δ1/Δ2 muscles show larger adipocyte areas. B and C represent similarly injured muscles (∼75% injury). (D) Box and whisker plot showing the 'adipogenesis index' (mean percentage adipogenesis values normalized to percentage injury values). $n$=11 wt, $n$=14 Δ1/ Δ2. *$P$=0.0213 (Welch's $t$-test). (E) Scatter plot of mean percentage adipogenesis for wt and Δ1/Δ2 muscles plotted against levels of injury exhibited by BaCl$_2$-injected muscles at 14 days. For wt muscles, a moderate positive correlation between percentage injury and percentage adipogenesis is seen [$R^2$=0.5702, F(1,9)=11.94, $P$=0.0072]. For Δ1/Δ2 data, simple linear regression analysis shows a poorer fit [$R^2$=0.2516, F(1,12)=4.04, $P$=0.0676]. Slopes of the regression lines are not significantly different between the two genotypes ($P$=0.4596) but the difference in intercept is significant, showing that adipogenesis is independent of injury level, but dependent on genotype [F(1, 21)=4.727, $P$=0.0407]. $n$=11 wt, $n$=14 Δ1/Δ2 muscles. (F) RT-qPCR of *Cebpa* expression following injury at 7 days. Uninjured: $n$=7 wt, $n$=14 Δ1/Δ2; BaCl$_2$ injured: $n$=7 wt, $n$=14 Δ1/Δ2. ***$P$=0.0003 (wt uninjured versus wt BaCl$_2$); ****$P$<0.0001 (Δ1/Δ2 uninjured versus Δ1/Δ2 BaCl$_2$) (one-way ANOVA with Kruskal–Wallis test). (G) RT-qPCR of *Pparg* expression at 7 days post-injury. Uninjured: $n$=5 wt, $n$=11 Δ1/Δ2; BaCl$_2$ injured: $n$=5 wt, $n$=11 Δ1/Δ2. *$P$=0.0221 (Δ1/Δ2 uninjured versus Δ1/Δ2 BaCl$_2$; ordinary one-way ANOVA). (H-J) Perilipin-stained muscle 45 days post-injury (arrowheads). (H) Representative example of perilipin staining in uninjured wt muscle. (I,J) Perilipin-stained wt (I) and Δ1/Δ2 (J) muscle at 45 days post-injury. Images representative of average perilipin staining levels at this time point are shown [calculated as 0.17% (wt) 0.67% (Δ1/Δ2)]. Overall average is 0.26% (wt), 0.74% (Δ1/Δ2). In D,F,G, box limits mark the 25th and 75th percentiles, whiskers mark minimum to maximum values, and horizontal line marks the median. In F,G, statistics were performed on ΔΔCT values. d, days post-injury; ns, not significant. Scale bars: 200 μm.

To assess the importance of Enhancers 2 and 3, we made germline deletions of this regulatory region. Uninjured animals lacking the enhancers are viable, fertile and display no obvious defects (Fig. 4C, Fig. S16), except for a slight reduction in expected frequency of progeny obtained from Mendelian crosses (Table 2). Muscles post-injury, however, show elevated levels of adipocytes in $\Delta1/\Delta2$ tissues compared to wt (Fig. 5, Figs S14, S15) revealing a mild but detectable change in fatty infiltration. Penetrance of the phenotype was variable, but this is not surprising given the deletion of regulatory, rather than coding sequence. Only a few examples of injury-responsive enhancers that produce phenotypes when deleted or mutated have been reported (Hewitt et al., 2017; Soukup et al., 2019; Suzuki et al., 2019; Harris et al., 2020; Wang et al., 2020; Zlatanova et al., 2023), and redundancy, a common feature of developmental enhancers (Osterwalder et al., 2018), may make it difficult to detect enhancer functions in adult tissues. The phenotype we present here, to our knowledge, is the first to describe an injury-responsive regulatory region in mouse that influences both Wnt expression post-injury and fatty infiltration in regenerating muscle.

Muscle regeneration involves complex cell-to-cell signaling interactions between FAPs, satellite cells and immune cells (Joe et al., 2010; Uezumi et al., 2010; Lemos et al., 2015; Wosczyna et al., 2019). Post-muscle injury, we observe *Wnt1* and *Wnt10b* in small cells scattered between myofibers that likely includes FAPs and other cell types (Fig. 1E,F). In the visceral fat, *Wnt10b* inhibits adipogenesis (Longo et al., 2004; Christodoulides et al., 2006; Cawthorn et al., 2012) and *Wnt10b*$^{-/-}$ mutants display fat in injured muscles (Vertino et al., 2005). A simple model is that *Wnt1* and *Wnt10b* prevent FAPs from making adipocytes. When *Wnt1* and *Wnt10b* expression is reduced, as observed in $\Delta1/\Delta2$ tissues, adipocytes may appear due to decreased suppressive signals. Other Wnts are induced following muscle injury, however, and several play important roles in regeneration (Van Winkle et al., 1995; Le Grand et al., 2009; Reggio et al., 2020). Whether their presence in $\Delta1/\Delta2$ muscles compensates for *Wnt1* and *Wnt10b* loss and explains the relatively mild adipogenic phenotypes we observe is unknown. Further work will be required to understand the specificity versus redundancy of different Wnt ligands, the specific cell–cell interactions between different muscle cell subpopulations post-injury, and how *Wnt1* and *Wnt10b* expression is regulated in different temporal and spatial domains within muscle, both by Enhancers 2 and 3, and likely by additional regulatory elements.

We acknowledge several limitations of our study. First, we did not rescue the adipogenic phenotype to demonstrate that it stems from loss of *Wnt1* and *Wnt10b*. Designing a robust rescue system for two Wnts *in vivo* is challenging, and Wnt overexpression by tools such as AAV injections incur additional tissue damage. *Wnt10b* plays an inhibitory role in visceral fat by preventing pre-adipocyte differentiation into adipocytes (Christodoulides et al., 2009; Perkins et al., 2023). Expressing *Wnt1 in vitro* has a similar inhibitory activity (Ross et al., 2000), although an endogenous adipogenic role for *Wnt1* has not been reported. In muscle, *in vitro* experiments failed to show a Wnt10b effect on isolated mouse FAPs (Reggio et al., 2020), but *Wnt10b*$^{-/-}$ mutants display excess adipocytes following muscle injury *in vivo* (Vertino et al., 2005), suggesting that *Wnt10b* (and possibly *Wnt1*) are inhibitory for adipocyte differentiation in muscle.

We also could not perform genetic sensitization experiments to assess the role of other genes that might interact with *Wnt1* and *Wnt10b* to regulate adipogenesis. We introduced one copy of a β-catenin (*Ctnnb1*) $\Delta$ allele (Brault et al., 2001) and one copy of a *Wntless* (*Wls*) $\Delta$ allele (Carpenter et al., 2010) (both in mixed backgrounds) into $\Delta1/\Delta2$ mice (FVB background) to determine whether reducing β-catenin or reducing all Wnts might worsen muscle fatty infiltration. The results were uninterpretable, with both wt and $\Delta1/\Delta2$ animals displaying variable adipogenesis. We discovered, perhaps not surprisingly, that genetic background profoundly affects enhancer phenotypes, particularly when the defects observed are subtle.

Finally, we examined whether older $\Delta1/\Delta2$ mice might display more severe defects, since aging increases fatty infiltration in muscle (Zhu et al., 2024). We performed experiments twice on 16- to 24-month-old wt and $\Delta1/\Delta2$ animals, but the mice did not survive. BaCl$_2$ can induce arrhythmias (Mattila et al., 1986; Jung et al., 2019), which perhaps old mice are less able to tolerate. When we reduced BaCl$_2$ dosage on a third cohort, the animals survived but displayed poor injury. Evaluating the effects of Enhancer 2 and 3 deletions in aged mice will require further experiments.

From work done in invertebrates, it has been proposed that an ancient and conserved injury-responsive gene regulatory network might exist for Wnt gene regulation (Srivastava, 2021). *Drosophila* harbors a cluster of Wnts that reside on the same chromosome consisting of *Wnt4*, *wingless*, *Wnt6* and *Wnt10*. A damage-responsive element between *wingless* and *Wnt6* utilizes AP-1 binding to drive Wnt expression (Harris et al., 2020), perhaps reminiscent of the close gene arrangement of the putative AP-1-binding sites and Wnts seen in mouse. Given Wnt induction in virtually all injury contexts that have been described to date, are there common mechanisms that activate different Wnts post-injury that span across invertebrates to vertebrates? Definitive answers to this possibility will require further functional studies in other non-mammalian species.

## MATERIALS AND METHODS
### Mouse husbandry and mouse lines
All animal experiments and methods were approved by the Institutional Animal Care and Use Committee (IACUC) at Stanford University. Experiments were performed on FVB mice (Charles River) at 12-18 weeks of age. Both males and females were used in this study. Data were combined, unless differences were noted and specified in the text. For timed matings, plugs were checked in the morning before 09.00 h and noon was considered embryonic day 0.5 (days post-coitum). Wnt1-LacZ mice (Echelard et al., 1994) and Castaneus mice (CAST/Eij strain) were obtained from The Jackson Laboratory (Wnt1-LacZ, stock# 002865; CAST/Eij, stock #000928). Transgenic mice were generated by the Stanford Transgenic Knockout and Tumor Model Center and Cyagen Biosciences (Santa Clara, CA, USA) through pronuclear injection of transgene constructs into FVB embryos (Hogan, 1994). CRISPR-Cas9-mediated deletion of Enhancers 2 and 3 was carried out by the Stanford Transgenic Knockout and Tumor Model Center.

### Injury models
Naphthalene injuries were performed before 09.00 h via a single intraperitoneal injection of 0.2 μm sterile-filtered naphthalene (Sigma-Aldrich, 147141) dissolved in corn oil (Sigma-Aldrich, C8267) at a concentration of 275 mg/kg, as previously described (Reynolds et al., 2000; Hong et al., 2001). For muscle injuries, mice were anesthetized with isoflurane and given either buprenorphine SR (1 mg/kg, subcutaneously) or Ethiqa XR (extended-release buprenorphine) (3.25 mg/kg subcutaneously) prior to injection of BaCl$_2$; 50 μl of sterile, filtered 1.2% BaCl$_2$ (Sigma-Aldrich, 342920) solution was injected into the left TA muscle. The muscle was poked to distribute the BaCl$_2$ as described by Wosczyna et al. (2019). The right TA muscle served as the uninjured control. For injuries of pancreatic islets (Furman, 2021), animals were initially fasted for 6 h, and their weights and blood glucose levels were measured. Then, a one-time injection of 175 mg/kg streptozotocin (Sigma-Aldrich, 0130) was administered intraperitoneally. After injection, animals were given sucrose water (15 g/l) for 48 h, and blood glucose levels were monitored at 2, 5 and 7 days. High blood glucose readings by 5-7 days indicated successful injury

to the pancreatic islets, and pancreatic tissues were isolated at 7 days. For liver injury using CCl₄, sterile-filtered CCl₄ (Sigma-Aldrich, 289116) dissolved in corn oil (Sigma-Aldrich, C8267) at a 1:4 ratio was injected at a dose of 1 ml/kg. Sterile corn oil served as the vehicle control (Zhao et al., 2019). Punch biopsy of the skin was performed similarly to the procedure described by Lindsey and Dunnwald (2020). Briefly, mice anesthetized with isoflurane were shaved and the skin was cleaned with 10% Povidone Iodine (PDI, B40600). Topical lidocaine/prilocaine cream (Alembic) was administered where the biopsy site would be located. Mice were then subcutaneously given carpofen (5 mg/kg) or Ethiqa XR (3.25 mg/kg). A 5 mm biopsy punch (Acuderm) was used to excise a full-thickness skin sample.

## mRNA *in situ* hybridization

Tissues were fixed in 4% neutral-buffered formalin for 24 h at room temperature and then processed for paraffin embedding. Sections were cut at 5 μm thickness, and mRNA *in situ* hybridization using either the RNAscope 2.5 HD Assay-RED or RNAscope 2.5 HD Duplex Assay kits was performed according to the manufacturer's instructions (ACDBio). Wnt Probes for *in situ* hybridization were Mm-Wnt1 (401091, NM_021279.4, region 1204-2325); Mm-Wnt2 (313601, NM_023653, region 857-2086); Mm-Wnt2b (405031, NM_009520.3, region 1307-2441); Mm-Wnt3 (312241, NM_009521.2, region 134-1577); Mm-Wnt3a (405041, NM-009522.2, region 667-1634); Mm-Wnt4 (401101, NM_ 009523.2, region 2147-3150); Mm-Wnt5a (316791, NM_009524.3, region 200-1431); Mm-Mm-Wnt5b (405051, NM_001271757.1, region 319-1807); Mm-Wnt6 (401111, NM_009526.3, region 780-2026); Mm-Wnt7a (401121, NM_009527.3, region 1811-3013); Mm-Wnt7b (401131, NM_009528.3, region 1597-2839); Mm-Wnt8a (405061, NM_009290.2, region 180-1458); Mm-Wnt8b (405071, NM_011720.3, region 2279-3217); Mm-Wnt9a (405081, NM_139298.2, region 1546-2495); Mm-Wnt9b (405091, NM_011719, region 727-1616); Mm-Wnt10a (401061, NM_009518.2, region 479-1948); Mm-Wnt10b (401071, NM_011718.2, region 989-2133); Mm-Wnt11 (405021, NM_009519.2, region 818-1643); Mm-Wnt16 (401081, NM_053116.4, region 453-1635). Additional probes were: Mm-Pax7 (314181-C2, NM_011039.2, region 602-1534); Mm-PDGFRa (480661-C2, NM_011058.2, region 223-1161), Mm-Adgre (460651-C2, NM_ 010130.4, region 85-1026), Mm-FosL1 (421981, NM_010235.2, region 2-1386), Mm-FosL2-C2 (421991-C2, NM_008037.4, region 1612-2669), Mm-Fosb (539721, NM_008036.2, region 370-1302); Mm-Fos (316921-C2, NM_010234.2, region 407-1427), MmWnt10b-C2 (401071-C2, NM_011718.2, region 989-2133), Mm-Wnt1-C2 (401091-C2, NM_021279.4, region 1204-2325). Every *in situ* experiment was run with a negative and positive control: *dapB* (negative control probe, EF _191515, region 414–862) and *Polr2a* ( positive control probe, NM_009089.2, region 2802–3678) or 2.5 Duplex Positive Control Probe (321651), 2-Plex Negative Control Probe (320751). Images were captured on a Zeiss Axio Imager.Z2 microscope. Brightness was adjusted globally on images using Adobe Photoshop to improve visualization in figures. For quantification of the number of nuclei expressing different Wnts, Gill's Hematoxylin-stained nuclei (StatLab HXGHE1LT) and nuclei stained with the different *in situ* probes were counted using the multi-point tool in ImageJ/Fiji. Values for the percentage of Wnt-positive nuclei were calculated as the total number of Wnt-positive nuclei divided by the total number of nuclei counted in each field×100. All data were analyzed using GraphPad (Prism) software.

## Construction of transgenic mice

The plasmid used for construction of transgenic mice was developed by Kothary et al. (1989) and was a gift of Dr David Kingsley (Stanford University, Stanford, CA, USA). For the generation of reporter mice, BAC RP2394M12 DNA (CHORI) and BAC RP23236-E21(CHORI, for the 6 kb construct) were used as templates. Enhancer segments were amplified using PCR primers designed with NotI restriction sites. Coordinates for the different enhancers (mm9 assembly) are: 6 kb: 98608459-98614672; Enhancer 1: 98609340-98609567; Enhancer 2: 98610564-98610827; Enhancer 3: 98610878-98611376.

Primers for the enhancer sequences were: 6kbF NotI: 5′-ATAAGAAT-GCGGCCGCAGATGTAGAACTCTCAGCTCCTCCTGAACC-3′; 6kbR

NotI: 5′-ATAAGAATGCGGCCGCGTCTGTCTGTCTGTCTGTCTGTCT-GTCTGT-3′; Enhancer1F NotI: 5′-ATAAGAATGCGGCCGCCGGCTCT-GGATCTATGTGACTT-3′; Enhancer1R NotI: 5′-ATAAGAATGCGGCC-GCAGGCAAAGGCACAAACTACG-3′; Enhancer2F NotI: 5′-ATAAGA-ATGCGGCCGCCCTCTCTGAAGTCTTGCTCTTTG-3′; Enhancer2R NotI: 5′-ATAAGAATGCGGCCGCTCCCCCTCTCATCTAGGTCTC-3′; Enhancer3F NotI: 5′-ATAAGAATGCGGCCGCTAGTAGACAAGGGGG-TAACATTCTG-3′; Enhancer3R NotI: 5′-ATAAGAATGCGGCCGCCA-TCTTCCTTTTGATGAGAAAAGTG-3′. Enhancer 2/3 was cloned using the Enhancer2F NotI and Enhancer 3F NotI primers. The Enhancer 2 element is 264 bp. The Enhancer 3 reporter carries a 524 bp element.

PCR fragments were cloned into the NotI sites of Hsp68-LacZ and sequenced. Plasmids for injection were isolated using the QIAGEN MaxiPrep kit (QIAGEN, 12662) and submitted for linearization and injection by the Stanford Transgenic Knockout and Tumor Model Center and by Cyagen Biosciences. Enzymes for linearization were designed to eliminate as much plasmid sequence as possible and were as follows: 6 kb and Enhancer 2: ApaI/NaeI; Enhancer 1: DraI/XhoI; Enhancer 3, Enhancer 2/3 and Hsp68-LacZ: NaeI/XhoI. For Enhancer3-eGFP, the eGFP sequence from pEGFP-N1 (Clontech) and the Enhancer3-hsp68 promoter sequence were cloned into pBluescript. An EcoRI/EcoRI PCR fragment containing the enhancer and basal promoter obtained from Enhancer3-LacZ was generated and fused to an EcoRI/XhoI PCR fragment carrying eGFP and an SV40 polyA tail. The plasmid sequenced and then linearized with PvuI/XhoI prior to pronuclear injection.

Transgenic mice were identified by genotyping with primers to *lacZ* (Galp1: 5′-TTTACAACGTCGTGACTG-3′; Galp2: 5′- TGATTTGTGTA-GTCGGTT-3′; Dr W.T. O'Brien, University of Pennsylvania, PA, USA). Transgenic mice carrying eGFP were genotyped using primers GFP03 (5′-ACGGCAAGCTGACCCTGAAGT-3′) and GFP04 (5′-GCTTCTCGT-TGGGGTCTTTGC-3′). To generate adult transgenic lines, *lacZ*- or GFP-positive pups were crossed and tested through at least the G2 generation to eliminate mosaicism, separate instances of multiple insertions on different chromosomes, and to confirm Mendelian segregation of single transgenic loci.

Because we were searching for injury-responsive enhancers, it was important to distinguish between the inability to be activated by injury versus lack of expression due to integration into a silenced locus. Because the mouse brain is thought to express over 80% of genes within the genome (Lein et al., 2007; Thompson et al., 2014), we reasoned that the brain would be one of the most transcriptionally active organs within the body, and that if our transgene failed to express in the brain, it could indicate that it was silenced. We therefore defined silencing of the transgene as those lines that failed to express in uninjured adult brain tissue and those that were not induced following injury. These animals were removed from the study and not analyzed further.

## X-gal staining

For whole-mount staining, tissues were harvested and fixed at room temperature in 4% paraformaldehyde (PFA) for 10 min. Lungs were gently inflated with 4% PFA using a 5 ml syringe attached to a 25 G needle prior to rocking in PFA. A short fixation was crucial to avoid losing activity of the β-galactosidase enzyme due to over-fixation. Tissues were washed in PBS containing 2 mM MgCl₂, 0.01% deoxycholate and 0.02% Nonidet P-40, and then placed in X-gal staining buffer [1 mg/ml X-gal (5-bromo-4-chloro-3-indolyl-b-D-galactopyranoside, Sigma-Aldrich), 4 mm K₃[Fe(CN)₆], 4 mm K₄[Fe(CN)₆·3H₂O], 2 mM MgCl₂, 0.01% deoxycholate and 0.02% Nonidet P-40 in PBS). After staining overnight, tissues were post-fixed in 4% PFA for 24 h and then washed in PBS for examination as whole-mount specimens on an M80 Leica dissecting scope, or for further processing into paraffin blocks, which were cut at 5 μm thickness. For X-gal staining on tissue sections, isolated tissues were fixed in 1% PFA for 1 h on ice and then washed three times in PBS for 5 min at room temperature before being placed in 30% sucrose overnight with rocking at 4°C. Specimens were then embedded in OCT embedding medium (Tissue-Tek) and frozen for cryosectioning. Tissues were sectioned at 8 μm, dried briefly, fixed in 4% PFA at room temperature for 5 min, and then placed in X-gal staining solution. To compare relative activities of the Enhancer 2 and 3 reporters,

slides were stained for 2 h at room temperature. To obtain stained specimens that reach saturation, slides were incubated in X-gal staining buffer for 24 h at room temperature. After staining, slides were post-fixed overnight at 4°C in 4% PFA. After sections were stained with X-gal, slides were counterstained with Nuclear Fast Red (Vector Laboratories) or EosinY (Sigma-Aldrich, HT110116), dehydrated through ethanol and Histoclear (National Diagnostics, HS-200), and mounted with Cytoseal-60 (Epredia, HT8310-4). Slides were imaged on a Zeiss Axio Imager.Z2 microscope. When compiling figures, global adjustments to brightness and contrast, color balance, and levels settings were made to provide uniformity across images.

### Identification of AP-1-binding sites and site-directed mutagenesis

GREAT (McLean et al., 2010) was initially used to generate a list of potential candidates for putative transcription factor binding sites in Enhancers 2 and 3, in which a single AP-1-binding site was identified. A second AP-1 site was identified using TFSEARCH (http://diyhpl.us/~bryan/irc/protocol-online/protocol-cache/TFSEARCH.html).

To generate mutant AP-1 sites, we identified two putative AP-1-binding sequences within Enhancer 3 with the sequence TGAGTCA [mm9, chr15:98610877-98611400 (reverse complement)] and designed non-complementary base transversions to create a GTCTGAC sequence. We noticed that the last four nucleotides of this altered sequence produced a new TGAC sequence (reminiscent of the AP-1 consensus sequence **TGAG/CTCA**), followed by a TCT, in one of the two predicted AP-1-binding regions. Although this new sequence was unlikely to function as another AP-1-binding site, we further modified GTCTGAC to GTCAGTC to ensure that we were not creating a new site with AP-1-binding affinity. As of this writing, the sequence GTCAGTC resembles no known consensus binding sites.

Primers carrying the desired sequence GTCAGTC were generated. Primers were: NPP1F 5′-GGGTAACATTCTGTCTTGGT**GTCAGTC**G-AAGACTCCTTGG-3′; NPP1R 5′-CCAAGGAGTCTTC**GACTGAC**AC-CAAGACAGAATGTTACCC-3′; NPP2F: 5′ GCTTAGCAACAGA**GTC-AGTC**CCCAAGACCC-3′; NPP2R: 5′ GGGTCTTGGG**GACTGAC**TCT-GTTGCTAAGC-3′.

The two sites were mutated sequentially using the Quick-Change Site Directed Mutagenesis Kit (Agilent Technologies, 200523) following the manufacturer's instructions. Wild-type Enhancer 3-LacZ was used as a template. Introduction of the first mutant site was verified by sequencing before introducing the second site, and the second mutant site was also confirmed by sequencing. Before injecting into mice, to verify that a functional promoter and *lacZ* gene were still present in the construct, wild-type Enhancer 3-LacZ and AP-1 mutant Enhancer 3-LacZ constructs were transfected into HEK-293 cells along with a pCMV-GFP as a transfection control. Cells expressing the wt and AP-1 mutant constructs both showed β-gal activity in the transfected cells. Constructs were then linearized using XhoI and NaeI before pronuclear injection of the transgene (Cyagen Biosciences).

### Generation of mice carrying a deletion of regulatory sequences for Enhancers 2 and 3

To generate mice harboring a deletion of the Enh2/3 region, candidate guide RNAs were identified using CRISPRscan (Moreno-Mateos et al., 2015) (https://www.crisprscan.org/). Several candidate single guide RNAs (sgRNAs) were screened for their ability to cleave a target template *in vitro* using a Guide-it™ sgRNA In Vitro Transcription and Screening Systems Kit (Clontech), following the manufacturer's instructions. A 2 kb target template was generated by PCR using FVB genomic DNA amplified with the following primers: 5′-GAATGGTTGTGAGCCACCTT-3′ and 5′-GAGCTCCTTCCCATTTAGGG-3′. sgRNAs were synthesized using the HiScribe™ Quick T7 High Yield RNA Synthesis Kit (NEB, E2050). sgRNAs with an *in vitro* cutting efficiency of over 50% were considered acceptable candidates. Two sgRNAs were selected: gG18NGG-53: 5′-TGGGCAAGCCTTTGAAGCCC-3′; and gG18NGG-63: 5′-AGGAAGAT-TATAGCGCCCAG. sgRNAs (10 ng/μl) and Cas9 mRNA (30 ng/μl) were introduced into FVB mouse one-cell embryos by pronuclear micro-injection and transferred into pseudo pregnant CD-1 females by the Stanford Transgenic Knockout and Tumor Model Center. Genotyping primers for

CRISPR-Cas9 mice carrying deletions of Enhancers 2 and 3 are: Crispr Enh23 #2F:5′-GCTGCAGTTCCATTCACACGTTG-3′; and Crispr Enh23 #2R:5′-TGATCTCCCTCCCTCAACTTCCT-3′.

### qRT-PCR

Tissue homogenization was performed in TRIzol reagent (Invitrogen) using a glass bead homogenizer (BeadBug, Benchmark Scientific, D1030, speed at 4000×, three cycles of 1 min each). Linear Acrylamide (Thermo Fisher Scientific, AM9520) was added and then RNA was purified using the RNeasy Mini Isolation Kit (QIAGEN). Genomic DNA contamination was digested using RNase-free DNase (QIAGEN, 75254). cDNA was synthesized using a High-Capacity cDNA Reverse Transcription Kit (Life Technologies) according to the manufacturer's protocol. cDNA was quantified on a NanoDrop 2000 Spectrophotometer (Thermo Scientific). qRT-PCR reactions which were performed with a TaqMan Gene Expression Master Mix (Applied Biosystems) using a StepOnePlus Real-Time PCR Instrument (Applied Biosystems). The relative expression of target genes was calculated using the ΔΔCT method and fold changes were calculated relative to *Srp14* as a normalization control as it was most stable between injured and uninjured muscles (Welc et al., 2020). Other genes tested were *Gapdh*, *Actb*, *Hagh* and *Rps2*; *Hagh* exhibited similar stability as *Srp14*, but, in our hands, the other reference genes were not stable between uninjured and injured conditions. Probes utilized for assays were: *Srp14* (Mm00726104_s1), *Pax7* (Mm01354484_m1), *Pdgfra* (Mm00440701_m1), *Cebpb* (Mm00843434_s1), *Wnt1* (Mm01300555_g1), *Wnt10b* (Mm00442104_m1), *Wnt5a* (Mm00437347_m1), *Cebpa* (Mm00514283_s1), *Pparg* (Mm00440940_m1), *Axin2* (Mm_00443610). For ease of visualization, the graphs display the $2^{\wedge}(-\Delta\Delta CT$ values), but statistical significance was calculated on the ΔΔCT values. Data were analyzed using Microsoft Excel and GraphPad (Prism) software.

### Quantification of myofiber size

To measure myofiber cross-sectional area, we utilized Alexa 647-conjugated Wheat Germ Agglutinin (WGA) (Molecular Probes, W32466) to stain myofibers because it allowed staining on paraffin sections (Aishwarya et al., 2022). Muscle sections (5 μm thick) were deparaffinized and then rehydrated into PBS-0.1% Tween-20 (PBST) and incubated in WGA overnight (Molecular Probes, Thermo Fisher Scientific, W32466; 1:500). The next day, slides were washed in PBST and mounted in ProlongGold with DAPI (Invitrogen, P36931). Slides were imaged on a Zeiss Axio Imager.Z2 with an Apotome and images were saved as .czi files and fed into Cellpose (Stringer et al., 2021) to segment the myofibers in WGA-stained sections. Cellpose has been used successfully to quantify CSAs of muscle tissue (Waisman et al., 2021). Sections were analyzed using the 'Labels to ROIs' plugin in Fiji to generate area measurements. GraphPad (Prism) was used to generate histograms and violin plots of the cross-sectional areas. Histograms are displayed as mean±s.e.m.

### Picrosirius Red staining and quantification of collagen deposition

To measure collagen deposition in tissue sections, 5 μm-thick paraffin sections were deparaffinized and rehydrated into PBS. Then they were stained using a Picrosirius Red Stain Kit (Connective Tissue Stain) (Abcam, ab150681) following the manufacturer's instructions. Slides were then imaged on a Zeiss Axio Imager.Z2. To quantify the staining, images were processed in NIH Image (Fiji). The total number of pixels was calculated per field. Then, using the 'Threshold' function the darkly stained collagen areas were selected. Images were changed to binary and the number of pixels marking the collagen-stained region was quantified. The number of collagen pixels was divided by total pixels in each field to give a measurement for Picrosirius Red staining. Data was entered into GraphPad (Prism) for analysis. For each mouse examined, three to five sections per mouse were generated and multiple fields were analyzed to calculate an average for each animal. Graphs were plotted as mean±s.d. Statistical analysis was performed using GraphPad (Prism).

### Pyrosequencing

To measure allele-specific expression of *Wnt1* and *Wnt10b* using pyrosequencing (Wittkopp, 2011), SNPs between CAST/EIJ mice and

FVB/NJ mice were identified using the MGI SNP query tool (https://www.informatics.jax.org/snp, GRCm38).

Primers were selected to amplify a region containing at least one SNP with products less than 300 bp. All primers were generated by Integrated DNA Technologies. Forward and reverse primers flanking the SNP for Wnt10b were: SNP_A-F, 5′-biotin-GAAAGGGCCTCCAAGAGTTAT-3′; SNP_A-R, 5′-TGTGGAGTCAATAAGACCCGTATA-3′. Sequencing primer for SNP_A was 5′-GAAAGGGTCTCTCCAA-3′. PCRs were run at 56°C for 45 cycles, giving a 266 bp product. Forward and reverse primers flanking the SNP for Wnt1 were: SNP_2-3-F, 5′-TTGCGC-TGTGACCTCTTTGG-3′; SNP_2-3-R, 5′-biotin-AGCTTTCCGTGCCC-TTTCAAC-3′. Sequencing primer for SNP_2-3 was 5′-ACCTGTAGCT-GAAGAGTT-3′. PCRs were run at 58°C for 45 cycles, giving a 154 bp product.

Note that these primers did not span intron–exon boundaries. Primer sets were initially tested on purified FVB/NJ and CAST/EIJ genomic DNA that was extracted using TRIzol (Invitrogen), precipitated, and quantified on a NanoDrop 2000 Spectrophotometer (Thermo Scientific). For pyrosequencing experiments, TA muscles belonging to FVB$^{wt}$, CAST/EIJ$^{wt}$, FVB$^{\Delta 1}$/CAST/EIJ$^{wt}$ and FVB$^{\Delta 2}$/CAST/EIJ$^{wt}$ were injected with BaCl$_2$, and tissues were collected at 3 days post-injury.

Tissue homogenization was performed in TRIzol reagent (Invitrogen) using a glass bead homogenizer (BeadBug, Benchmark Scientific, D1030, speed at 4000×, three cycles of 1 min each). Linear acrylamide (Thermo Fisher Scientific, AM9520) was added and then RNA was purified using the RNeasy Mini Isolation Kit (QIAGEN). Genomic DNA contamination was digested using RNase-free DNase (QIAGEN, 75254). cDNA was synthesized using a High-Capacity cDNA Reverse Transcription Kit (Life Technologies) according to the manufacturer's instructions. cDNA was quantified on a NanoDrop 2000 Spectrophotometer (Thermo Scientific). cDNA was used as input for PCR reactions. Products were run on an agarose gel to check for amplification of correctly sized products. Pyrosequencing runs were performed by the Stanford Protein and Nucleic Acid Facility on a Pyromark Q24, with results reported as percentages of each nucleotide that was observed at the SNP site.

Statistics were performed on percentage values. Allelic percentages were baseline-corrected using the observed minor-allele percentages in the 0% and 100% controls, which defined the lower and upper bounds. Sample values were then linearly re-scaled before performing statistical analysis. Corrected proportions were then arcsine square-root-transformed for ANOVA (Hsiao et al., 2012). Plots display the corrected FVB allele percentages for interpretability, but significance values are from the statistical analysis using ANOVA on arcsine-transformed values. Data were analyzed using Microsoft Excel and GraphPad (Prism) software.

## Perilipin staining

Wt and $\Delta 1/\Delta 2$ mice were injured with BaCl$_2$, and at 14 days post-injury TA muscles were isolated and fixed in NBF overnight at 4°C. Muscles were then washed in PBS and processed for paraffin embedding and sectioning. To sample a large region of the muscle, serial sections with ten sections per slide, consisting of sections spaced 50 μm apart for a total of ~2.5 mm, were generated on five slides. Sections were deparaffinized and processed into PBS by standard methods. Slides were placed in 200 ml Tris Antigen Unmasking Solution (Vector Laboratories, H-3301) in an Instant Pot Duo Plus Mini (3qt) and pressure cooked for 20 min. Pressure was released manually, and slides were left to cool for 30-45 min on the benchtop. Slides were removed and a hydrophobic barrier was drawn around each section. Slides were blocked for 1 h at room temperature in 10% normal donkey serum in PBST before incubation overnight in perilipin 1 antibody (Sigma-Aldrich, P-1998, RRID:477326; 1:200). The next day, slides were washed in PBST and then incubated in secondary antibody (donkey anti-rabbit Cy5, Jackson ImmunoResearch, 711-175-152, RRID:AB_2340607; 1:1000). After washing in PBST, slides were mounted in Prolong Gold Antifade Reagent with DAPI (Invitrogen, P36931) and imaged on a BZ-X800 microscope (Keyence) and montages were generated to make images of entire cross-sections through the muscle. Images were re-oriented in Adobe Photoshop and placed on dark backgrounds to generate the figures, and brightness levels were globally adjusted to make staining easier to visualize,

but no changes were made to the sections themselves, unless they were cropped by the software during image acquisition.

## Quantification of adipogenesis by perilipin staining

To quantify the extent of adipogenesis in wt versus $\Delta 1/\Delta 2$ muscle, we first examined how uniformly adipocytes were distributed throughout the muscle at 14 days post-injury. We reasoned that the focal nature of the BaCl$_2$ injury model might affect adipocyte distribution in the muscle post-regeneration. We cut 40-50 serial sections per muscle with each 5 μm section spaced 50 μm apart, to survey a total of 2250-2750 μm along the length of the muscle. We identified adipocytes using a perilipin antibody, evaluated section area using DAPI staining, and generated a percentage adipogenesis value for each section (see paragraph below). Both wild-type and $\Delta 1/\Delta 2$ muscle sections show that the presence of adipocytes can fluctuate over the length of the muscle, suggesting that surveying a larger number of sections distributed along the muscle would allow for a more accurate assessment of the overall extent of adipogenesis within the muscle tissue. Sections that displayed significant folds or tears were eliminated from the data set but, overall, 30-50 sections were captured and quantified per muscle.

For quantification of perilipin staining and generation of the percentage adipogenesis value, montaged images of entire sections acquired on the BZ-X800 microscope (Keyence) were imported into Fiji. If images contained extra debris, the EDL muscle, or portions of the epimysium, they were manually erased with the paintbrush tool to avoid quantifying non-TA muscle tissue. No portions of the TA muscle were eliminated. We were unable to develop a good automated sequence for image processing due to differences in DAPI and perilipin staining intensity between slides and specimens, and other features such as folds or wrinkles in some sections that made it difficult to uniformly apply the same parameters across all images. Therefore, processing was carried out manually but the following steps were applied. The 'Split Channels' function was used to generate separate images for DAPI and perilipin staining. To quantify the total section area in the DAPI image, the slider of the 'Threshold' function was used to fill the entire section area. The image was then made binary, and the number of pixels in the total section was quantified using the 'Histogram' function. For perilipin staining, the original image was opened, and the brightness was adjusted using the 'Auto' function to clearly show all perilipin-stained adipocytes. The perilipin-only image was then placed next to the original image, and the 'Threshold' function was used to highlight the perilipin adipocytes to make them match the original image. Care was taken not to erase or diminish the perilipin signals in the single-channel image or to make the perilipin contours of the adipocytes thicker or brighter than the original image. Speckled background signals in the perilipin channel that were not part of the adipocyte staining pattern were then eliminated using the 'Despeckle' function. Any folds or speckles that could not be eliminated that added non-adipocyte signals in the perilipin channel were erased by hand to avoid including them in the quantifications. Binary images were generated, and the number of adipocyte pixels were counted using the 'Histogram' function. Values for the number of pixels calculated in the perilipin channel was divided by the number of pixels calculated in the DAPI channel ×100, and this generated the percentage adipogenesis value. All data were analyzed using GraphPad (Prism) software.

## Quantification of adipogenesis by H&E staining

Quantification of adipogenesis by H&E staining was performed similarly to perilipin measurements. Automated segmentation was difficult due to the inability of software to distinguish between myofiber and adipocyte shapes in the tissue sections, so analysis was performed manually. H&E-stained sections were opened in Adobe Photoshop as RGB images, and contrast was enhanced to sharpen boundaries between the unstained adipocyte 'holes' and the surrounding stained myofibers. Adipocytes were selected and filled with black using the 'paint bucket' tool. Brightness was decreased to intensify the black adipocyte shapes compared to the background and then identified with the magic wand tool and pasted into a separate image that was imported into Fiji. Data were converted into binary image and the number of pixels was quantified using the histogram tool. Overall section area was quantified by importing the same RGB file into Fiji, splitting it into separate channels. Using the green channel file, which displayed the most

contrast, the 'Threshold' tool was used identify the contour of the section. Similar threshold values were applied across all files although small adjustments were made due to slight differences between images. Images were converted into binary and then the histogram tool was used to measure pixel values. Percentage adipogenesis was calculated as the number of adipocyte pixels/total number of pixels in the section ×100. All data were analyzed using GraphPad (Prism) software.

## Quantification of extent of injury

To quantify the extent of injury, three well-spaced sections were quantified for each muscle. In general, the values across all three sections were similar. If values diverged significantly, more sections were counted (up to five). Total section area was quantified based on DAPI staining. Perilipin-stained montaged images of whole muscle sections were imported into Adobe Photoshop and channels were split into single colors. The DAPI image was then imported into Fiji. The 'Threshold' function was used to highlight the entire section area, a binary image was generated and using the 'Histogram' function, and the total number of pixels comprising the section area was quantified. In Photoshop, the injured area was then marked in the same DAPI-stained TIFF file by using the pencil tool to draw outlines around all myofibers containing centrally located nuclei. Often, injured myofibers were found in large contiguous patches that could be outlined. These injured areas were then filled with the bucket tool to mask all portions of the section that were injured. This file was imported into Fiji, converted into a binary image and the pixels representing the injure regions were quantified using the 'Histogram' function. Values for the number of injured area pixels was divided by the number of total pixels calculated for the section area, and this generated the percentage injury value. Additionally, an 'Adipogenesis Index' was calculated to normalize the percentage adipogenesis to the different percentage injury values, which was reported as (% adipogenesis/ % injury×100). All data were analyzed using GraphPad (Prism) software.

## Assessment of dorsal punch biopsy wound closure over time

To follow skin wound healing over time, mice were anesthetized with isoflurane and placed briefly in a cardboard chamber with two holes cut out on the top, to allow for illumination with a fiber-optic light source and placement of an iPhone SE for daily photography of wound closure. Animals were placed next to a ruler, dorsal side up. Images were collected every 24 h. Quantification of wound size over time was measured by importing images into Fiji. Images were split into different channels using the 'Color' function, and then the 'Threshold' function was applied to the green channel to demarcate the wound area. The 'Make Binary' function was applied to the image and the number of pixels was counted to quantify the size of the wound. Closure of the wound was compared to the initial size to generate a value for the percentage of starting area over time. Animals were followed for 9 days. All data were analyzed using GraphPad (Prism) software.

## Acknowledgements
We thank the Stanford Core facilities, particularly the Stanford Protein and Nucleic Acid Facility for Pyrosequencing, and the Stanford Transgenic Knockout and Tumor Model Center. We thank Dr Hong Zeng (Stanford Transgenic Knockout and Tumor Model Center) for discussions during generation of transgenic and CRISPR-Cas9 animals. We thank Pauline Chu, Department of Comparative Medicine, Stanford University for tissue processing and embedding during early stages of this work. Additional contributors to early phases of this project include Dr Barbara Brott, Johanna Kirby, Kia Fathi and Dr Si Hui Tan. We thank Drs Ross Metzger, Hernan Espinoza, Catherine Gunther and Ian Heller for many stimulating scientific discussions and Drs Mark Krasnow and David Kingsley for support and encouragement over many years. We also thank Drs Aaron Wenger and Jim Notwell from Dr Gil Bejerano's laboratory for exploring possible transcription factor-binding sites within our enhancers. Thank you to members of Dr Thomas Rando's lab: Mike Wosczyna for showing us the BaCl2 muscle injury procedure and for Alisa Mueller for sharing her observation that BaCl2 induces Wnt10b. We also thank Dr Joseph Wu for teaching us the myocardial infarction technique. Thank you to Dr Zhibo Zhang for critical comments on this manuscript. This work is dedicated to the memory of Johanna Kirby and also YL and DHML.

## Competing interests
R.N. is a board member of Bio-Techne and a member of the Scientific Advisory Board of Surrozen Inc. The authors have no other competing interests.

## Author contributions
Conceptualization: C.Y.L., X.L.; Formal analysis: C.Y.L.; Funding acquisition: R.N.; Investigation: C.Y.L., X.L., M.P.F., M.M., B.S.; Methodology: C.Y.L., X.L.; Project administration: C.Y.L.; Resources: C.Y.L., X.L., M.P.F.; Supervision: C.Y.L.; Validation: C.Y.L.; Visualization: C.Y.L.; Writing – original draft: C.Y.L.; Writing – review & editing: C.Y.L., R.N.

## Funding
The Howard Hughes Medical Institute provided funding to R.N. and support for C.Y.L. and M.P.F. The Virginia and D.K. Ludwig Fund for Cancer Research and the Steinhardt Reed Foundation also provided funds to R.N. X.L. received support from National Science Scholarships from the Agency for Science, Technology and Research (A*STAR), Singapore. Open Access funding provided by Stanford University. Deposited in PMC for immediate release.

## Data and resource availability
Enhancer 2-LacZ, Enhancer 3-LacZ, Enhancer 3-eGFP, Enhancer 2+3-LacZ and Δ1/Δ2 animals will be made available at The Jackson Laboratory. All other relevant data and details of resources can be found within the article and its supplementary information.

## Peer review history
The peer review history is available online at https://journals.biologists.com/dev/lookup/doi/10.1242/dev.204933.reviewer-comments.pdf

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
