## [Peer Review File · Development (Cambridge, England)]

Deletion of an enhancer that controls Wnt gene expression following tissue injury produces increased adipogenesis in regenerated muscle

Catriona Y. Logan, Xinhong Lim, Matt P. Fish, Makiko Mizutani, Brooke Swain and Roel Nusse

DOI: 10.1242/dev.204933

Editor: Meritxell Huch

Review timeline

Original submission:	12 May 2025
Editorial decision:	21 July 2025
Rebuttal received:	8 August 2025
Editorial decision:	11 August 2025
First revision received:	15 September 2025
Editorial decision:	8 October 2025
Second revision received:	22 October 2025
Accepted:	23 October 2025

Original submission

First decision letter

MS ID#: dev.204933

MS TITLE: Deletion of an enhancer that controls Wnt gene expression following tissue injury produces increased adipogenesis in regenerated muscle

AUTHORS: Catriona Logan, Xinhong Lim, Matt P. Fish, Makiko Mizutani, Brooke Swain and Roel Nusse

First of all, my sincere apologies for the delay on taking a decision on your manuscript. I have now received all the referees reports on the above manuscript, and have reached a decision. The referees' comments are appended below, or you can access them online: please go to: .

As you will see from their reports, the referees recognise the potential of your work, but they also raise significant concerns about it.

After carefully assessing the comments from the Reviewers and consulting with the Editorial team, we felt that some of the experiments required for the revision go beyond what we would usually expect and might take longer. Given the nature of these concerns, I am afraid I have little choice other than to reject the paper at this stage.

However, having evaluated the paper, I do recognise the potential importance of this work. I would therefore be prepared to consider as a new submission an extension of this study that contains new experiments, data and discussions and that address fully the major concerns of the referees. The work required goes beyond a standard revision of the paper.

Reviewer 1

Advance summary and potential significance to field

The manuscript identifies a conserved regulatory region between Wnt1 and Wnt10b that is activated by tissue injury and required for proper muscle regeneration. This work represents the first identification of a vertebrate injury-responsive enhancer driving Wnt ligands critical for regeneration, with multi-tissue validation across lung, muscle, liver, and skin that strengthens the enhancer's conserved role in damage sensing. The methodology is rigorous, combining CRISPR deletion, transgenic reporters, and allele-specific expression analysis through pyrosequencing to establish enhancer necessity and sufficiency, while AP-1 site mutagenesis cleanly links enhancer activity to injury signaling. The physiological relevance is demonstrated through germline enhancer deletion that produces a clear regenerative defect characterized by adipogenesis without developmental confounders, highlighting the enhancer's adult-specific role in tissue repair. While the study is well-executed and addresses an important gap in understanding injury-responsive enhancers, several major and minor revisions are needed to strengthen the conclusions and clarity.

Comments for the author

While the study is well-executed and addresses an important gap in understanding injury-responsive enhancers, several major and minor revisions are needed to strengthen the conclusions and clarity.

Major Concerns

1. Enhancer Specificity and Redundancy:

The partial reduction of Wnt1/Wnt10b expression in $\Delta 1/\Delta 2$ mice (Fig. 4F-G) and residual adipogenesis (Fig. 5) suggest compensatory mechanisms. The authors should test whether other nearby conserved regions (e.g., Enhancer 1) contribute to injury-responsive Wnt expression. Alternatively, the authors could, if possible, perform chromatin conformation capture (3C or Hi-C) to confirm physical interaction between the enhancer and Wnt promoters in injured muscle.

2. Mechanistic Link Between Wnt Loss and Adipogenesis:

The adipogenic phenotype is attributed to reduced Wnt signaling, but direct evidence is lacking. The authors should either measure β -catenin activation (e.g., Axin2 expression) in $\Delta 1/\Delta 2$ muscles post-injury, or rescue the phenotype by administering Wnt1/Wnt10b agonists or inhibiting adipogenic pathways (e.g., PPAR γ).

3. Tissue Specificity of the Enhancer:

The enhancer is active in lung, liver, and skin (Fig. 3B), but regeneration defects are only analyzed in muscle. The authors may wish to assess regeneration in other tissues (e.g., lung after naphthalene injury) to determine if the enhancer's role is muscle-specific.

4. Developmental Impact of Enhancer Deletion:

Sub-Mendelian ratios in $\Delta 1/\Delta 2$ mice (Table 2) suggest possible embryonic roles. Evaluate embryonic Wnt1/Wnt10b expression and developmental phenotypes (e.g., CNS defects) in $\Delta 1/\Delta 2$ embryos.

Minor Concerns

1. Quantification and Statistics:

- Clarify sample sizes in figure legends (e.g., Fig. 1C-D: n=8 uninjured vs. n=5 injured).
- Include individual data points in violin plots (Fig. 4F-H) and specify statistical tests used for each analysis.

2. Figure Clarity:

- Improve resolution of LacZ staining in whole-mount images (Fig. 2B, 3B-D).
- Label cell types in Fig. 1E-G (e.g., FAPs, macrophages) to aid interpretation.

3. AP-1 Binding Site Conservation:

The second AP-1 site in Enhancer 3 is noted as divergent in newer genome alignments. Perform phylogenetic analysis to assess conservation across 240 mammalian species (as referenced) and discuss implications.

4. Discussion:

- a. Temper evolutionary claims about conserved Wnt regulatory mechanisms until functional studies in non-mammalian models (e.g., zebrafish, killifish) are performed.
- b. Address why Wnt5a (Fig. 4H) is unaffected by enhancer deletion, given its known role in muscle regeneration.

This study provides very valuable insights into injury-responsive enhancers and their role in regeneration. Addressing these points will solidify the mechanism linking enhancer function to Wnt signaling and tissue repair. The work is suitable for Development after revisions.

Reviewer 2

Advance summary and potential significance to field

In this manuscript, Logan et al. identify a regulatory element that drives injury-responsive expression of Wnt1 and Wnt10b in several injury contexts. These two are neighboring genes but transcribed in opposite directions, separated by a 12kb interval. A comprehensive analysis of all Wnt gene expressions in four different tissue injury models suggests that Wnt1 and Wnt10b exhibit similar injury-responsive expression patterns, implying that their injury-responsive transcription may be regulated by shared regulatory elements. Sequence conservation analysis followed by transgenic mouse assays defines two short enhancer fragments (Enh2 and Enh3) capable of driving injury-responsive expression. Mouse transgenic assay further revealed that two AP-1 binding motifs within Enh3 function as crucial cis-regulatory elements for injury-inducible expression. Deleting these enhancer sequences likely reduced injury-responsive Wnt1 and Wnt10b expression in skeletal muscle tissues. Their regeneration assays suggest that enhancer deletion may impair regeneration, as evidenced by elevated adipogenic features in regenerated muscle. While the data provide some support for their central claim that deletion of injury-responsive enhancers causes regenerative defects, the observed changes in gene expression and adipogenic phenotypes require more thorough analysis. Additionally, improvements to data presentation are needed for publication. I provide major and minor comments below.

Comments for the author

Majors

1. Regeneration assays

According to the methods and main text, the authors calculate an injury index to account for variability in injury severity induced by BaCl₂. Figure 5E shows that injury levels among mice range from ~20% to 80%, a probably, unacceptably large variation that complicates reliable analysis. Furthermore, the reported adipogenesis level appears extremely limited (~0 - 2%). There is also no clear evidence that adipogenic signals are present nearby or among regenerated muscle fibers. The authors should evaluate regenerative defects using alternative approaches. For example, they could quantify lipid accumulation in the regenerated muscle area or assess how many regenerated muscle fibers (central-nucleated cells) are positive for lipid staining (e.g., Oil Red O). Similar phenotypes have been reported in Wnt10b mutants (Vertino et al., 2005).

2. Regeneration phenotypes

The regenerative defects in the enhancer deletion lines appear minor, making it difficult to conclude that a meaningful regenerative impairment exists. The authors should consider additional experimental settings to more clearly demonstrate regeneration defects:

1) Sensitized genetic background

Enhancer redundancy may mask phenotypic effects in homozygous enhancer deletion lines. Combining the enhancer deletion allele with the Wnt1 and/or Wnt10b mutant could reveal more pronounced defects. Such combinations could clarify regenerative defects of enhancer deletion.

2) Aging

Since Wnt10b mutants display adipogenic defects in aged animals, evaluating regenerative and adipogenic outcomes in aged mice could better demonstrate the functional importance of these enhancers in tissue regeneration.

3. Enh3 activity quantification and control images

The use of non-transgenic animal images as controls is not appropriate for main figures. The proper control should be uninjured tissue from the enhancer reporter lines, but uninjured images are missing from several key figures. Please provide uninjured lacZ images for the enhancer reporter lines in Figures 2C, 3B, S5, and S7. Figure 3B should include images of vehicle day 1 (d1) liver, uninjured skin, and uninjured heart. Non-transgenic images can be moved to supplementary figures.

In addition, please quantify the reporter gene expression levels for the enhancer lines to better demonstrate injury-responsive activity. For example, Figure 2C should include a quantification graph comparing uninjured tissue and each post-injury time point (d1, d3, and d5) to clearly show the injury-responsive induction. Also, quantification data of uninjured and each post-injury time point for AP-1 mutated mice should be required as non-specific or leaky expression is visible for some AP-1 mutated mice. Note that AP-1 line 21 1d and 5d in Fig. 3D exhibits LacZ signal.

4. Enhancer activity

Lines 237 and 238 propose distinct activities of Enh2 and Enh3. Since the endogenous regulatory activity of Wnt1/Wnt10b likely reflects the combined contributions of these enhancers, it would be informative to evaluate the activity of the 6 kb and the Enh2+3 reporter lines. Do these lines exhibit distinct reporter gene expression patterns compared to the individual enhancers? A more comprehensive and comparative analysis with 6 kb and Enh2+3 lines together with Enh2 and Enh3 will provide mechanistic insights into how multiple enhancers coordinate to regulate a single gene.

5. Comprehensive analysis Wnt expression in distinct injury models

One important data in this manuscript is the comprehensive transcriptional analysis of 19 Wnt genes across 4 different injury models. This analysis identified Wnt1 and Wnt10b as being consistently induced in multiple tissue injuries with similar expression patterns, as stated in Lines 112-114. However, the current presentation does not adequately convey the broader Wnt expression results and makes it difficult for readers to interpret the data. To address this, the authors should provide a summary table showing the expression patterns of all 19 Wnt genes across uninjured and post-injury time points, including mouse numbers for each condition. Such a table would be essential to properly support and present this important dataset.

6. Table 1

The current table 1 requires to be revised to add more detailed information. Please indicate line number and corresponding expression results for each time point. For example, Enh line 2 number x, uninjured: no expression or -. Muscle BaCl2 injury day 1 - +, damaged myofiber; day3: ++ damaged and regenerating myofibers; day5: +++++, regenerating myofiber.

7. Wnt1 and Wnt10b expression changes in enhancer deletion lines

The authors employ Pyrosequencing approach to measure Wnt1 and Wnt10b expression level change by enhancer deletion. However, there is no statistical analysis result. Please perform statistical analysis with sufficient number of samples and provide data.

In enhancer deletion mutants, Wnt1/10b expressing cell numbers decline more than half at day 5, so their transcript levels may decline significantly at day 5. Moreover, SNPs in Wnt1/10b regulatory elements across distinct wild-type strains may affect injury-responsive expression. In addition to Pyrosequencing approach, the authors require to perform RT-qPCR or RNA-seq analysis (preferable RNA-seq as this is an unbiased method) to measure Wnt1/10b expression change in wild-type and enhancer deletion lines in day 5, compared to uninjured.

8. Active enhancer signature

There are multiple published ATAC-seq profiles, active enhancer profiles or scATAC-seq datasets for uninjured and injured skeletal muscles. Please examine whether Enh2 and Enh3 become accessible or are marked by active enhancer markers upon injury.

Minor

1. Abstract first sentence is overstated. Please revise it.

"The capacity to detect and respond to injury is critical for the survival of all organisms" -> Please weaken by removing "all"

2. Wnt1 and Wnt10b gene structure

Fig 2A and S4A are low resolution, making difficult to evaluate the data. Which one is Wnt10b and what is the direction? Please annotate "Wnt10b" above left gene and "Wnt1" above right one. Please clearly indicate the direction of both genes. Please include 5.5kb 3' regulatory region containing the embryonic regulatory element in this genome browser. The current format should be improved for publication.

3. AP-1 sequence conservation

Which species are used for sequence conservation study in Fig 2A? Please provide more detailed information of conservation data. Line 276-278 suggests that two AP-1 site sequences across multiple species were compared. If so, please provide the data. Providing actual data will improve this manuscript.

4. Line 291-294 and Table2.

This interpretation is overstated as $del1/+$ and $del2/+$ ratios are not increased. If there is any detrimental effect of homozygotes, all of other genotypes, wt, $del1/+$ and $del2/+$, should increase rather than only wt. Please revise this statement.

5. Correct typo in line 336. It says (Figure D, E, ...) and should be corrected to reference Fig 5D, E.

6. Correct typo in lines 912 and 914, which says $*P<0.05$, $**P<0.05$.

7. In the discussion in lines 455-458, the authors claim that this is the first example of an enhancer region driving developmental signaling molecules that is re-used in adulthood to facilitate regeneration. However, no data are provided to show that Enh2 or Enh3 can drive expression during development. Please perform reporter assays in developing tissues to demonstrate their developmental activity. If such data are not available, this statement should be revised accordingly to avoid overinterpretation.

8. Wnt1 and Wnt10b co-regulation by Enh2 and 3

If Wnt1 and Wnt10b are regulated by the same enhancer elements, why do most nuclei express either one gene or the other, rather than both (as shown in Fig. 1G and discussed in Lines 149-154)? The fact that only about 10% of cells co-express both genes may not provide strong support for the claim that "common regulatory element" drives their expression. The authors should address this discrepancy in the discussion. Note that Wnt10b level seems to be more influenced by enhancer deletion, suggesting the presence of other regulatory element controlling Wnt1. Additionally, while TAD (topologically associating domain) data are mentioned in the discussion, no such data are provided. Including a TAD map as a supplementary figure would strengthen this claim - please provide this data if available.

Rebuttal

Summary of plan to address reviewer comments, and items that we submit here:

Reviewer#1

Major comment 1: Discussion of why examining Enhancer 1 may be challenging and a complete survey of other regulatory regions is beyond scope of this paper

Major comment 2: **NEW FIGURE:** QPCR of Axin2 is provided (d7 timepoint), *we aim to provide data at d5 timepoint for the reviewers.*

Major comment 3: **NEW FIGURE:** Figures from skin injury are provided

Major comment 4: **NEW FIGURE:** Figures from embryonic Enhancer 3 LacZ expression is provided. *Embryonic Enhancer 2+3 staining data is being generated now.*

Minor concerns 1: We will clarify sample sizes and have re-made violin plots with larger dots to make them more visible (Fig. 4F-H).

Minor concerns 2: We can revise the figures. Revised figure 2B and 3C are shown here.

Minor concerns 3: **NEW FIGURE:** We provide figures and discussion on the conservation of the two AP-1 binding sites.

Minor concerns 4--Discussion: We temper our evolutionary claim and provide discussion about why Wnt5a is not affected in our experiments.

Reviewer#2

Major comment 1: Adipogenesis vs. injury linear regression plot is re-plotted to focus on more highly injured specimens, **NEW FIGURE:** QPCR data for 2 adipogenesis markers (CEBPa and PPARg), **NEW FIGURE:** Quantification of adipogenesis on H&E slides as a different measure of fatty infiltration, discussion of why measuring adipogenesis with Oil Red O was not performed. *We aim to provide a more complete H&E adipogenesis analysis for the reviewers.*

Major comment 2: We discuss experiments that were attempted on aged animals and with trying to generate sensitized backgrounds.

Major comment 3: We can address the reviewer's concerns about including uninjured tissue panels. We provide a revised figure 2C. We also provide a **NEW FIGURE** with uninjured tissues for heart, liver and skin for Fig. 3B and a **NEW FIGURE** showing injured non-transgenic animals that could be supplied as a supplementary figure. We also provide a **NEW TABLE** that quantifies AP-1 mutant LacZ expression levels compared to Enhancer 3.

Major comment 4: **NEW FIGURE:** We provide a figure of Enhancer 2+3 expression at different timepoints in the muscle.

Major comment 5: We are assembling the full in situ data results that we generated.

Major comment 6: We do not supply a more detailed Table 1, as requested by the reviewer, as the consistency of observing similar expression patterns across all samples was more important than expression level in the initial enhancer screen for injury-responsive elements. However, we do provide a **NEW TABLE** focusing on the muscle and on reporter expression levels at different timepoints post-injury.

Major comment 7: We will examine the appropriate statistical test for pyrosequencing data.

Major comment 8: We provide examples of publicly available data (CHIP-seq/DNase 1 hypersensitivity and ATAC-seq) that supports the idea that Enhancers 2 and 3 can function as active regulatory elements.

Minor comment 1: We will change the wording of the abstract to address reviewer's concern.

Minor comment 2: We will improve resolution and figure quality of Figure 2A and S4A.

Minor comment 3: We provide figures addressing questions around sequence conservation (see response to Reviewer #1, Minor comment 3)

Minor comment 4: We will improve the explanation of how we understand reduced Mendelian frequencies of del/del animals.

Minor comment 5: Typos to be corrected

Minor comment 6: Typos will be corrected

Minor comment 7: We will re-write our original statement, as it was poorly worded, and did not convey the meaning we were trying to impart to the reader.

Minor comment 8: We discuss why we think Wnt1 and Wnt10b expression does not overlap very well, and the possibility that other enhancers may also participate in regulating Wnt1 and Wnt10b expression. **NEW FIGURE:** Data from the UCSC genome browser that supports our claim in the discussion that Wnt1 and Wnt10b may reside within a TAD

In-depth description of data and rebuttal:

Reviewer #1:

Major Concerns

1. **Enhancer Specificity and Redundancy:**
The partial reduction of Wnt1/Wnt10b expression in $\Delta 1/\Delta 2$ mice (Fig. 4F-G) and residual adipogenesis (Fig. 5) suggest compensatory mechanisms. The authors should test whether other

nearby conserved regions (e.g., Enhancer 1) contribute to injury-responsive Wnt expression. Alternatively, the authors could, if possible, perform chromatin conformation capture (3C or Hi-C) to confirm physical interaction between the enhancer and Wnt promoters in injured muscle.

We did not mean to imply that we identified the *only* injury-responsive Enhancer that could drive Wnt1 and Wnt10b post-injury. This study was not intended to be a comprehensive characterization of injury responsive enhancers that regulate Wnt1 and Wnt10b, and it is likely that there are indeed, other regulatory elements that control Wnt1 and Wnt10b expression during regeneration.

As the reviewer noticed, Enhancer 1 could indeed be an interesting redundant candidate enhancer, as it displayed reporter expression in one out of two lines. We cannot distinguish if reporter expression from the single positive Enhancer 1 line represents a real injury response or simply the result of position effects on the construct. Enhancer 1 is less than 1 kb from the transcriptional start site Wnt10b, making deletion experiments that would help to show requirement of the enhancer difficult to interpret due to proximity to the promoter. Definitely proving a requirement for Enhancer 1 or any other candidate Enhancer would necessitate producing multiple new independently generated lines of adult reporter mice (with each independent adult founder mouse bred at least through the G2 generation to adulthood verify activity), as well as deletions to test for a requirement in driving Wnt1 and Wnt10b expression, a multi-year endeavor that we cannot do at this time and we feel is beyond the scope of the focus of this work.

2. Mechanistic Link Between Wnt Loss and Adipogenesis:

The adipogenic phenotype is attributed to reduced Wnt signaling, but direct evidence is lacking. The authors should either measure β -catenin activation (e.g., Axin2 expression) in $\Delta 1/\Delta 2$ muscles post-injury, or rescue the phenotype by administering Wnt1/Wnt10b agonists or inhibiting adipogenic pathways (e.g., PPAR γ).

We did try to measure Axin2 levels but were not able to detect much difference between wild-type and del/del animals. There are many Wnts present in muscles post-injury and during regeneration, and it is highly likely that any effect on Axin2 expression due to loss of Wnt1 and Wnt10b may be very difficult to detect in a whole-muscle assay.

We did compare markers of fat formation in wild-type vs del/del animals and we do see elevation of CEBP α and PPAR γ by QPCR, although the differences were not statistically significant. Detecting the subtle changes in adipogenesis generated by enhancer deletion may be difficult to see in whole muscle assays but there is an upward trend in the gene expression we observe.

3. Tissue Specificity of the Enhancer:

The enhancer is active in lung, liver, and skin (Fig. 3B), but regeneration defects are only analyzed in muscle. The authors may wish to assess regeneration in other tissues (e.g., lung after naphthalene injury) to determine if the enhancer's role is muscle-specific.

Our current data indicates that the role of Enhancer 2 and 3 is muscle specific.

In the lung, injured airways of mice lacking Enhancer 2+3 were able to regenerate, and airway cells were restored. We were unable to detect an obvious defect. The lung displays very strong Wnt7b expression (Figure S1C) which might mask and compensate the loss of Wnt1 and Wnt10b, and we did not follow up with the lung further. As this is a negative result, however, it is possible that a subtle defect may exist that we missed.

The skin displayed Enhancer 2 (Enh2) and Enhancer 3 (Enh3) reporter expression following punch biopsy skin injury (2 days-post injury), but unlike the muscle and lung, there was undetectable induction of Wnt1 or Wnt10b transcripts in the areas surrounding injury (see in situ data shown here, injured area is marked with a dotted red line). Moreover, Enhancer 2+3 LacZ reporter mice failed to stain with X-gal. This result suggests that Enhancers 2 and 3 likely can individually respond to transcription factors induced by injury. However, when put together, there is no reporter expression. It is tantalizing to hypothesize that the inability of Enhancer 2+3 to drive reporter expression could be one reason why Wnt1 and Wnt10b fails to be up-regulated at the wound site following punch biopsy wounds, although in vivo, there are likely other enhancers that regulate their expression as well.

We also examined the skin for phenotypes. In punch biopsy experiments, we did not observe any detectable wound closure defects when wound size was measured over time (1-9 days), although it is possible that there is a mild impairment that we did not detect. Top panel to the left shows a dorsal punch biopsy wound and wound closure in a wild-type mouse over time. The bottom panel shows the results of wound closure comparing wt vs. del/del animals (n=4 wt, n=4 del, p=n.s., d1, d5, d7 unpaired t-test; d3 = Welch's test; d9 Mann-Whitney test).

These data from examining other tissues suggest that activation of individual injury-sensing elements in a given locus may not always necessarily translate into target gene expression, and this highlights the role of tissue context and enhancer interactions in determining if a gene is ultimately expressed.

4. Developmental Impact of Enhancer Deletion:

Sub-Mendelian ratios in $\Delta 1/\Delta 2$ mice (Table 2) suggest possible embryonic roles. Evaluate embryonic *Wnt1/Wnt10b* expression and developmental phenotypes (e.g., CNS defects) in $\Delta 1/\Delta 2$ embryos.

As the reviewers point out, reduced Mendelian ratios of del/del mice suggest that Enhancers 2 and 3 in may have embryonic functions. Because only about a quarter of del/del embryos are lost sometime during embryogenesis, the number of litters required to identify a defect at the right timepoint and in the right tissue was prohibitive and we did not look for a phenotype. LacZ reporter staining on Enhancer 3 embryos at different stages, shows that the reporter is active during embryogenesis, but sites of reporter expression do not match known sites of *Wnt1* and *Wnt10b* expression. *Wnt1* is well-known to be expressed in the developing CNS and we see a dorsal axial stripe of LacZ staining at e10.5 and e11.5, but the pattern does not mark the developing CNS or neural crest cells as has been reported for *Wnt1* (Echelard et al., 1994 <https://doi.org/10.1242/dev.120.8.2213>; Danielian and McMahon, 1997 <https://doi.org/10.1006/dbio.1997.8762>). *Wnt10b* is also expressed in a similar pattern to *Wnt1* in the CNS (Veltmaat et al., 2004 doi.org/10.1002/dvdy.10441). Ye et al., (2013) (doi:10.7150/ijms.6170) report *Wnt10b* expression at the base of the hair shaft at p8, but Enhancer 3 is expressed more distally. There are other sites such as lacZ stripes around the wrists and feet but we do not know what these cells are. These data indicate that enhancers may bind transcription factors present in embryos but in contrast to expression in the muscle post-injury, sites of reporter activity do not appear to recapitulate *Wnt1* or *Wnt10b* expression.

We interpret these data to suggest that these enhancers may detect transcription factors that can bind them and drive reporter expression during embryogenesis, but do not drive Wnt1 or Wnt10b expression. Given these results, we think identifying tissues and timepoints that could explain lethality of the del/del mice due to loss of Wnt1 and Wnt10b could be challenging and we have not pursued this further.

Enhancer 3-LacZ:

Minor Concerns

1. Quantification and Statistics:

a. Clarify sample sizes in figure legends (e.g., Fig. 1C-D: n=8 uninjured vs. n=5 injured). We will check all of sample sizes and fix them.

b. Include individual data points in violin plots (Fig. 4F-H) and specify statistical tests used for each analysis.

We were unsure what the reviewer was asking for, as individual data points were included for the d3 timepoints for Wnt1 and Wnt10b. In case the dots were too small to see, we provide plots with larger dots. We also include a violin plot for Wnt5a here as well.

Statistical tests will state:

(F-H) Whiskers in Box plots represent the Min to Max. On violin plots, the red line represents

the median. (* $p < 0.05$: Wnt1 d5, Mann-Whitney test; Wnt10b d5, Unpaired t-test; ns = not significant: Wnt1 d3, Mann-Whitney test; Wnt10b d3, unpaired t-test; Wnt5a d3, Welch's t-test; Wnt5a d5, unpaired t-test) (Scale bar = 20 μm)

2. Figure Clarity:

a. Improve resolution of LacZ staining in whole-mount images (Fig. 2B, 3B-D).

We provide an improved Figure 2B with sharper images. Because the cells are deeper inside the muscle tissue, the staining will always be somewhat indistinct, but resolution should be improved.

We provide a new Figure 3C here. The other images (3B, D) will be similarly adjusted.

b. Label cell types in Fig. 1E-G (e.g., FAPs, macrophages) to aid interpretation.

We provide a revised panel.

In response to the reviewer's comments regarding Figure 1G, the two in situ signals shown are Wnt1 and Wnt10b which are labeled in red and blue font to correspond to the in situ signal color. Since they do not refer to cell types, there is no additional cell type information we can add to those panels.

3. AP-1 Binding Site Conservation:

The second AP-1 site in Enhancer 3 is noted as divergent in newer genome alignments. Perform phylogenetic analysis to assess conservation across 240 mammalian species (as referenced) and discuss implications.

At the time of our initial analysis of Enhancer 3, two putative AP-1 sequences appeared to be present in the mouse sequenc for Enhancer 3, so we opted to mutate both sequences to be sure that we eliminated AP-1 binding activity from the enhancer in our site-directed mutagenesis experiments. In all alignments that are available (mm9, mm10 and mm39) the mouse reference sequence (and our FVB animals from sequencing) contains consensus AP-1 binding sequences at both sites. When other species are included in the alignment however, we see that the second site is not a conserved AP-1 binding site. Phylogenetic analysis across 240 mammalian species indeed confirms that the second AP-1 sequence is likely not conserved. We show a multi-species alignment below with a focus on rodents to demonstrate that the second consensus AP-1 site is an exception rather than the rule and is only seen in the house mouse and Western wild mouse, but not other species.

AP-1 site #1 (the highly conserved site, TGAG/CTCA):

AP-1 site #2 (less conserved site, TGAG/CTCA) Note that mouse still carries this sequence but it is divergent in other species:

AP-1 site #2

Because the mouse sequence possessed two putative AP-1 sites and we wanted to ensure removal of AP-1 binding activity, deleting this site does not necessarily change the interpretation of our observations in that one or both AP-1 sites are likely required for the injury-responsiveness of Enhancer 3. It is possible that the second “non-conserved” site is not utilized.

4. Discussion:

a. Temper evolutionary claims about conserved Wnt regulatory mechanisms until functional studies in non-mammalian models (e.g., zebrafish, killifish) are performed.

We could add a statement (shown in bold) to stress the importance of further studies: “Although speculation at this point, given Wnt induction in virtually all injury contexts that have been described to date, there may indeed be deeply conserved regulatory mechanisms that drive Wnts following injury that spans across invertebrate to vertebrate species. **A definitive answer to this possibility, however, will require further functional studies in other non-mammalian species.**”

b. Address why Wnt5a (Fig. 4H) is unaffected by enhancer deletion, given its known role in muscle regeneration.

Although Wnt5a is thought to function in muscle regeneration and has been postulated to regulate adipogenesis, it is unaffected by enhancer deletion because it is likely regulated by its own injury-responsive enhancers and its expression is not dependent or downstream of Wnt1 and Wnt10b. This point is something we can add to the discussion.

Reviewer #2:

Majors

1. Regeneration assays

According to the methods and main text, the authors calculate an injury index to account for variability in injury severity induced by BaCl₂. Figure 5E shows that injury levels among mice range from ~20% to 80%, a probably, unacceptably large variation that complicates reliable analysis.

For Figure 5E, we thought that showing the full span of possible injury ranges would be the most honest way to display our results. We would normally consider a 20% injury to be due to poor injection technique and would not include such mildly injured specimens in any other type of analysis, preferring to examine sets of more similarly severely injured samples to simplify the analysis. For these experiments, however, because we saw fairly subtle changes in fatty infiltration (perhaps not surprising since we were trying to analyze loss of a small regulatory region), we wanted to keep all of the samples that we generated, in order to assess adipogenesis together with the level of perturbation, and to avoid reporting differences in fatty infiltration simply because the cohorts were injured to different degrees.

If we analyze the data only including the more highly injured samples, we see a similar upward shift in the adipogenesis exhibited by del/del animals to when the full dataset is plotted. Regression lines still support the idea that the wild-type mice overall generate lower levels of adipogenesis in the muscles compared to the del/del animals. We obtain a p-value of 0.0863 when we ask if the elevations of the lines are significantly different (ie. not quite significant), which is not surprising given the spread of the del/del data points. But we posit that these data reflect a defect in which control of adipogenesis becomes more dysregulated when the enhancer is deleted.

As show already in response to Reviewer #1 comment #1, an alternative way to demonstrate elevated adipogenesis in Enhancer del/del animals compared to wild type, it so measure expression of genes involved in driving adipogenesis, such as CEBPa and PPARg. Using QPCR, we compare levels between wild-type and del/del mice at 7 days post injury and show that there is elevation of these genes in del/del animals compared to wild-type. Because the difference in adipogenesis is subtle between wt and del animals, we were unable to obtain clear statistical significance, but one can see an increase for both markers in del/del tissue compared to wild-type.

Furthermore, the reported adipogenesis level appears extremely limited (~0 - 2%). There is also no clear evidence that adipogenic signals are present nearby or among regenerated muscle fibers. The authors should evaluate regenerative defects using alternative approaches. For example, they could quantify lipid accumulation in the regenerated muscle area or assess how many regenerated muscle fibers (central-nucleated cells) are positive for lipid staining (e.g., Oil Red O). Similar phenotypes have been reported in Wnt10b mutants (Vertino et al., 2005).

NOTE: Figure provided for reviewer has been removed. It showed the panel from figure 4a from Biltz, N.K. and Meyer, G.A. (2017). A novel method for the quantification of fatty infiltration in skeletal muscle. *Skeletal Muscle*. 7, 7. doi: 10.1186/s13395-016-0118-2. We have removed unpublished data that had been provided for the referees in confidence.

As the reviewer notes, we agree that the levels of fat accumulation in the muscles that we report are low and the defect is subtle. We have attempted to examine fatty infiltration by other methods that include H&E (which appear as round 'holes' in sections), Oil Red O, and with Perilipin as the adipocyte marker (a figure from Biltz and Meyer, 2017 is shown above to illustrate this point.)

We asked if del/del muscles stained by H&E exhibit more adipocyte 'holes' compared to wild type in a similar way that del/del muscles exhibit more Perilipin staining compared to wild type. We show results below from a preliminary experiment. The wild-type sample size was small and the data failed to show significance, but we see a trend in which del/del muscles display increased fat compared to wild type, similar to our analysis by Perilipin staining.

As the reviewer points out, Oil Red O is also commonly used (eg. Vertino et al., (2010), (doi/10.1091/mbc.E04-08-0720)), to demonstrate adipogenesis in muscles. Reviewer #2 asks us to quantify Oil Red O in muscles, particularly in the myofibers, which the Vertino paper describes as regenerative defect that is seen in *Wnt10b*^{-/-} muscles. The reviewers should note that *Wnt10b*^{-/-} mice reported in the Vertino et al., paper are of a different strain and were maintained on a high fat diet from 4 weeks onwards in order to see a phenotype. We do not know if the Oil Red staining levels in myofibers reported by the Vertino group stems from the feeding regimen, the strain, the injury model (freeze injury), or because the constitutive *Wnt10b*^{-/-} mutant state promotes a different distribution of staining than what we see following loss of our injury-responsive enhancer on regular chow. We have not observed obvious fatty accumulation in the myofibers but rather observe fatty infiltration that resides *between* the myofibers.

When we performed Oil Red staining (see below), we also failed to observe clear Oil Red accumulation in myofibers. Staining appeared to reside largely between myofibers. We often observed Oil Red droplets spilling or smearing outside of the tissue boundaries however, and smudging and shifting of Oil Red staining into myofiber areas was observed but was non-uniform and not reproducible. We thought this method would likely be not sensitive enough to detect subtle changes in fatty infiltration and opted to use quantification of fatty infiltration using H&E slides instead.

2. Regeneration phenotypes

The regenerative defects in the enhancer deletion lines appear minor, making it difficult to conclude that a meaningful regenerative impairment exists. The authors should consider additional experimental settings to more clearly demonstrate regeneration defects:

1) Sensitized genetic background

Enhancer redundancy may mask phenotypic effects in homozygous enhancer deletion lines. Combining the enhancer deletion allele with the *Wnt1* and/or *Wnt10b* mutant could reveal more pronounced defects. Such combinations could clarify regenerative defects of enhancer deletion.

We tried to set up experiments with sensitized genetic backgrounds but we discovered that the genetic background can have very profound effects on the ability to detect a phenotype, particularly when the defects observed are subtle. Briefly, we tried to cross one copy of a β -catenin Δ allele or one copy of a *Wntless* Δ allele (both in a mixed C57Bl/6 background) into a Δ/Δ (FVB) background in order to ask if we could worsen our phenotype by either reducing β -catenin or by reducing all Wnts that might act together with *Wnt1* and *Wnt10b* to regulate adipogenesis. We were unable to interpret our phenotype in mixed background mice. The

resulting mice displayed increases in adipogenesis but this was observed in both wild-type and $\Delta 1/\Delta 2$ animals, and it was extremely variable. Our attempts at back-crossing back into FVB were not successful past ~5 crosses, and in the end, we failed to perform these experiments.

We also thought about eliminating a single copy of Wnt1 or Wnt10b in Enhancer Δ/Δ mice to ask if we could worsen the phenotype, but these experiments were never performed, due to concerns with background effects as mentioned above. Additionally, we knew that Mendelian ratios from crossing $\Delta 1/+$ mice to $\Delta 2/+$ mice gave reduced numbers of Δ/Δ animals suggesting that a low percentage of animals were dying prior to birth. Given the important role of Wnt1 in CNS development and embryonic lethality of Wnt1 mutant embryos as well as embryonic roles of Wnt10b, we were uncertain how many litters we would have to generate in order to produce enough Δ/Δ ; Wnt $+/+$ animals for injury experiments, and we felt that these studies were prohibitive.

2) Aging

Since Wnt10b mutants display adipogenic defects in aged animals, evaluating regenerative and adipogenic outcomes in aged mice could better demonstrate the functional importance of these enhancers in tissue regeneration.

We tried twice to perform experiments on wt and del FVB mice of ages ranged from 16 to 24 months. Both times, both wild type and del did not survive the injury. This could be due to arrhythmias that are known to occur with the BaCl₂ injury model (eg. Jung et al., 2019 doi:10.4235/agmr.19.0012, Matilla et al., DOI: 10.1007/978-3-642-71248-7_26), which perhaps old mice are less able to tolerate. Properly studying aged animals will require much larger cohorts of mice to establish a proper dose that both permits survival and induces a reasonable level of injury in which adipogenesis can be evaluated. We do not think these experiments will be feasible for us to do at the current time.

We are currently seeing survival of 1 year old mice, however, and in ~one week, we will have tissue samples that can be examined for adipogenesis in wild-type vs. del/del animals. We have some very preliminary data to indicate that fat is elevated in 1 year-old mutant animals compared to wild-type.

3. Enh3 activity quantification and control images

The use of non-transgenic animal images as controls is not appropriate for main figures. The proper control should be uninjured tissue from the enhancer reporter lines, but uninjured images are missing from several key figures. Please provide uninjured lacZ images for the enhancer reporter lines in Figures 2C, 3B, S5, and S7.

We can address the reviewer's concerns about including uninjured tissue panels. We provide a revised figure 2C here. Uninjured Enhancer 3 panel has been added to Figure 2C, and the non-transgenic image has been removed.

Uninjured LacZ images for 3B

Uninjured controls have been provided for Figure S5 and S7.

We will add these into the images for the rebuttal.

Figure 3B should include images of vehicle day 1 (d1) liver, uninjured skin, and uninjured heart.

Non-transgenic images can be moved to supplementary figures.

The figure has been re-made:

Supplementary figure with injured non-transgenics is also included here:

In addition, please quantify the reporter gene expression levels for the enhancer lines to better demonstrate injury-responsive activity. For example, Figure 2C should include a quantification graph comparing uninjured tissue and each post-injury time point (d1, d3, and d5) to clearly show the injury-responsive induction.

A table rather than a graph was generated (please see below, response to Reviewer #2 major comment #6) to describe the expression of each enhancer line that was examined. We do not think a graph is appropriate, as the cells are relatively rare and not easy to quantify accurately, but the table shows that in the uninjured state and is no reporter expression whereas injury induces the appearance of reporter signals.

Also, quantification data of uninjured and each post-injury time point for AP-1 mutated mice should be required as non-specific or leaky expression is visible for some AP-1 mutated mice. Note that AP-1 line 21 1d and 5d in Fig. 3D exhibits LacZ signal.

There is LacZ expression in some of the AP-1 mutant lines post-injury, and this is not surprising or unexpected. Much like embryonic mouse reporter assays where the activity of a given regulatory region is examined by looking at multiple embryos within a litter to identify common expression patterns between individual pups, adult transgenic lines are assessed similarly. Here, each different mutant AP-1 line represents an independent insertion of the transgene into the genome and therefore will likely be influenced by the transcriptional activity of surrounding sequences, or position effect. In these experiments, the critical read-out is the reporter expression across all of the lines that were generated to ask if the transcriptional output is reduced overall.

Since we are assaying multiple tissues, observing elevated activity in some tissues is not surprising. For example, any lines with overall stronger staining (eg. line 14 Fig. S7, lung whole mount, trachea section, liver whole mount) may be due to insertion of the transgene into a particularly active locus. On the other hand, line 21 appears to have less staining in trachea, liver, and skin, but there is still some expression in the lung as assessed in sections, although overall staining in whole-mounts is reduced relative to wild-type Enhancer 3. This may be due to insertion into a locus where the reporter may be less sensitive to transcriptional changes in response to injury due to neighboring loci that may drive higher expression in the lung.

A table that summarizes these data is shown below:

Table: Reporter activity of AP-1 mutant lines compared to wild-type Enhancer 3-LacZ

Reporter Line	Naphthalene injury					CCl4 injury		Punch Biopsy
	lung	lung	lung	lung	trachea	liver	liver	skin
	12h	1 day	3 days	5 days	12h	1 day	1 day	1 day
	whole mount	section	section	section	section	whole mount	section	Whole mount
Enhancer 3 (Wild type)	++++	+++	+++	++	+	++++	+++	++++
AP-1 mutant #21	+	++	+	+++	-	-	-	-
AP-1 mutant #27	-	-	-	-	+	-	-	-
AP-1 mutant #39	-	-	+	+	-	-	-	+
AP-1 mutant #44	+	+	+	+	++	-	-	+

AP-1 mutant #14	++	+	+	+	++	+	+	+
AP-1 mutant #22	-	-	+	+	-	-	-	-
AP-1 mutant #32	-	-	-	-	-	-	+	-
AP-1 mutant #37	+	+	+	+	++	-	+	++

4. Enhancer activity

Lines 237 and 238 propose distinct activities of Enh2 and Enh3. Since the endogenous regulatory activity of Wnt1/Wnt10b likely reflects the combined contributions of these enhancers, it would be informative to evaluate the activity of the 6 kb and the Enh2+3 reporter lines. Do these lines exhibit distinct reporter gene expression patterns compared to the individual enhancers? A more comprehensive and comparative analysis with 6 kb and Enh2+3 lines together with Enh2 and Enh3 will provide mechanistic insights into how multiple enhancers coordinate to regulate a single gene.

The 6kb line is no longer available and we were unable to test its activity in muscle. Enhancer 2+3, however, is induced by injury. Whole mount muscles are shown at 6 hours, 1 day and 3 days post BaCl₂. Arrowheads mark some of the LacZ signals observed at 6 hours and 1 day. Enhancer 2+3 reporter expression is reduced compared to Enhancers 2 and 3 individually at 1 and 3 days suggesting that transcriptional output from the combination of the two enhancers differs from each enhancer alone. This may be consistent with the relatively low induction of Wnt1 and Wnt10b that we observe in the in situ, and supports the idea that this regulatory region can drive Wnt expression following injury. We could re-make figure 2B to include all 3 enhancers.

5. Comprehensive analysis Wnt expression in distinct injury models

One important data in this manuscript is the comprehensive transcriptional analysis of 19 Wnt genes across 4 different injury models. This analysis identified Wnt1 and Wnt10b as being consistently induced in multiple tissue injuries with similar expression patterns, as stated in Lines 112-114. However, the current presentation does not adequately convey the broader Wnt expression results and makes it difficult for readers to interpret the data. To address this, the authors should provide a summary table showing the expression patterns of all 19 Wnt genes across uninjured and post-injury time points, including mouse numbers for each condition. Such a table would be essential to properly support and present this important dataset.

We can generate a summary of our in situ results.

6. Table 1

The current table 1 requires to be revised to add more detailed information. Please indicate line number and corresponding expression results for each time point. For example, Enh line 2 number x, uninjured: no expression or -. Muscle BaCl2 injury day 1 - +, damaged myofiber; day3: ++ damaged and regenerating myofibers; day5: +++++, regenerating myofiber.

Table 1 is a summary table of the expression that was observed when we generated the reporter lines aimed at searching for injury-responsive elements. Because we are analyzing individual adult stable transgenic lines, and each line generated and tested represents an independent transgene insertion at a unique locus within the genome, the assay entails evaluating the consistency of a given expression pattern rather than the levels of reporter expression (which can be influenced by insertion site (ie. position effects)). For this reason, we opted not to include levels data in this table. (This is also the reason why the Enhancer 1 line was considered inconclusive with only an n=2 independently generated lines.)

We can, however, generate a table of reporter activity of the enhancer lines that were analyzed for the muscle, which reflects one representative line each for Enhancer 2, Enhancer 3, and Enhancer 2+3. It would look something like the table below and can certainly be included in the manuscript if the reviewers feel this aids in interpreting the data:

Table: Expression of Injury responsive reporters in muscle

Reporter Line	Timepoint:	Expression Pattern	Expression Level
Enhancer 2, uninjured	6, 12, Day 1, Day 3, Day 5	None (Note: Occasional uninjured myofibers show staining but this is considered non-specific background and is frequently found in lines carrying the hsp68-LacZ construct; number and intensity varies between tissues, this pattern can be seen in all specimens, both uninjured and injured muscles and is therefore not included in the assessment we present here)	-
Enhancer 2	6h	None	-
Enhancer 2	12h	Speckled distribution (whole mount)	+
Enhancer 2	Day 1	Speckled distribution (whole mount), individual cells residing between myofibers (section)	++
Enhancer 2	Day3	Uniform staining throughout tissue (whole mount), individual cells residing between myofibers, staining in myofibers of varying intensity (section)	++++
Enhancer 2	Day 5	Individual cells residing between myofibers, staining in regenerating myofibers of varying intensity (section)	+++++
Enhancer 3	6, 12, Day 1, Day 3, Day 5	None	-
Enhancer 3	6h	Speckled distribution (whole mount), individual cells residing between myofibers (section)	+++
Enhancer 3	12h	Speckled distribution (whole mount), individual cells residing between myofibers (section)	+++
Enhancer 3	Day 1	Speckled distribution (whole mount), individual cells residing between myofibers (section)	+++

Enhancer 3	Day3	Speckled distribution (whole mount), individual cells residing between myofibers (section)	+++
Enhancer 3	Day 5	individual cells residing between myofibers (section), background staining in uninjured myofibers starts to appear	+
Enhancer 2+3	6, Day 1, Day 3	None	-
	6h	Speckled distribution (whole mount)	+
Enhancer 2+3	12h	NA	NA
Enhancer 2+3	Day 1	Speckled distribution (whole mount)	++
Enhancer 2+3	Day3	Speckled distribution (whole mount)	++
Enhancer 2+3	Day 5	NA	NA

7. Wnt1 and Wnt10b expression changes in enhancer deletion lines
 The authors employ Pyrosequencing approach to measure Wnt1 and Wnt10b expression level change by enhancer deletion. However, there is no statistical analysis result. Please perform statistical analysis with sufficient number of samples and provide data. In enhancer deletion mutants, Wnt1/10b expressing cell numbers decline more than half at day 5, so their transcript levels may decline significantly at day 5. Moreover, SNPs in Wnt1/10b regulatory elements across distinct wild-type strains may affect injury-responsive expression.

We will examine the appropriate statistical test for these pyrosequencing data.

8. Active enhancer signature
 There are multiple published ATAC-seq profiles, active enhancer profiles or scATAC-seq datasets for uninjured and injured skeletal muscles. Please examine whether Enh2 and Enh3 become accessible or are marked by active enhancer markers upon injury.

Although there are many deposited data sets that are available, very few replicate our experimental design or are from subsets of cells that may or may not reflect the transcriptional state of our cells of interest. Nevertheless, we provide two examples here.

First, we show a summary figure showing data from Encode and CHIP-seq from various (uninjured) tissues in which Enhancers 2 and 3 are predicted to function as enhancers, and data from skeletal muscle also supports the idea that these regulatory regions consist of open DNA as evidenced by DNase1 hypersensitivity sites in this region.

We also plot data from an ATAC-seq experiment in which fixed satellite cells were analyzed at 1 hour, 16 hours, 32 hours and 60 hours post-BaCl₂ injury and compared to uninjured sample (Dong et al., 2022; DOI:10.1016/j.jisci.2022.104954). It is evident that slightly shifting peaks are present at the site of Enhancers 2 and 3 (marked by a red box), in support of the idea that

these sequences represent sites of open chromatin where there may be increased access to transcription factors.

Minor

- Abstract first sentence is overstated. Please revise it.
"The capacity to detect and respond to injury is critical for the survival of all organisms" -
> Please weaken by removing "all". We can change this wording
- Wnt1 and Wnt10b gene structure
Fig 2A and S4A are low resolution, making difficult to evaluate the data. Which one is Wnt10b and what is the direction? Please annotate "Wnt10b" above left gene and "Wnt1" above right one. Please clearly indicate the direction of both genes. Please include 5.5kb 3' regulatory region containing the embryonic regulatory element in this genome browser. The current format should be improved for publication. We will provide revised figures for the rebuttal.
- AP-1 sequence conservation
Which species are used for sequence conservation study in Fig 2A? Please provide more detailed information of conservation data. Line 276-278 suggests that two AP-1 site sequences across multiple species were compared. If so, please provide the data. Providing actual data will improve this manuscript.

We originally examined conservation of Wnt1 and Wnt10b using the mm9 mouse genome

alignment. The original alignment was performed on a limited number of species:

Please see our response to Reviewer 1, Major comment #3 for alignments of the two AP-1 sites.

4. Line 291-294 and Table2.

This interpretation is overstated as $del1/+$ and $del2/+$ ratios are not increased. If there is any detrimental effect of homozygotes, all of other genotypes, wt, $del1/+$ and $del2/+$, should increase rather than only wt. Please revise this statement.

We would change this to something such as:

We obtained viable $\Delta1/\Delta2$ animals, but we noticed that Mendelian ratios were lower than expected for the $\Delta1/\Delta2$ (19.44%) mice, suggesting that alleles carrying deletions might be detrimental to the animals. Consistent with this, the reduced numbers of $\Delta1/\Delta2$ animals predicted that, if all equally healthy, the remaining classes each should produce ratios of ~26.85%. However, $\Delta1/+$ and $\Delta2/+$ animals were also obtained at slightly lower ratios ($\Delta1/+$, 25.66%; $\Delta2/+$, 24.34%) suggesting that loss of the Enhancer 2 and 3 region may have mild deleterious effects even in a heterozygous state.

5. Correct typo in line 336. It says (Figure D, E, ...) and should be corrected to reference Fig 5D, E. Will be corrected in manuscript.

6. Correct typo in lines 912 and 914, which says $*P<0.05$, $**P<0.05$. Will be corrected in manuscript.

7. In the discussion in lines 455-458, the authors claim that this is the first example of an enhancer region driving developmental signaling molecules that is re-used in adulthood to facilitate regeneration. However, no data are provided to show that Enh2 or Enh3 can drive expression during development. Please perform reporter assays in developing tissues to demonstrate their developmental activity. If such data are not available, this statement should be revised accordingly to avoid overinterpretation.

Our wording of “...first example of an enhancer region driving developmental signaling molecules that is re-used in adulthood to facilitate regeneration” here was confusing and incorrect. What we meant to say was that the Wnt genes, which are important embryonic signaling molecules, are re-used to drive regeneration, and that our enhancer is the first example in the mouse of a regulatory region that facilitates Wnt expression during regeneration. We did not mean to state that it was the enhancer region that was used during development. We will fix this wording in the manuscript.

We do see embryonic reporter expression (as shown above, see Reviewer #1, Major comment #4).

8. Wnt1 and Wnt10b co-regulation by Enh2 and 3

If Wnt1 and Wnt10b are regulated by the same enhancer elements, why do most nuclei express either one gene or the other, rather than both (as shown in Fig. 1G and discussed in Lines 149-154)? The fact that only about 10% of cells co-express both genes may not provide strong

support for the claim that "common regulatory element" drives their expression. The authors should address this discrepancy in the discussion. Note that Wnt10b level seems to be more influenced by enhancer deletion, suggesting the presence of other regulatory element controlling Wnt1.

Briefly, we would add something to the effect of "the main reason why we would consider our enhancer region to be a 'common regulatory element' is because loss of the Enhancer 2 and 3 region results in reduced expression of *both* Wnt1 and Wnt10b post-injury (Fig. 4F, G)".

The lack of frequent overlap between the two Wnts in our in situs, however, which reflect gene expression driven by the full cohort of enhancer elements that influence their transcription, could be due to several other factors. For example, other enhancers could fine tune the spatial and temporal pattern of Wnt expression. Alternatively or additionally, differences in the stability of the two Wnt transcripts and/or how the two Wnt promoters engage with the enhancer region to activate expression could explain the presence of Wnts in one or another cell at a given time but not both.

Additionally, while TAD (topologically associating domain) data are mentioned in the discussion, no such data are provided. Including a TAD map as a supplementary figure would strengthen this claim - please provide this data if available.

There is TAD data to suggest that the two Wnt genes may reside in a single topological domain (as shown further below) and although the TAD data come from cell lines in a non-injury context, they suggest that there may be short-range contact points that could allow for both Wnts to be regulated by Enhancers 2 and 3. A figure from the UCSC genome browser showing data from human ESCs and human foreskin fibroblast (HFFc6) cells is included. We have marked the triangular zone of interaction points in the region of the Wnt1 and Wnt10b genes with dotted lines. The fact that two different cell lines show a similar TAD map supports the idea that this region is a single TAD. Locations of mapped CTCF sites that have been identified are also included here (data from all cell types, not only hESCs and HFFs). We can see CTCF sites that lie in the vicinity of the edges of the TAD spanning Wnt1 and Wnt10b. We did not think it appropriate to add a data figure in the supplement when raising this point in the discussion section, but we could move this point to the main results section if the reviewers this would be useful for readers.

First revision

Author response to reviewers' comments

Thank you to the reviewers for their thoughtful comments on the manuscript. Many of the questions raised were also ones that we had thought about and tried to address experimentally. We have added data where we could directly reply to the concerns. When this was not possible, and where we were unable to perform conclusive experiments, we have included a discussion below or have included explanations as part of a 'limitations of the study' section in the discussion. We have added several new tables addressing the reviewer's comments (Supplementary Tables 1, 2, 3) and new supplementary figures. We refer to them as we reply to the reviewer's comments.

We also include below a more concise list of changes that we made to the paper in response to the reviewer's comments which we hope is helpful as an overview of what was done.

Summary of changes and responses to reviewer comments:

Reviewer#1

Major comment 1:

- A discussion of why a complete survey of other regulatory regions is beyond scope of this paper, and why examining Enhancer 1 will be challenging due to proximity to the Wnt10b promoter has been added to the rebuttal.

Major comment 2:

- QPCR of Axin2 at d5 post-injury is now included in Fig. S14.
- QPCR data for PPAR γ and CEBP α , two regulators of adipogenesis are added to show that expression of these genes increase when Enhancers 2+3 are lost (Figure 5F, G)

Major comment 3:

- Experiment to look for effects of Enh2+3 deletion on regeneration in another tissue (skin) has been added as Fig. S17

Major comment 4:

- Embryonic expression of Enhancers 2, 3, and 2+3 is provided in Fig. S12.

Minor concerns 1:

- We clarified sample sizes and re-made violin plots with larger dots to make them more visible (Fig. 4F-H).

Minor concerns 2:

- Fig. 2B has new sharper whole-mount images, 3B-D were revised where we could add new images.

Minor concerns 3:

- A figure to describe the conservation of the AP-1 binding sites has been added as Fig. S10 and we discuss this more clearly in the results section.

Minor concerns 4, Discussion:

- We temper our evolutionary claim and provide discussion about why Wnt5a is not affected in our experiments.

Reviewer#2**Major comment 1:**

- We re-plotted adipogenesis vs. injury linear regression to focus on more highly injured specimens and discuss this in the rebuttal (the initial graph is still included in Figure 5).
- QPCR data for 2 adipogenesis markers (CEBP α and PPAR γ) is added to Fig. 5F, G.
- Quantification of adipogenesis on H&E slides as a different measure of fatty infiltration is added to the manuscript (Fig. 5E).
- We discuss in the rebuttal why measuring adipogenesis with Oil Red O was not performed.

Major comment 2:

- Both experiments in sensitized backgrounds and aged animals were performed but were unsuccessful. We discuss this in the rebuttal and added these topics to the discussion as part of a 'limitations of this study' section.

Major comment 3:

- Figure 3B has uninjured controls from Enhancer 3 mice.
- Examples of injured non-transgenic tissue from Fig. 2C is now move to Fig. S8.
- We have added representative controls to Fig. S6 (previously S5)
- We have added controls to Fig. S11 (previously S7) and Figure 3, and we have simplified the figures.
- Supplementary Table 3: A new table that quantifies AP-1 mutant LacZ expression.

Major comment 4:

- Enhancer 2+3 expression at different timepoints in the muscle has been added to Figure 2.
- Supplementary Table

Major comment 5:

- Data from the in situ screen showing examples of all 19 Wnts in 4 tissues are added as Fig, S1A-D with a summary in Supplementary Table 1.

Major comment 6:

- We do not supply a more detailed Table 1 but discuss our reasoning in the rebuttal. We do provide Supplementary Table 2, focusing on reporter expression levels at different timepoints post-injury in muscle.

Major comment 7:

- We simplified the pyrosequencing data by only focusing on the injured samples, clarified our sample numbers in the figure legend, and performed statistics on wtFVB/wtCAST vs. delFVB/wtCAST samples (Fig. S13)

Major comment 8:

- Fig. S7 has been added to show data from the UCSC genome browser to support the idea that Enhancers 2 and 3 could function as enhancers (eg. DNase1 hypersensitivity, H3K27Ac). We also include some ATAC-seq data in the rebuttal but did not add this to the main text.

Minor comment 1:

- We changed wording of the abstract to address reviewer's concern.

Minor comment 2:

- We have revised the figures with better resolution.

Minor comment 3:

- Multi-species alignment of the interval between Wnt1 and 10b to show how our sequences were identified as conserved has been added to Figure 2.
- The discussion of AP-1 sequence conservation has been clarified in the main text, and Fig. S10 shows alignments and conservation of the two putative AP-1 sites.

Minor comment 4:

- We provide improved wording on the reduced Mendelian frequencies that result from our Enhancer deletion crosses in the main text.

Minor comment 5:

- Typos have been corrected.

Minor comment 6:

- Typos have been corrected.

Minor comment 7:

- Our original statement was poorly worded, and did not convey the meaning we were trying to impart to the reader. This text has been corrected and simplified.
- We did not ask why embryos carrying Enhancer deletions exhibit a low level of lethality as observed from our Mendelian ratios, but we provide Fig. S12 which shows some of the embryonic expression patterns displayed by our reporter mice that may reflect important sites of Wnt expression.

Minor comment 8:

- We added to the discussion why we think Wnt1 and Wnt10b expression does not overlap completely, and the possibility that other enhancers may also participate in regulating Wnt1 and Wnt10b expression.
- We also provide a new figure from the UCSC genome browser showing that Wnt1 and Wnt10b may reside in a TAD (Fig. S7) .

Responses to Reviewer's comments:**Reviewer #1:****Major Concerns****1. Enhancer Specificity and Redundancy:**

The partial reduction of Wnt1/Wnt10b expression in $\Delta 1/\Delta 2$ mice (Fig. 4F-G) and residual adipogenesis (Fig. 5) suggest compensatory mechanisms. The authors should test whether other nearby conserved regions (e.g., Enhancer 1) contribute to injury-responsive Wnt expression. Alternatively, the authors could, if possible, perform chromatin conformation capture (3C or Hi-C) to confirm physical interaction between the enhancer and Wnt promoters in injured muscle.

We did not mean to imply that we identified the *only* injury-responsive Enhancer that

could drive Wnt1 and Wnt10b post-injury. This study was not intended to be a comprehensive characterization of injury responsive enhancers that regulate Wnt1 and Wnt10b, and it is very likely that there are indeed, other regulatory elements that control Wnt1 and Wnt10b expression during regeneration. Moreover, we observe other Wnts such as Wnt5a within injured muscles that have been implicated in regulating adipogenesis and that may act redundantly with Wnt1 and Wnt10b. We have added into the discussion more points about what we do and do not know in terms of the regulation of Wnt by our enhancers. We also clarify to the readers that a systematic approach may someday be able to reveal more completely, the regulatory logic of Wnt1 and Wnt10b expression post-injury.

As the reviewer noticed, Enhancer 1 could indeed be an interesting redundant candidate enhancer, as it displayed reporter expression in one out of two lines. However, we cannot distinguish if reporter expression from this single positive Enhancer 1 line represents a real injury response or simply the result of position effects on the construct. Additionally, Enhancer 1 is less than 1 kb from the transcriptional start site Wnt10b, so the proximity to the promoter makes deletion experiments that would show requirement of the enhancer difficult to interpret, so we did not opt to study Enhancer 1 further.

Definitively proving a requirement for Enhancer 1 or any other candidate Enhancer would necessitate producing multiple new independently generated lines of adult reporter mice (with each independent adult founder mouse bred at least through the G2 generation to adulthood to verify activity), as well as deletions to test for a requirement in driving Wnt1 and Wnt10b expression, a multi-year endeavor that we is not currently feasible and beyond the scope of this work.

2. Mechanistic Link Between Wnt Loss and Adipogenesis:

The adipogenic phenotype is attributed to reduced Wnt signaling, but direct evidence is lacking. The authors should either measure β -catenin activation (e.g., Axin2 expression) in $\Delta 1/\Delta 2$ muscles post-injury, or rescue the phenotype by administering Wnt1/Wnt10b agonists or inhibiting adipogenic pathways (e.g., PPAR γ).

We have included Axin2 qPCR data at 5 days post-injury and expression of adipogenic regulators (CEBPa and PPARg) at 7 days post-injury to show that in del animals there is a slight drop in Axin2 expression and an upregulation of adipogenesis markers in del animals compared to wild type. Although the data do not show statistical significance, we see a slight decrease for Axin2 and an upwards shift for the fat markers. We added these data to **S14F** (Axin2), and **Figure 5F, G** (CEBPa and PPARg).

As the reviewers point out, the phenotype is rather subtle. Given that the mice carry an enhancer (rather than a coding sequence) deletion, combined with the variability in injury response between mice and variations in administering the injury model itself, it has been difficult to get dramatic changes in gene expression in whole-muscle assays. We argue, however, that the trends in gene expression are consistent with the model that loss of Wnt expression results in reduced Wnt signaling which is manifested as an increase in fatty infiltration of muscle, as supported both by the phenotype as well as up-regulation of adipogenic markers (Perilipin, CEBPa, PPARg **Figure 5** and **Fig. S14**).

Additionally, as now shown in **Fig. S1B** and **Supplementary Table 1** which provides data from the full 19-Wnt in situ screen in the muscle, there are many Wnts expressed in the muscle post-injury, including Wnt5a that has been implicated in adipogenesis in muscle (Reggio et al., 2020). Wnt5a does not change expression when our enhancers are deleted (**Figure 4H**). It is possible that these other Wnts provide compensatory signals and largely maintain proper fate choice of cells within the muscle. In our previous draft we did not propose a model because we do not perform experiments aimed at identifying which differentiate into adipocytes and how that occurs, but we now include some discussion around this topic in the discussion section, with the caveat that further experiments will be needed.

3. Tissue Specificity of the Enhancer:

The enhancer is active in lung, liver, and skin (Fig. 3B), but regeneration defects are only analyzed in muscle. The authors may wish to assess regeneration in other tissues (e.g., lung after naphthalene injury) to determine if the enhancer's role is muscle specific.

Our current data indicates that the role of Enhancer 2 and 3 is muscle specific. We looked at lung, skin, and muscle and only detected a phenotype in the muscle. We have generated a new Figure S17 which shows data from the skin.

To summarize:

- 1) In the lung, injured airways of mice lacking Enhancer 2+3 can regenerate, and airway cells are restored. We were unable to detect an obvious defect. The lung displays very strong Wnt7b expression (Fig. S2C) which might mask and compensate for the loss of Wnt1 and Wnt10b, and we did not follow up with the lung further. It is possible that there could be a subtle phenotype that we missed.
- 2) In the skin, punch biopsy wounds healed over a similar time course in wild type vs. del animals and we did not detect any obvious defect (Fig. S17D). In contrast to the lung where Wnt1 and 10b are induced and Enhancers 2 and Enhancer 3 are activated in response to injury, the skin fails to up-regulate Wnt1 and Wnt10b at the wound site (Fig. S17B) even though Enhancers 2 and 3 are activated (Fig. S17A). Moreover, there is no activation of Enhancer 2+3 lacZ following punch biopsy (Fig. S17A). These data suggest that the skin and muscle regulate Wnt10b expression from the enhancers differently, and that activation of individual injury-sensing elements in a given locus may not always necessarily translate into target gene expression. This highlights the role of tissue context and enhancer interactions in determining if a gene is ultimately expressed.

4. Developmental Impact of Enhancer Deletion:

Sub-Mendelian ratios in $\Delta 1/\Delta 2$ mice (Table 2) suggest possible embryonic roles. Evaluate embryonic Wnt1/Wnt10b expression and developmental phenotypes (e.g., CNS defects) in $\Delta 1/\Delta 2$ embryos.

As the reviewers point out and that we stated in the text, reduced Mendelian ratios of del/del mice suggest that Enhancers 2 and 3 in may have embryonic functions that are detrimental when deleted. We do see embryonic expression of Enhancer 2, 3 and 2+3 (Fig. S12). We did not attempt to evaluate why the animals would die for two main reasons.

- 1) First, embryonic LacZ reporter staining of Enhancer 2, Enhancer 3, and Enhancer 2+3 reveal very complex and dynamic staining patterns, making it difficult for us to identify a specific spatial pattern and timepoint most likely to produce lethality. We now include examples of embryonic lacZ staining in Fig. S12 to illustrate some of the LacZ expression patterns that we observed. Patterns range from axial ventral neural tube expression to the appearance of various domains in the brain, face, limbs, tail as well as genital tubercle in older embryos. Whether all of these patterns reflect sites of Wnt1 and/or Wnt10b expression is not clear. The well-known Wnt1 CNS expression pattern that is known to produce lethality in Wnt1 mutant embryos was not observed, consistent with the idea that a 3' 5.5kb regulatory element in Wnt1 is responsible for this expression domain.
- 2) Second, because only about a quarter of del/del embryos are lost sometime during embryogenesis, we would predict that most del/del embryos would be fairly normal. We would need to generate a large number of litters at different embryonic timepoints to assess when animals are being lost and then focus on that timepoint to identify a defect in a minority of del/del mice. Although identifying how del animals die might reveal new patterns or sites of Wnt1/Wnt10b function, we felt that this was a separate project unto itself and would be better done in a different study.

Minor Concerns

1. Quantification and Statistics:

- a. Clarify sample sizes in figure legends (e.g., Fig. 1C-D: n=8 uninjured vs. n=5 injured). Sample sizes have been clarified.

b. Include individual data points in violin plots (Fig. 4F-H) and specify statistical tests used for each analysis.

We were unsure what the reviewer was asking for here, as individual data points were included in the original manuscript for the d3 timepoints for Wnt1 and Wnt10b. However, we wondered if the dots were too small for readers to see, so we increased the size of dots in the violin plots (Figure 4F, G). We also added a violin plot for Wnt5a for comparison (Figure 4H).

Statistical tests state in the figure legend: (F-H) Whiskers in Box plots represent the Min to Max. (Wnt1 d3, Mann-Whitney test; Wnt1 d5, * $p < 0.05$., Mann-Whitney test; Wnt10b d3, Unpaired t-test; Wnt10b d5, * $p < 0.05$ Unpaired t-test; Wnt5a d3, Welch's t-test; Wnt5a d5 Unpaired t-test; ns = not significant) On violin plots, the red line represents the median. (Scale bar = 20 μ m)

2. Figure Clarity:

a. Improve resolution of LacZ staining in whole-mount images (Fig. 2B, 3B-D).

Figure 2B: We added a new Figure 2B with sharper images of the whole mounts. Because the cells are deeper inside the muscle tissue, the staining will always be somewhat indistinct, but resolution should be improved.

Figure 3B-D: We tried to improve the resolution of the images in Figure 3B, C and to improve the other images where we could. We are aware that the accessory lobe images, from Figure 3D in particular (Also Fig. S11), are poor (due to the imaging method we used at the time). We think the whole-mount lung lobes provides a different way to view the lung tissue, but we can certainly eliminate these panels.

b. Label cell types in Fig. 1E-G (e.g., FAPs, macrophages) to aid interpretation.

We have provided a revised panel in Figure 1E and F.

In response to the reviewer's comments regarding Figure 1G, the two in situ signals shown are Wnt1 and Wnt10b which are labeled in red and blue font to correspond to the in situ signal color. Since they do not refer to cell types, there is no additional cell type information we can add to those panels.

3. AP-1 Binding Site Conservation:

The second AP-1 site in Enhancer 3 is noted as divergent in newer genome alignments. Perform phylogenetic analysis to assess conservation across 240 mammalian species (as referenced) and discuss implications.

At the time of our initial analysis of Enhancer 3, two putative AP-1 sequences appeared to be present in the mouse sequence. We opted to mutate both sequences to be sure that we eliminated AP-1 binding activity from the enhancer in our site-directed mutagenesis experiments. In all alignments that are available (mm9, mm10 and mm39) the mouse reference sequence (and our FVB animals from sequencing) contains consensus AP-1 binding sites at both positions. When other species are included in the alignment however, we see that the second site is not a conserved AP-1 binding site.

We have added text to explain this more completely in the results section. We have added Fig. S10 which contains 3 panels: Fig. S10A shows the conserved consensus sequence (site #1), Fig. S10B shows the non-conserved consensus sequence (site #2), and Fig. S10C shows a larger multi-species alignment with site #2, focused on rodents to illustrate that only the house mouse has the consensus AP-1 binding sequence at this position. For our experiments, we still proceeded to mutate both putative AP-1 sites, as we wanted to ensure that we would eliminate AP-1 binding. It is possible that the second "non-conserved" site is not utilized, but we would interpret our observations to say that one or

both AP-1 sites are likely required for the injury-responsiveness of Enhancer 3.

4. Discussion:

a. Temper evolutionary claims about conserved Wnt regulatory mechanisms until functional studies in non-mammalian models (e.g., zebrafish, killifish) are performed.

We added the following:

“Given Wnt induction in virtually all injury contexts that have been described to date, it is interesting to consider whether there might be common mechanisms that activate different Wnts post-injury, or even whether there could be deeply conserved injury-induced regulatory mechanisms for Wnt expression that spans across invertebrate to vertebrate species. **A definitive answer to these possibilities will require further functional studies in other non-mammalian species.**”

b. Address why Wnt5a (Fig. 4H) is unaffected by enhancer deletion, given its known role in muscle regeneration.

Although Wnt5a is thought to function in muscle regeneration and has been postulated to regulate adipogenesis, it is unaffected by enhancer deletion because it is likely regulated by its own injury-responsive enhancers and its expression is not dependent or downstream of Wnt1 and Wnt10b. A sentence has been added to the text in the results section as follows:

In contrast to Wnt1 and Wnt10b, Wnt5a expression appeared unchanged in injured wt vs $\Delta 1/ \Delta 2$ tissues, as assessed visual examination of transcript staining in sections (Figure 4, D, E), when the number of Wnt5a positive nuclei was quantified (Figure 4H). These data are consistent with the hypothesis that Enhancers 2 and 3 specifically drive Wnt1 and Wnt10b expression and deletion of this Enhancer does not affect other Wnts. Specifically, any role that Wnt5a may have in regulating adipogenesis (Reggio et al., 2020) is likely to be independent of Wnt1 and Wnt10b.

Reviewer #2:

Majors

1. Regeneration assays

According to the methods and main text, the authors calculate an injury index to account for variability in injury severity induced by BaCl₂. Figure 5E shows that injury levels among mice range from ~20% to 80%, a probably, unacceptably large variation that complicates reliable analysis.

For Figure 5E, we thought that showing the full span of possible injury ranges would be the most honest way to display our results. We would normally consider a 20% injury to be due to poor injection technique and would not include such mildly injured specimens in any other type of analysis, preferring to examine sets of more similarly severely injured samples to simplify the analysis. For these experiments, however, because we saw subtle changes in fatty infiltration (perhaps not surprising since we were trying to analyze loss of a small regulatory region), we wanted to keep all the samples that we generated. By assessing adipogenesis together with the level of perturbation at all levels of injury we thought we would avoid reporting differences in fatty infiltration simply because the cohorts were injured to different degrees.

If we analyze the data only including the more highly injured samples, we see a similar upward shift in the adipogenesis exhibited by del/del animals to when the full dataset is plotted. Regression lines still support the idea that the wild-type mice overall generate lower levels of adipogenesis in the muscles compared to the del/del animals. We obtain a p-value of 0.0863 when we ask if the elevations of the lines are significantly different (ie. not quite significant), which is not surprising given the spread of the del/del data points. But we posit that these data reflect a defect in which control of adipogenesis becomes more dysregulated when the enhancer is deleted. We have kept the original graph in Figure 5 but can change this if necessary.

As shown already in response to Reviewer #1 comment #1, an alternative way to demonstrate elevated adipogenesis in Enhancer del/del animals compared to wild type, is to measure expression of genes involved in driving adipogenesis, such as CEBPa and PPARg. Using QPCR, we compared levels between wild-type and del/del mice at 7 days post injury and observed elevation of these genes in del/del animals compared to wild-type. Because the difference in adipogenesis is subtle between wt and del animals, we were unable to obtain clear statistical significance, but one can see an upwards shift for both markers in del/del tissue compared to wild type. These graphs have been added to Figure 5F, G.

Furthermore, the reported adipogenesis level appears extremely limited (~0 - 2%). There is also no clear evidence that adipogenic signals are present nearby or among regenerated muscle fibers. The authors should evaluate regenerative defects using alternative approaches. For example, they could quantify lipid accumulation in the regenerated muscle area or assess how many regenerated muscle fibers (central-nucleated cells) are positive for lipid staining (e.g., Oil Red O). Similar phenotypes have been reported in Wnt10b mutants (Vertino et al., 2005).

NOTE: Figure provided for reviewer has been removed. It showed the panel from figure 4a from Biltz, N.K. and Meyer, G.A. (2017). A novel method for the quantification of fatty infiltration in skeletal muscle. *Skeletal Muscle*. 7, 7. doi: 10.1186/s13395-016-0118-2. We have removed unpublished data that had been provided for the referees in confidence.

As the reviewer notes, we agree that the levels of fat accumulation in the muscles that we report are low and the defect is subtle. We have attempted to examine fatty infiltration by other methods that include H&E (which appear as round 'holes' in sections), Oil Red O, and with Perilipin as the adipocyte marker (a figure from Biltz and Meyer, 2017 is shown above to illustrate this point.)

Early on when we generated our first Enhancer deletion animals, we We asked if del/del muscles stained by H&E exhibit more adipocyte 'holes' compared to wild type in a similar way that del/del muscles exhibit more Perilipin staining compared to wild type. We preferred to use some sort of molecular marker for fat, but we did try a small experiment looking at whether elevations in fat could be observed when both using H&E vs Perilipin. The original wild-type sample size was small, and the data failed to show significance, but we see a trend in which del/del muscles display increased fat compared to

wild type, similar to our analysis by Perilipin staining.

In Fig. 14E, we have now added a more recent analysis with a side-by-side comparison of Perilipin and H&E analysis. Samples for H&E analysis are muscles that show >45% injury and 10 fields per muscle that were quantified for the area of the holes observed where fatty infiltration has occurred. Adipogenesis index values (% adipogenesis/extent of injury) are plotted to normalize for the amount of injury displayed by the muscles.

The overall y-axis values are larger because these are areas rather than circumference, but there is a significant difference between wt vs. del muscles, similar to the shift that is observed when Perilipin is quantified.

As the reviewer points out, Oil Red O is also commonly used to detect adipogenesis and has been used in muscle (eg. Vertino et al., (2010)). Reviewer #2 asks us to quantify Oil Red O in muscles, particularly in the myofibers, which the Vertino paper describes as a regenerative defect that is seen in *Wnt10b*^{-/-} muscles. The reviewers should note that *Wnt10b*^{-/-} mice reported in the Vertino et al., paper are of a different strain and were maintained on a high fat diet from 4 weeks onwards to see a phenotype. We do not know if the Oil Red staining levels in myofibers reported by the Vertino group stems from the feeding regimen, the strain, the injury model (freeze injury), or because the constitutive *Wnt10b*^{-/-} mutant state promotes a different distribution of staining than what we see following loss of our injury-responsive enhancer on regular chow. We have not observed obvious uniform fatty accumulation in the myofibers but rather observe fatty infiltration that resides *between* the myofibers.

We previously performed a preliminary experiment using Oil Red staining on slides at 21 days post-injury. We quantified Oil Red staining area in n=3 wt vs. n=3 del/del animals with 3 fields per animal, and we did see a three-fold increase in Oil Red signal in mutant injured muscles compared to wild-type (*P<0.05, Welch's t-test). We did not add this to the paper due to the small sample size, but we include this here to show that we detect an upwards shift in adipogenesis in del/del muscles compared to wild-type by this method as well.

We generally did not like Oil Red staining. First, it requires cryosections in which, in our

hands, shows poorer tissue preservation than paraffin sections, and is not as useful for other analyses. We also often observed Oil Red droplets spilling or smearing outside of the tissue boundaries, and smudging and shifting into myofiber areas in a way that was non-uniform and inconsistent. Some examples of staining we observed are shown in the images below to illustrate this point. We felt that Perilipin offered a more specific and more consistently staining marker for the presence of fat. Now with H&E data (Fig. S14E) and CEBPs/PPAR α qPCRs (Fig. 5F, G) we hope that we can convince readers that the adipogenic phenotype is a believable defect that can be detected by multiple methods.

2. Regeneration phenotypes

The regenerative defects in the enhancer deletion lines appear minor, making it difficult to conclude that a meaningful regenerative impairment exists. The authors should consider additional experimental settings to more clearly demonstrate regeneration defects:

1) Sensitized genetic background

Enhancer redundancy may mask phenotypic effects in homozygous enhancer deletion lines. Combining the enhancer deletion allele with the Wnt1 and/or Wnt10b mutant could reveal more pronounced defects. Such combinations could clarify regenerative defects of enhancer deletion.

We also thought about these types of experiments and attempted them but were unable to interpret the results due to the influence of genetic background on the phenotype. Specifically, we tried reducing b-catenin by one copy (b-catenin Δ) to see if loss of Wnt signaling would worsen the adipogenic phenotype, and we also tried reducing Wntless by one copy (Wls Δ) to see if reduction of Wnt secretion overall would have a similar effect. We added text to describe this in the 'limitations of this study' section in the discussion.

To the reviewer's specific suggestion: We also thought about eliminating a single copy of Wnt1 or Wnt10b in Enhancer Δ/Δ mice to ask if we could worsen the phenotype, but these experiments were never performed, due to concerns with background effects as mentioned

above. Additionally, we knew that Mendelian ratios from crossing $\Delta 1/+$ mice to $\Delta 2/+$ mice gave reduced numbers of Δ/Δ animals suggesting that a low percentage of animals were dying prior to (or very shortly after) birth. Given the important role of Wnt1 in CNS development and embryonic lethality of Wnt1 mutant embryos as well as embryonic roles of Wnt10b, we were uncertain how many litters we would have to generate to produce enough Δ/Δ ; Wnt $+/ -$ animals for injury experiments, and we felt that these studies were prohibitive.

2) Aging

Since Wnt10b mutants display adipogenic defects in aged animals, evaluating regenerative and adipogenic outcomes in aged mice could better demonstrate the functional importance of these enhancers in tissue regeneration.

We also thought examining aged animals would be useful. We tried twice to perform experiments on wt and del FVB mice of ages ranged from 16 to 24 months. Both times, neither wild type nor del animals survived the injury. This could be due to arrhythmias that are known to occur with the BaCl₂ injury model (eg. Jung et al., 2019, Matilla et al., 1986), which perhaps old mice are less able to tolerate. We tried a third time on a cohort of animals ranging from 15-20 months, at a slightly lower dose. These animals survived but we observed poor injury. Studying aged animals will require much larger cohorts of mice to establish a proper dose that both permits survival and induces a reasonable level of injury in which adipogenesis can be evaluated and will be better suited to a separate study. We include a description of these attempts in the discussion as one of the limitations of our study.

3. Enh3 activity quantification and control images

The use of non-transgenic animal images as controls is not appropriate for main figures. The proper control should be uninjured tissue from the enhancer reporter lines, but uninjured images are missing from several key figures. Please provide uninjured lacZ images for the enhancer reporter lines in Figures 2C, 3B, S5, and S7.

Figure 2C: (representative examples of staining from uninjured Enhancer 2 and 3 mice are shown)

Figure 3B should include images of vehicle day 1 (d1) liver, uninjured skin, and uninjured heart. Non-transgenic images can be moved to supplementary figures.

For **Figure 3B**, uninjured Enhancer 3 LacZ images now replace the non-transgenic animal images, and non-transgenic data are now shown in **Fig. S8**.

Uninjured controls have been provided for Figure S5 and S7.

S5 (now S6): Representative uninjured Enhancer 2 and Enhancer 3 images are now included. We did not include images for each timepoint, because uninjured samples show no LacZ staining at any timepoint and we felt that adding more panels would make the figure unnecessarily large.

S7 (now S11): We re-made **Figure 3D** to show uninjured Enhancer 3 tissues in the main figure, compared to injured Enhancer 3 and one example of an AP-1 mutant. We Also re-made **Figure S11** to show one representative example of uninjured AP-1 mutant tissue staining and all 7 AP-1 mutant lines, as we felt that Figure 3 was unnecessarily large and repetitive.

In addition, please quantify the reporter gene expression levels for the enhancer lines to better demonstrate injury-responsive activity. For example, Figure 2C should include a quantification graph comparing uninjured tissue and each post-injury time point (d1, d3, and d5) to clearly show the injury-responsive induction.

A table rather than a graph was generated to describe the expression of each enhancer line that was examined (please see below, response to Reviewer #2 major comment #6). Because there is non-specific hsp-68 LacZ that is found sporadically in uninjured muscles

and because the single cells in tissues post-injury are relatively rare, we did not think our quantifications would be accurate. In table form, however, we could better summarize the levels of signal that we observed and describe the spatial distribution as well. We have included this as Supplementary Table 2.

Also, quantification data of uninjured and each post-injury time point for AP-1 mutated mice should be required as non-specific or leaky expression is visible for some AP-1 mutated mice. Note that AP-1 line 21 1d and 5d in Fig. 3D exhibits LacZ signal.

We did not specifically quantify AP-1 expression levels in each sample, but we do provide a new table (Supplementary Table 3) that summarizes the overall intensity of expression observed by categorizing occasional (+/-), low(+), medium(++), and high(+++), very high (+++) expression.

For these transgenic assays read-outs, we were more interested in whether there was an overall reduction in expression *across all lines generated*, rather than specific levels in each line in different tissues. Much like embryonic mouse reporter assays where the activity of a given regulatory region is examined by looking at multiple embryos within a litter to test for similar expression patterns between individual pups, adult transgenic lines are assessed similarly. Here, each different mutant AP-1 line represents an independent insertion of the transgene into the genome and therefore will likely be influenced by the transcriptional activity of surrounding sequences, or position effects. In these experiments, the critical read-out is the overall *consistency* of reporter expression across all lines that were generated to ask if the transcriptional output is reduced. From the table, it is evident that Enhancer 3 has the highest activity, and the mutant lines display lower induction of the reporter.

4. Enhancer activity

Lines 237 and 238 propose distinct activities of Enh2 and Enh3. Since the endogenous regulatory activity of Wnt1/Wnt10b likely reflects the combined contributions of these enhancers, it would be informative to evaluate the activity of the 6 kb and the Enh2+3 reporter lines. Do these lines exhibit distinct reporter gene expression patterns compared to the individual enhancers? A more comprehensive and comparative analysis with 6 kb and Enh2+3 lines together with Enh2 and Enh3 will provide mechanistic insights into how multiple enhancers coordinate to regulate a single gene.

The 6kb line is no longer available and we were unable to test its activity in muscle. Enhancer 2+3, however, is induced by injury and we have added these data to a new version of Figure 2B, C. Enhancer 2+3 reporter expression is reduced compared to Enhancers 2 and 3 at 1 and 3 days, and does not display expression in regenerating myofibers, in contrast to Enhancer 2. These data suggest that each enhancer region is integrating transcription factor binding individually, but that the combined output of the two enhancers differs from each enhancer alone.

Enhancer 2+3 expression is also more consistent with the relatively low induction of Wnt1 and Wnt10b that we observe in the in situs and supports the idea that this regulatory region allows Wnt1 and Wnt10b to become expressed after injury. Then, upon enhancer deletion, sustained expression of the Wnts is not possible and expression decreases between days 3-5 more quickly than that seen in wild-type animals. This point is now included in the discussion.

5. Comprehensive analysis Wnt expression in distinct injury models

One important data in this manuscript is the comprehensive transcriptional analysis of 19 Wnt genes across 4 different injury models. This analysis identified Wnt1 and Wnt10b as being consistently induced in multiple tissue injuries with similar expression patterns, as stated in Lines 112-114. However, the current presentation does not adequately convey the broader Wnt expression results and makes it difficult for readers to interpret the data. To address this, the authors should provide a summary showing the expression patterns of all 19 Wnt genes across uninjured and post-injury time points, including mouse numbers for each condition. Such a table would be essential to properly support and present this

important dataset.

Data from the in situ screen for all 4 tissues as well as a summary table are now included in the manuscript. (Supplementary Figure 1 A-D shows all 4 tissues, and Supplementary Table 1 summarizes the expression of the Wnts.)

6. Table 1

The current table 1 requires to be revised to add more detailed information. Please indicate line number and corresponding expression results for each time point. For example, Enh line 2 number x, uninjured: no expression or -. Muscle BaCl₂ injury day 1 - +, damaged myofiber; day3: ++ damaged and regenerating myofibers; day5: + + + +, regenerating myofiber.

Table 1 is a summary table of the expression that was observed when we generated the reporter lines aimed at searching for injury-responsive elements. Because we are analyzing individual adult stable transgenic lines, and each line generated and tested represents an independent transgene insertion at a unique locus within the genome.

The assay entails evaluating the consistency (presence or absence) of a given expression pattern across different lines, rather than the levels of reporter expression (which can be influenced by insertion site (ie. position effects)). For this reason, we opted not to include levels data in this table. (This is also the reason why the Enhancer 1 line was considered inconclusive with only an n=2 independently generated lines.)

As mentioned in response to comment #3 (Reviewer #2), we did, however, generate a separate table of reporter activity of the enhancer lines that were analyzed for the muscle, which reflects one representative line each for Enhancer 2, Enhancer 3, and Enhancer 2+3. This is now included in the manuscript as Supplementary Table 2.

7. Wnt1 and Wnt10b expression changes in enhancer deletion lines

The authors employ Pyrosequencing approach to measure Wnt1 and Wnt10b expression level change by enhancer deletion. However, there is no statistical analysis result. In enhancer deletion mutants, Wnt1/10b expressing cell numbers decline more than half at day 5, so their transcript levels may decline significantly at day 5.

Pyrosequencing was performed on samples at 3 days post-injury, before there is the significant drop in Wnt expression. It is true that we need to be aware that the Wnt expression measured in the assay could reflect reduced transcription from the enhancer that would normally happen by 3 days, rather than a less efficient initial induction of Wnts due to the deletion. Due to myofiber damage and cell death post-BaCl₂ injection, we did not think we could perform the assay much earlier than 3 days; injury results in low levels of RNA isolation until the muscles start to recover. Given the reviewer's concern and given that by 3 days Wnt levels start to drop in del muscles, perhaps rather than using Pyrosequencing to look for differences in injury-responsive Wnt *induction*, we should simply use the word *expression*. Nevertheless, Wnt expression levels that are driven from the wild-type enhancer alleles are still higher than that from the Δ allele and that loss of the enhancer region changes the Wnt expression levels relative to wild-type.

We re-worded the pyrosequencing section to state:

To ask if Enhancers 2 and 3 regulate Wnt1 and Wnt10b expression levels by a different method, we performed Pyrosequencing at 3 days-post injury. By crossing two different mouse strains (FVB and CAST/EIJ) carrying different SNPs that allowed us to measure Wnt1 and Wnt10b gene expression driven by the wt vs. Δ 1 or Δ 2 alleles, we could assess how well the Wnts were expressed when Enhancers 2 and 3 were lost in the same muscle under identical injury conditions. Wnt1 and Wnt10b genes were similarly expressed by both alleles in FVB^{wt}/CAST^{wt} mice, but expression was reduced from the Δ 1 or Δ 2 alleles in FVB ^{Δ 1}/CAST/EIJ^{wt} or FVB ^{Δ 2}/CAST/EIJ^{wt} mice (Fig. S13). Together, data show that Enhancers 2 and 3 are injury responsive regulatory sequences that indeed likely regulate Wnt1 and Wnt10b in injured muscle; not only are the enhancers sufficient to drive injury-

responsive reporter activity, but they are also necessary for Wnt expression in damaged tissues.

Please perform statistical analysis with sufficient number of samples and provide data.

During these experiments, we encountered some difficulties that prevented us from running as many pyrosequencing samples as we would have liked, and although we tried to perform genomic PCR controls that gave products, we were unable to get good pyrosequencing reads. Moreover, the number of samples from 100% wt FVB and 100% wt CAST PCRs is too low to include in the statistical analyses. In the absence of genomic controls, however we did use percentage expression of the minor allele in the 100% FVB or CAST controls for baseline subtraction. From this, we analyzed the data from the FVBwt/CASTwt, FVB Δ 1/CASTwt and FVB Δ 2/CASTwt which had sufficient sample numbers (ranging from n=3-6). We have included this in Fig. S13. The graphs show the percent FVB allele expressed but the statistics were done on baseline corrected and arcsine transformed values to stabilize the variance and permit use of standard statistical tests (eg. Hsiao et al. BMC Genomics 2012, 13:345; <http://www.biomedcentral.com/1471-2164/13/345>).

We have removed the uninjured condition from the graphs, since the low levels of Wnt1 in the uninjured state required many more PCR cycles to generate readable sample (and possibly represented genomic data), and it seemed that comparing expression in the injured state was more relevant to the question in hand.

We admit that these pyrosequencing experiments are perhaps interesting but not as strongly executed as we would like. We could remove these results entirely, although the data are included here for now.

Moreover, SNPs in Wnt1/10b regulatory elements across distinct wild-type strains may affect injury-responsive expression.

Within the Enhancer 2 and 3 regions, there are 8 SNPs that differ between FVB and CAST mice. 3 lie within the Enhancer 3 sequence, 3 lie within the region between Enhancers 2 and 3, and 2 reside within Enhancer 2. None reside in the predicted AP-1 binding sites. One of the Enhancer 3 changes, which is an insertion in CAST resides at the very beginning of the Enhancer 3 sequence. A table is shown below:

SNP location	Gene	CAST	FVB	Change	Location
15_98780267_G_A	Rhebl1	A	G	SNP	Enh 2
15_98780396_A_G	Wnt10 b	G	A	SNP	Enh 2
15_98780402_G_A	Wnt10 b	A	G	SNP	Between Enh2 and 3
15_98780426_G_C	Wnt10 b	C	G	SNP	Between Enh2 and 3
15_98780444_C_CATGCACACACAC ACAT	Rhebl1	CATGCACACACACACA T	C		Between Enh2 and 3
15_98780447_G_GCACACACACACA TATGCA	Rhebl1	GCACACACACACATAT GCA	G	Insertio n	Enh 3*
15_98780525_CTG_C	Wnt10 b	C	CTG	Deletio n	Enh 3
15_98780526_A_G	Rhebl1	G	A	SNP	Enh 3

*Note: We define Enh3 to start at the 98780446 position; this change resides at the very beginning of the Enhancer 3 sequence

It is formally possible that these nucleotide differences could influence the regulation of Wnt expression between FVB and CAST. However to the best of our knowledge and based on what we observe in the pyrosequencing analyses, it does not appear that there is a significant difference in Wnt1 or Wnt10b expression between FVB and CAST in the wild-type state, and the relative representation of the FVB allele is very close to 50%, the

expected value if the two alleles were to function similarly.

8. Active enhancer signature

There are multiple published ATAC-seq profiles, active enhancer profiles or scATAC-seq datasets for uninjured and injured skeletal muscles. Please examine whether Enh2 and Enh3 become accessible or are marked by active enhancer markers upon injury.

Although there are many deposited data sets that are available, very few replicate our experimental design or are from subsets of cells that may or may not reflect the transcriptional state of our cells of interest. Nevertheless, we provide two examples here that support the general idea that Enhancers 2 and 3 could function as regulatory elements:

First, we have added a figure (Fig. S7) showing data from ENCODE and CHIP-seq from various (uninjured) tissues in which Enhancers 2 and 3 are predicted to function as enhancers. Data from skeletal muscle and lung also supports the idea that these regulatory regions consist of open DNA as evidenced by DNase1 hypersensitivity sites in this region.

Second, we plot data from an ATAC-seq experiment that we did not include in the paper, but show here, in which fixed satellite cells were analyzed at 1 hour, 16 hours, 32 hours and 60 hours post-BaCl₂ injury and compared to uninjured sample (Dong et al., 2022; DOI:10.1016/j.isci.2022.104954). It is evident that slightly shifting peaks are present at the site of Enhancers 2 and 3 (marked by a red box), in support of the idea that these sequences represent sites of open chromatin where there may be increased access to transcription factors. From these data, we believe there is some evidence that the Enhancer 2+3 regulatory region can serve as enhancers for gene regulation, although further studies in the future will be needed to examine this region in greater detail.

Minor

1. Abstract first sentence is overstated. Please revise it.

"The capacity to detect and respond to injury is critical for the survival of all organisms" -
 > Please weaken by removing "all".

We have changed this sentence to: "For many organisms, a capacity to detect and respond to injury is critical for recovery and long-term survival."

2. Wnt1 and Wnt10b gene structure

Fig 2A and S4A are low resolution, making difficult to evaluate the data. Which one is Wnt10b and what is the direction? Please annotate "Wnt10b" above left gene and "Wnt1" above right one. Please clearly indicate the direction of both genes. Please include 5.5kb 3' regulatory region containing the embryonic regulatory element in this genome browser. The current format should be improved for publication.

We have revised the figures with better resolution images from the browser (Figure 2A, Fig. S5a). For Figure 2A, we focused only on Enhancers 2 and 3 and did not include Wnt1, as it is not directly addressed in this figure. However, S5A (previously S4A) has a revised version of the schematic with the 3 conserved Enhancers residing between Wnt1 and Wnt10b, as well as the 3' regulatory sequence near Wnt1. Labels have been clarified.

3. AP-1 sequence conservation

Which species are used for sequence conservation study in Fig 2A? Please provide more detailed information of conservation data.

We examined conservation of Wnt1 and Wnt10b using the mm9 mouse genome alignment. The original alignment was performed on a limited number of species, but we agree that adding this to the paper helps to clarify what was done and why we picked the regions that we studied. In re-making Fig. 2A, we added this information to the panel in the manuscript:

Line 276-278 suggests that two AP-1 site sequences across multiple species were compared. If so, please provide the data. Providing actual data will improve this manuscript.

We did not explain this clearly enough in the first version of this paper, so we have revised this section to more explicitly state what was done. Main points are that:

- 1) We identified the two AP-1 sites in the Enhancer 3 sequence using bioinformatics tools that were available to us at that time. (GREAT and TF-Search). GREAT identified one site, and TF-search identified a second.
- 2) In alignments, the first site AP-1 binding site is conserved across multiple species. The second is a consensus binding site for AP-1 in the mouse but is not well conserved in other species.
- 3) For the purposes of our experiments, we opted to mutate both sites. We reasoned that only focusing on the single conserved site might cause us to miss an important function for the other site. Given the lack of conservation of the second site in other animals, particularly now that there are many more species to use for comparison, it is possible that there is only a single primary AP-1 site that is required for the injury response of Enhancer 3. On the other hand, it is formally possible, that in the mouse specifically, both sites function to drive injury-responsive gene expression. Our experiments here do not allow us to distinguish between these possibilities. We added a short sentence to summarize this point and include a new supplementary figure showing alignments of the AP-1 binding sites (Fig. S10).

4. Line 291-294 and Table2.

This interpretation is overstated as $del1/+$ and $del2/+$ ratios are not increased. If there is any detrimental effect of homozygotes, all of other genotypes, wt , $del1/+$ and $del2/+$, should increase rather than only wt . Please revise this statement.

We agree that we did not explain this fully in the last draft. We added the following to the text to highlight the fact that the loss of the enhancer may be detrimental in both the homozygotes and likely the heterozygotes as well:

We obtained viable $\Delta1/\Delta2$ animals, although we noticed that ratios were slightly lower for this class than predicted when assessed at weaning age, suggesting that there could be some mild detrimental effect of enhancer deletion in homozygotes (Table 2; expected frequency was 25% but 19.44% was observed). The reduced percentage of $\Delta1/\Delta2$ animals also predicted that the remaining classes, if all equally healthy, each should

produce ratios of ~26.85, but heterozygotes ($\Delta 1/+$ and $\Delta 2/+$) were obtained at slightly lower frequencies than expected ($\Delta 1/+$, 25.66%; $\Delta 2/+$, 24.34%; Table 2). These data indicated that loss of the enhancer region may negatively affect both homozygotes and heterozygotes.

5. Correct typo in line 336. It says (Figure D, E, ...) and should be corrected to reference Fig 5D, E.

Thank you. This has been corrected.

6. Correct typo in lines 912 and 914, which says *P<0.05, **P<0.05.

Thank you. We have corrected this.

7. In the discussion in lines 455-458, the authors claim that this is the first example of an enhancer region driving developmental signaling molecules that is re-used in adulthood to facilitate regeneration. However, no data are provided to show that Enh2 or Enh3 can drive expression during development.

Our wording was confusing and incorrect. What we meant to say was that the Wnt genes, which are important embryonic signaling molecules, are re-used to drive regeneration, and that our enhancer is the first example in the mouse of a regulatory region that facilitates Wnt expression during regeneration. We did not mean to state that it was the enhancer region that was used during development (although this is a possibility, given expression of the reporters in embryos as shown in Figure S12).

The preceding text has now been changed, and the sentence in the manuscript now simply reads: "To our knowledge, the phenotype we describe here is the first to show requirement for a regulatory region that facilitates activation of Wnt expression during regeneration in higher vertebrates."

Please perform reporter assays in developing tissues to demonstrate their developmental activity. If such data are not available, this statement should be revised accordingly to avoid overinterpretation.

We do see embryonic reporter expression, but it is not clear how well expression recapitulates known sites of Wnt1 and Wnt10b expression, and we did not pursue figuring out the underlying cause of lethality in embryos or early post-natal mice. (Please see Reviewer #1, Major comment #4 and Fig. S12 for images of embryonic staining and discussion).

8. Wnt1 and Wnt10b co-regulation by Enh2 and 3

If Wnt1 and Wnt10b are regulated by the same enhancer elements, why do most nuclei express either one gene or the other, rather than both (as shown in Fig. 1G and discussed in Lines 149-154)? The fact that only about 10% of cells co-express both genes may not provide strong support for the claim that "common regulatory element" drives their expression. The authors should address this discrepancy in the discussion. Note that Wnt10b level seems to be more influenced by enhancer deletion, suggesting the presence of other regulatory element controlling Wnt1.

We agree that the lack of frequent co-expression of the Wnts is concerning, but the loss of a single enhancer region seems to reduce both Wnts albeit to different levels. We posit that this is the strongest evidence that a single regulatory region can control both genes. We have added a paragraph to the discussion that states:

The arrangement of Wnt1 and Wnt10b as neighboring genes with a shared regulatory region is supported by our observations that the expression of these Wnts is lost when Enhancers 2 and 3 are deleted (Fig. 4). Additionally, publicly available data indicates that Enhancers 2 and 3 can function as enhancers and may reside within a single topologically associated domain, with possible interactions between the enhancers and the Wnt1 and 10b promoters (Fig. S7). On the other hand, although we posit that Enhancers 2

and 3 regulate both Wnt1 and Wnt10b, co-expression of these two Wnts in the same cells is infrequently observed (Fig. 1G), which could be due to differences in stability of the Wnt1 and Wnt10b mRNAs, the duration and frequency of enhancer/promoter contacts, and the specific interactions that occur within the different Wnt1 and Wnt10b expressing cell types. Future studies may be able to elucidate the dynamic nature of Enhancer 2 and 3 engagements with the Wnt1 and Wnt10b promoters and other regulatory elements to produce the expression patterns of Wnt1 and Wnt10b expression in the injured state.

Additionally, while TAD (topologically associating domain) data are mentioned in the discussion, no such data are provided. Including a TAD map as a supplementary figure would strengthen this claim - please provide this data if available.

There is TAD data to suggest that the two Wnt genes may reside in a single topological domain. Although the TAD data come from cell lines in a non-injury context, they suggest that there may be short-range contact points that could allow both Wnts to be regulated by Enhancers 2 and 3. A figure from the UCSC genome browser shows data from human ESCs and human foreskin fibroblast (HFFc6) cells. We have marked the triangular zone of interaction points in the region of the Wnt1 and Wnt10b genes with dotted lines. The fact that two different cell lines show a similar TAD map supports the idea that this region is a single TAD. As Development's guidelines discourage inclusion of supplementary figures to support points raised in the discussion, we moved this point to the results and have generated a new supplementary figure (Fig S7).

Second decision letter

MS ID#: dev.204933R1

MS TITLE: Deletion of an enhancer that controls Wnt gene expression following tissue injury produces increased adipogenesis in regenerated muscle

AUTHORS: Catriona Logan, Xinhong Lim, Matt P. Fish, Makiko Mizutani, Brooke Swain and Roel Nusse

Dear Dr Logan,

I have now received all the referees reports on the above manuscript, and have reached a decision. The referees' comments are appended below, or you can access them online: please go to .

The overall evaluation is positive and we would like to publish a revised manuscript in Development, provided that the referees' comments can be satisfactorily addressed.

At this stage I would strongly encourage that you edit the manuscript with textual edits. I am not expecting any addition of experimental data. It will be important to address the comments regarding overstatements.

Please attend to all of the reviewers' comments in your revised manuscript and detail them in your point-by-point response. If you do not agree with any of their criticisms or suggestions explain clearly why this is so. If it would be helpful, you are welcome to contact us to discuss your revision in greater detail.

Reviewer 1

Advance summary and potential significance to field

The manuscript identifies a conserved regulatory region between Wnt1 and Wnt10b that is activated by tissue injury and required for proper muscle regeneration. This work represents the first identification of a vertebrate injury-responsive enhancer driving Wnt ligands critical for regeneration, with multi-tissue validation across lung, muscle, liver, and skin that strengthens the enhancer's conserved role in damage sensing. The methodology is rigorous, combining CRISPR deletion, transgenic reporters, and allele-specific expression analysis through pyrosequencing to establish enhancer necessity and sufficiency, while AP-1 site mutagenesis cleanly links enhancer activity to injury signaling. The physiological relevance is demonstrated through germline enhancer deletion that produces a clear regenerative defect characterized by adipogenesis without developmental confounders, highlighting the enhancer's adult-specific role in tissue repair.

Comments for the author

The authors have satisfactorily addressed my earlier concerns and should be commended for their efforts.

Reviewer 2

Advance summary and potential significance to field

The important concern was that enhancer deletion lines are unlikely to display a significant, biologically meaningful regeneration-defective phenotype. The authors acknowledge that the observed regeneration phenotype (increased fat accumulation) in the enhancer deletion line is subtle. The authors provide reasons for not performing additional regeneration assays, such as using sensitized genetic background and analyzing aged animal, which were proposed in my first comment. Although the authors clarified their previous regeneration assay analyses and reorganized the presentation, I felt that the functional contribution of these enhancers to regeneration remains elusive. This negative outcome is presumably common given potential redundancy by other enhancers or compensatory adjustment after enhancer deletion. This is also noted by the authors in discussion.

Despite this, the current Abstract, parts of Results, and Discussion still assert that Wnt1/10b enhancers identified here are functional and required for regeneration. If the authors wish to retain this strong claim that wnt-linked enhancers functionally regulate development and regeneration, the evidence presented here is likely insufficient.

Nevertheless, I would like to note that this manuscript has important strengths. The authors systematically profile transcriptional regulation of all wnt gene family upon injury in four distinct tissues. The authors comprehensively dissect several non-coding regions whether they can drive injury-responsible expression by generating a large panel of mouse reporter strains. The authors identified the importance of AP-1 binding motifs in injury-responsible enhancers.

I therefore offer the following comments to help reframe the manuscript around these robust contributions and to better align the claims with the current evidence, thereby strengthening its suitability for publication.

1. Revise Abstract

"Injured muscles in mice carrying a germ-line deletion of the enhancer region display reduced Wnt1 and Wnt10b expression and show elevated intramuscular adipogenesis-- a hallmark of impaired regenerative capacity—revealing a requirement of this enhancer for proper regeneration. Enhancer redundancy is common in development, but our in vivo analysis shows that loss of a single injury-responsive regulatory region in adult tissues can produce a detectable regenerative phenotype" There is no direct evidence that elevated intramuscular adipogenesis in the enhancer deletion line demonstrates a requirement for proper regeneration. Please weaken the regeneration claim that these enhancers are required for regeneration. Please avoid ending with "detectable regenerative phenotype". I strongly recommend reframing the Abstract to emphasize: (i) systemic Wnt expression dynamics across distinct tissues post-injury, and (ii) dissection of Wnt enhancers in injury contexts by reporter assay, rather than asserting a regeneration phenotype.

2. Weaken overstated statement about regeneration phenotype

Page 4, lines 92 and 93 "is required for proper regeneration of damaged muscle."

Current data do not support that the enhancer deletion influences muscle regeneration. Please delete or soften this substantially.

Page 12, Line 332 "Enhancers 2 and 3, once born, are mostly unaffected in the absence of injury." -> Remove "in the absence of injury". This can be misinterpreted that enhancer deletion lines exhibit significant defects with injury.

Page 13, Line 360 "regulatory sequences that indeed likely regulate Wnt1 and Wnt10b in injured muscle;" -> Remove "indeed"

Page 13, Line 362 "are also necessary for Wnt1 and Wnt10b expression in damaged tissues" -> Add "partially" between "also" and "necessary". Note that Wnt1 and Wnt10b transcription levels upon injury are not robustly abolished.

Page 13, Line 364 "Deletion of Enhancers 2 and 3 produces adipogenesis in injured muscle" -> Change this section title. For example, "Deletion of Enhancers 2 and 3 leads to a mild, but significant, increase in fat accumulation in injured muscle"

Page 13, Line 370-371 "Highest levels of Perilipin staining observed in wt muscles was always reduced compared to that of $\hat{1}^{\prime}1/ \hat{1}^{\prime}2$ muscles (Figure 5D,E, Fig. S14C,H,I)." -> Remove "always".

Note that Fig 5D shows that multiple animals exhibit lower adipogenesis index values, compared to wild-type.

Page 14, Line 386 "muscles displayed much higher and more variable levels of fat" -> Change "much higher" to "mildly higher"

Page 16, Line 454 "We do not know if levels of adipogenesis seen here adversely affect muscle function." -> Remove this sentence. Given the subtle increased fat accumulation, the predicted outcome should not be adverse. If retained, functional test is required.

Page 16, Line 457 - 460 "Our results ... response to injury, is required to produce the correct balance of cell types as muscles recover from injury" -> There is no evidence of correct balance of cell types. The authors examine only fat accumulation test rather than cell type validation. Please revise this sentence by removing "is required ... from injury"

Page 17, Line 467 "and proper cell fate choices during muscle regeneration" -> Remove this as there is no experiment examining cell fate choice.

Page 17, Line 489 "Except for mild detrimental effects during embryogenesis" -> Remove this as the authors did not assess detrimental effects. Only embryo ratios were reported.

Page 17, Line 480-482 "Muscles post-injury, however, show higher levels of adipocytes in $\hat{1}^{\prime}1/ \hat{1}^{\prime}2$ tissues compared to wt (Figure 5, Fig. S14, S15) indicating impaired muscle regeneration, as fatty infiltration is often observed in diseased, disused, or aging muscle (Marcus et al., 2010; Hamrick et al., 2016; Zhu et al., 2024)." -> Revise this. There is no evidence of impaired muscle regeneration but only subtle increased fat accumulation.

Page 17, Line 490 - 492. The phenotype we describe here, to our knowledge, is the first to show a requirement for a regulatory region that regulates Wnt expression during regeneration in higher vertebrates." -> Revise this. Without strong evidence for biologically meaningful expression reduction and impaired regeneration phenotype, this claim cannot become the first example of enhancer requirement of Wnt expression during regeneration.

3. Title should be revised by removing "proper muscle regeneration"

4. Figures

Fig. S5 and S6: Wnt enhancer reporter mouse generation and expression analyses in the lung are central findings of this manuscript. Please promote Fig S5 and S6 to the main figure (suggest new Fig. 2).

Page 10, Line 258. Please include ATAC_seq plot data presented in "response to reviewers comment" file. This data is important to demonstrate that enhancers 2/3 are accessible in muscle tissues.

Fig. S12: Reporter assays to examine developmental activity of Enh 2, 3, 2+3 are important data to demonstrate Enh2 and 3 play roles as developmental enhancers. Please consider to promote S12 to a main figure.

Fig. 2 -> Please clarify color coding for green and dark blue bars. Green: minimal hsp60 promoter. Dark Blue: LacZ reporter gene.

5. Other

Page 11, Line 287 "Interestingly, in multiple genome" -> Remove "Interestingly"

Page 15, Line 416 Figure 5J, K -> may be Fig. 5H, J. There is no Fig. 5K.

AP-1 mutation: If there is rationale to change TGAGTCA to GTCTGAC, please provide it.

p value - please provide the actual value

Second revision

Author response to reviewers' comments

Reviewer 1: SUMMARY OF THE ADVANCE MADE IN THIS PAPER AND ITS POTENTIAL SIGNIFICANCE TO THE FIELD

The manuscript identifies a conserved regulatory region between Wnt1 and Wnt10b that is activated by tissue injury and required for proper muscle regeneration. This work represents the first identification of a vertebrate injury-responsive enhancer driving Wnt ligands critical for regeneration, with multi-tissue validation across lung, muscle, liver, and skin that strengthens the enhancer's conserved role in damage sensing. The methodology is rigorous, combining CRISPR deletion, transgenic reporters, and allele-specific expression analysis through pyrosequencing to establish enhancer necessity and sufficiency, while AP-1 site mutagenesis cleanly links enhancer activity to injury signaling. The physiological relevance is demonstrated through germline enhancer deletion that produces a clear regenerative defect characterized by adipogenesis without developmental confounders, highlighting the enhancer's adult-specific role in tissue repair.

SUGGESTIONS TO AUTHORS

The authors have satisfactorily addressed my earlier concerns and should be commended for their efforts.

Reviewer 2: The important concern was that enhancer deletion lines are unlikely to display a significant, biologically meaningful regeneration-defective phenotype. The authors acknowledge that the observed regeneration phenotype (increased fat accumulation) in the enhancer deletion line is subtle. The authors provide reasons for not performing additional regeneration assays, such as using sensitized genetic background and analyzing aged animal, which were proposed in my first comment. Although the authors clarified their previous regeneration assay analyses and reorganized the presentation, I felt that the functional contribution of these enhancers to regeneration remains elusive. This negative outcome is presumably common given potential redundancy by other enhancers or compensatory adjustment after enhancer deletion. This is also noted by the authors in discussion.

Despite this, the current Abstract, parts of Results, and Discussion still assert that Wnt1/10b enhancers identified here are functional and required for regeneration. If the authors wish to retain this strong claim that wnt-linked enhancers functionally regulate development and regeneration, the evidence presented here is likely insufficient.

Nevertheless, I would like to note that this manuscript has important strengths. The authors systematically profile transcriptional regulation of all wnt gene family upon injury in four distinct tissues. The authors comprehensively dissect several non-coding regions whether they can drive injury-responsible expression by generating a large panel of mouse reporter strains. The authors identified the importance of AP-1 binding motifs in injury-responsible enhancers.

I therefore offer the following comments to help reframe the manuscript around these robust contributions and to better align the claims with the current evidence, thereby strengthening its suitability for publication.

1. Revise Abstract

"Injured muscles in mice carrying a germ-line deletion of the enhancer region display reduced Wnt1 and Wnt10b expression and show elevated intramuscular adipogenesis-- a hallmark of

impaired regenerative capacity—revealing a requirement of this enhancer for proper regeneration. Enhancer redundancy is common in development, but our *in vivo* analysis shows that loss of a single injury-responsive regulatory region in adult tissues can produce a detectable regenerative phenotype"

There is no direct evidence that elevated intramuscular adipogenesis in the enhancer deletion line demonstrates a requirement for proper regeneration. Please weaken the regeneration claim that these enhancers are required for regeneration. Please avoid ending with "detectable regenerative phenotype". I strongly recommend reframing the Abstract to emphasize: (i) systemic Wnt expression dynamics across distinct tissues post-injury, and (ii) dissection of Wnt enhancers in injury contexts by reporter assay, rather than asserting a regeneration phenotype.

As reviewer #2 points out, the initial work on identifying and characterizing the regulatory sequence presented in this study involved performing *in situ* in multiple tissues to identify candidate enhancer regions, followed by generation of reporter mice. As such, one way to write the paper was simply to present the reporter assay data in the lung, reporter expression in embryos, and the AP-1 binding requirement of Enhancer for injury-responsive activity. However, we felt that it was important to ask whether the Enhancer 2+3 region was important or functional in a detectable way. This led us to explore the activity of the enhancers in the muscle, culminating in the observation that recovery post-BaCl₂ injury in *del/del* muscles deviates from that observed in wild-type muscles by producing elevated levels of fatty infiltration. We think shifting the emphasis of the paper to the initial reporter generation rather than reporter function will necessitate a re-writing of the manuscript. We have therefore kept the flow of the paper as submitted in versions 1 and 2, but we have eliminated our claims of a 'regenerative defect.'

For the abstract, the last two sentences have been changed to:

Injured muscles in mice carrying a germ-line deletion of the enhancer region display reduced *Wnt1* and *Wnt10b* expression and show elevated intramuscular adipogenesis, which can be a hallmark of impaired muscle regeneration or tissue maintenance. Enhancer redundancy is common in development, but our *in vivo* analysis shows that loss of a single injury-responsive regulatory region in adult tissues can produce a detectable phenotype.

2. Weaken overstated statement about regeneration phenotype

Page 4, lines 92 and 93 "is required for proper regeneration of damaged muscle." Current data do not support that the enhancer deletion influences muscle regeneration. Please delete or soften this substantially.

We have changed this sentence to:

We report the identification of a regulatory region residing between *Wnt1* and *Wnt10b* that *in vivo* responds to tissue injury and when deleted, results in elevated adipogenesis in regenerated muscle.

Page 12, Line 332 "Enhancers 2 and 3, once born, are mostly unaffected in the absence of injury." -> Remove "in the absence of injury". This can be misinterpreted that enhancer deletion lines exhibit significant defects with injury.

This was removed.

Page 13, Line 360 "regulatory sequences that indeed likely regulate *Wnt1* and *Wnt10b* in injured muscle;" -> Remove "indeed"

This was removed.

Page 13, Line 362 "are also necessary for *Wnt1* and *Wnt10b* expression in damaged tissues" -> Add "partially" between "also" and "necessary". Note that *Wnt1* and *Wnt10b* transcription levels upon injury are not robustly abolished.

This was added.

Page 13, Line 364 "Deletion of Enhancers 2 and 3 produces adipogenesis in injured muscle" -> Change this section title. For example, "Deletion of Enhancers 2 and 3 leads to a mild, but significant, increase in fat accumulation in injured muscle"

We have modified the title.

Page 13, Line 370-371 "Highest levels of Perilipin staining observed in wt muscles was always reduced compared to that of $\Delta 1/ \Delta 2$ muscles (Figure 5D,E, Fig. S14C,H,I)." -> Remove "always". Note that Fig 5D shows that multiple animals exhibit lower adipogenesis index values, compared to wild-type.

This was removed.

Page 14, Line 386 "muscles displayed much higher and more variable levels of fat" -> Change "much higher" to "mildly higher"

'Much' was eliminated. The sentence now reads: "... muscles displayed higher and more variable levels of fat."

Page 16, Line 454 "We do not know if levels of adipogenesis seen here adversely affect muscle function." -> Remove this sentence. Given the subtle increased fat accumulation, the predicted outcome should not be adverse. If retained, functional test is required.

This was removed.

Page 16, Line 457 - 460 "Our results ... response to injury, is required to produce the correct balance of cell types as muscles recover from injury" -> There is no evidence of correct balance of cell types. The authors examine only fat accumulation test rather than cell type validation. Please revise this sentence by removing "is required ... from injury"

We have re-worded this as follows:

Our results suggest that the Enh2/3 regulatory region, a ~1kb sequence that regulates *Wnt1* and *Wnt10b* in response to injury, influences the levels of adipogenesis produced as muscles recover from tissue damage.

Page 17, Line 467 "and proper cell fate choices during muscle regeneration" -> Remove this as there is no experiment examining cell fate choice.

We have changed this to:

Using an in vivo approach to study a single injury-responsive enhancer locus, we describe its spatial and temporal activities, examine its requirement for Wnt expression and muscle recovery post-injury, and extend our understanding of how Wnts activated in response to injury function during tissue regeneration.

Page 17, Line 489 "Except for mild detrimental effects during embryogenesis" -> Remove this as the authors did not assess detrimental effects. Only embryo ratios were reported.

We changed the sentence here to :

Uninjured animals lacking the enhancers are viable, fertile, and display no obvious defects (Figure 4C, Fig. S16), except for a slight reduction in expected frequency of progeny obtained from Mendelian crosses (Table 2).

Page 17, Line 480-482 "Muscles post-injury, however, show higher levels of adipocytes in $\Delta 1/ \Delta 2$ tissues compared to wt (Figure 5, Fig. S14, S15) indicating impaired muscle regeneration, as fatty infiltration is often observed in diseased, disused, or aging muscle (Marcus et al., 2010; Hamrick et al., 2016; Zhu et al., 2024)." -> Revise this. There is no evidence of impaired muscle regeneration

but only subtle increased fat accumulation.

We changed the sentence here to :

Muscles post-injury, however, show elevated levels of adipocytes in $\Delta 1/\Delta 2$ tissues compared to wt (Figure 5, Fig. S14, S15) revealing a mild but detectable change in fatty infiltration.

Page 17, Line 490 - 492. The phenotype we describe here, to our knowledge, is the first to show a requirement for a regulatory region that regulates Wnt expression during regeneration in higher vertebrates." -> Revise this. Without strong evidence for biologically meaningful expression reduction and impaired regeneration phenotype, this claim cannot become the first example of enhancer requirement of Wnt expression during regeneration.

We have changed this sentence the following:

The phenotype we describe here, to our knowledge, is the first to describe an injury- responsive regulatory region in mouse that influences both Wnt expression post-injury and fatty infiltration in regenerating muscle.

We do understand that reviewer #2 questions the interpretation of our findings, but we argue that the reduction in Wnt expression is meaningful, and the data do provide the first example in mouse of an enhancer that affects Wnt expression post-injury and produces a phenotype.

Supporting evidence is the pyrosequencing data, which reveals inefficient Wnt transcription from the mutant allele, and in situ data, which shows a decline in Wnt1 and Wnt10b expression over 3-5 days, that represents roughly a ~30-50% decrease over this timeframe (eg. as shown in violin plots at 3 days and bar graphs at 5 days). Wnt5a, in contrast, which is not regulated by our enhancers, does not display this type of reduction. Although wild-type Wnt expression also declines over 3-5 days post injury, the reduction in del/del animals is greater. The 3-5 days post-injury are a highly critical phase of the regenerative process, when active reciprocal cell-cell signaling occurs in regenerating muscle between multiple cell types that include the satellite cells and Fibro-Adipogenic progenitors. We know that perilipin staining is detectable by 5 days (see figure below) and QPCR data (shows elevated Pparg and Cebpa by d7 Figure 5F, G). These results suggest that Wnt expression declines during the time that the formation of adipocytes is underway. If our model is correct and Wnt1 and Wnt10b play a role in repressing adipocyte formation, it may not be so surprising that we observe fat levels that are greater in del/del animals compared to wild type when we examine the muscles at 14 days. We admit that the phenotype is fairly mild, but this is the first example of a Wnt enhancer that affects levels of fatty infiltration in regenerating muscle.

3. Title should be revised by removing "proper muscle regeneration"

The new title reads as follows:

Deletion of an enhancer that controls Wnt gene expression following tissue injury produces increased adipogenesis in regenerated muscle

4. Figures

Fig. S5 and S6: Wnt enhancer reporter mouse generation and expression analyses in the lung are

central findings of this manuscript. Please promote Fig S5 and S6 to the main figure (suggest new Fig. 2).

We agree that building the transgenic mice was a central part of our search for injury responsive enhancers. However, due to word and length limits on the manuscript, we have opted to keep these figures in the supplement, since the main paper figures are mostly centered on the enhancer activity in the muscle. In earlier versions and drafts of the paper, we found that a focus on the initial transgene construction and subsequent switching between multiple tissues made it confusing for readers. We therefore opted to keep the main story mostly muscle-focused with other tissues mentioned as needed in the main text and in the supplementary material.

Page 10, Line 258. Please include ATAC-seq plot data presented in "response to reviewers comment" file. This data is important to demonstrate that enhancers 2/3 are accessible in muscle tissues.

We have added the ATAC-seq data to Fig. S7 in the supplement to accompany the other UCSC genome data showing that Enhancer 2 and 3 are likely enhancers that function within the same TAD. Text was also added to the supplementary figure legend. We have attached revised Fig. S7 and the legend to the end of this document (highlighted to show changes).

Fig. S12: Reporter assays to examine developmental activity of Enh 2, 3, 2+3 are important data to demonstrate Enh2 and 3 play roles as developmental enhancers. Please consider to promote S12 to a main figure.

As mentioned in response to comment "Fig. S5 and S6:" above, we feel that with the word and length limits on the manuscript, it is impractical to add more panels to the main figures, which are already lengthy and contain full-page data.

We do hope that people will view the supplementary figures, however, and anyone interested in using the reporters as read-outs for particular tissues or structures, or to study possible Wnt expression in these lacZ patterns will be able to use these animals.

Fig. 2 -> Please clarify color coding for green and dark blue bars. Green: minimal hsp60 promoter. Dark Blue: LacZ reporter gene.

Thank you for this suggestion. We agree that this was not explicit. We have added this information to the figure legend as "Minimal hsp60 promoter (green), LacZ reporter (blue)".

Additionally, to make the figure itself more intuitive, we also color coded the labels beside the constructs for "hsp68" and "LacZ" to match the color of the boxes in the schematic. A screen shot of the constructs from the revised figure is shown below and a new Figure 2 has been uploaded:

5. Other

Page 11, Line 287 "Interestingly, in multiple genome" -> Remove "Interestingly"

This was removed.

Page 15, Line 416 Figure 5J, K -> may be Fig. 5H, J. There is no Fig. 5K.
 Thank you very much for catching this typo. We have corrected this.

AP-1 mutation: If there is rationale to change TGAGTCA to GTCTGAC, please provide it. Rationale to change TGAGTCA to GTCTGAC is provided in the materials and methods, (lines 716-724) to explain why this sequence was altered in this manner.

p value - please provide the actual value

Thank you for pointing this out. We have changed the p-values to the actual values.

We also noticed p-value errors in the supplementary figures, and these have been fixed as well.

Figures where specific p-values were added are:

- **Figure 1 Legend**

(note: We noticed that we had originally written $**P < 0.005$; it should have said $**P < 0.01$, but we fixed this to show the actual p values.)

- **Figure 4 Legend**

- **S14 C, E Legend**

- **S16 A, B, N Legend** (note: in **S16 A** we caught a mistake—the wrong statistical test had been applied. This has been fixed. We lost significance on the wt uninjured vs. wt BaCl₂ bar for PDGRa expression and del uninjured vs del BaCl₂ significance dropped to *P rather than **P and the specific p-value has been added to the legend). We do not think this change alters the interpretation of the result, but S16A is now the correct data figure (attached at the end of this document).

Fig. S7: Examples of publicly available data that suggest that Enhancers 2 and 3 can function as regulatory regions.

(A) Output from the UCSC Genome Browser (mm10) to look for candidate cis-regulatory sequences (cCREs) and signatures of open chromatin shows that Enhancers 2 and 3 could be cis-regulatory regions in both lung and muscle. Both enhancer sites display peaks of DNase1 hypersensitivity (red boxes marking green peaks). The Enhancer 2 and 3 regions, which can be identified by their

conserved peaks (blue peaks, Placental Cons.) on multi-species alignments are also marked (red boxes). (B) Output from the UCSC Genome Browser (hg38) summarizes data from ENCODE, ReMap and ORegAnno outputs and H3K27Ac marks to suggest that Enhancers 2 and 3 could be functional enhancers. The Enhancer 2 and 3 regions, which can be identified by their conserved peaks (blue peaks) on multi-species alignments (100 vertebrates Basewise Conservation by Phylo P and Multiz Align) are outlined by red boxes. The ENCODE data suggests that Enhancer 2 and Enhancer 3 display DNase-H3K4me3 activity (pink box), and a distal enhancer-like signature (orange box) respectively. ReMap, which integrates multiple CHIP-seq and other datasets to output possible regulatory elements shows higher density signals in Enhancer 3. H3K27Ac marks are also seen within Enhancers 2 and 3. The ORegAnno output indicates that Enhancer 2 and 3 regions harbor transcription factor binding sites (orange boxes), and that Enhancer 3 is a possible regulatory region (light blue box). (C) ATAC-seq peaks to show that Enhancers 2 and 3 (red box) can bind transcription factors in injured muscle. Data from Dong et al., (2022) (GSE189044) shows satellite cells that were analyzed at 1, 16, 32, and 60 hours post-BaCl₂ injury and compared to uninjured sample. (D) Outputs from the UCSC Genome Browser (hg38) showing Micro-C Chromatin assays to identify possible contact points in and around *Wnt1* and *Wnt10b* suggests that this region could form a topologically associated domain. The overall boundaries spanning *Wnt1* and *Wnt10b* are marked with a dotted line. The locations of Enhancers 2 and 3 are outlined with red boxes where peaks of high conservation are observed (100 vertebrates Basewise Conservation by Phylo P and Multiz Align). Data are from human foreskin fibroblasts (HFFc6 cells) and human embryonic stem cells (H1-hESCs).

Fig. S14 Adipogenesis in BaCl₂ injured muscles at 14 days post-injury

(A) Line graphs showing % adipogenesis values obtained from BaCl₂ injected TA muscles at 14 days post-injury over 50 serial sections to show that there are fluctuations in the level of fat observed across sections. Two examples from wt mice and two examples of $\Delta 1/\Delta 2$ muscles are shown. (B) Scatter plots showing the individual % adipogenesis measurements for each section analyzed in individual animals to show the spread of values that were observed at 14 days post-injury. There is a general upward trend in the distribution of values observed in injured $\Delta 1/\Delta 2$ muscles compared to wild type.

Uninjured mice show very low % adipogenesis values regardless of genotype. Lines show the median. n=6 wt uninjured, n=5 $\Delta 1/\Delta 2$ uninjured, n=7 wt BaCl₂, n=12 $\Delta 1/\Delta 2$ (C) Bar graph to show the mean % adipogenesis values for each wt vs. $\Delta 1/\Delta 2$ animal. Only animals in which BaCl₂ injury produced >45% damage in the tissue are shown. Colored dots on the graph correspond to examples of numbered mouse tissue sections with the matching colors shown in the panels D-F to correlate the plotted mean % adipogenesis values with the staining levels observed in tissue. n=7 wt, n=12 $\Delta 1/\Delta 2$ (*P=0.0215, Welch's t-test) (D) Violin plot of mean % adipogenesis values from all fields quantified for Perilipin staining in uninjured wild-type muscles (n=181 fields, n=4 mice), injured wt muscles (n=375 fields, n=8 mice) and injured $\Delta 1/\Delta 2$ muscles (n=423 fields, n=12 mice) at 14-days post-BaCl₂ injury to show the distribution of all samples analyzed. The level of adipogenesis observed is elevated in $\Delta 1/\Delta 2$ muscles compared to wild type. 35-50 fields were counted per mouse. (E) Comparison of adipogenesis levels when fatty infiltration is evaluated by Perilipin vs. H&E staining. By both staining methods, a similar increase in adipogenesis is observed in which injured $\Delta 1/\Delta 2$ muscles display more fat than wt muscles (n=8 wt, n=9 $\Delta 1/\Delta 2$; 40-50 fields per mouse were quantified for Perilipin staining; n=10 fields per mouse were quantified for H&E staining. Perilipin, Welch's t-test *P=0.0260; H&E, Welch's t-test *P=0.0330) (F) RT-qPCR of *Axin2* expression following injury in wt and $\Delta 1/\Delta 2$ mice shows a slight decrease in expression level that might indicate reduced Wnt signaling when Enhancers 2 and 3 are deleted, but the difference is not statistically significant. n=8 wt, n=7 $\Delta 1/\Delta 2$ (Welch's t-test, ns = not significant. (G-I)

Representative examples of adipogenesis observed in sections of BaCl₂ injected TA muscles at 14 days from wt and $\Delta 1/\Delta 2$ animals that were stained with Perilipin1 antibody to mark adipocytes. Sections are arrayed on a single dark background separated by genotype (G), and/or treatment (H, I) for ease of viewing, but each section represents tissues from separate animals. Arrowheads point to patches of adipogenesis in the muscle. Grey boxes outline magnified regions that are shown in the row below. (G) Two representative examples from uninjured mice (n=1 wt, n=1

$\Delta 1/\Delta 2$) showing very low levels of adipogenesis. Neither wt (G') nor $\Delta 1/\Delta 2$ (G'') show large patches of Perilipin staining. (H) Representative examples from 3 injured wt muscles showing the range of adipogenesis that was observed. Colors of the numbers correspond to the mice shown in the dots on the wt BaCl₂ bar of the graph in panel C to show how mean %adipogenesis values obtained for the different experimental animals correlate with the appearance of staining in tissues. The section numbered 1 and magnified panel (H') shows the lowest levels of adipogenesis, with section 2 (H'') showing higher levels. Section 3 shows the most adipogenesis throughout the section with large patches of adipocytes (H'''). (I) Representative examples from 5 injured $\Delta 1/\Delta 2$ muscles showing the range of adipogenesis that was observed. Colors of the numbers correspond to the colored dots that represent different mice shown in the $\Delta 1/\Delta 2$ (del) BaCl₂ bar of the graph in panel C to show how the different mean % adipogenesis values obtained for the different experimental animals correlate with the appearance of staining in tissues. Section number 1 and magnified panel (I') shows the lowest levels of adipogenesis, Section 2 (I'') shows higher levels but sections 3-5 display the highest Perilipin1 staining, with increasingly large areas of the section showing large regions of fat accumulation. Magnified insets show large patches of adipocytes found within the tissues (I''', I''', I'''). Error bars are mean + s.d., EDL = Extensor Digitalis Longus (Scale Bars: 200um (whole muscle sections), 50um (insets))

Old figure with error

Fig. S16

NEW corrected figure

Fig. S16: Comparison of Zt and $\Delta 1\Delta/2$ in injured and uninjured muscles

(A-H, N) Assessment of responses of wt and $\Delta 1\Delta/2$ muscles to BaCl₂ injury in injured mice. (A) RT-qPCR of *PDGFRa* expression following injury in wt and $\Delta 1\Delta/2$ mice showed significant up-regulation in injured muscles post-BaCl₂ injection in both genotypes, but the differences between genotypes are not significant. Uninjured: n=4 wt, n=4 $\Delta 1\Delta/2$, BaCl₂ injured: n=6 wt, n=6 $\Delta 1\Delta/2$. (wt, ns; $\Delta 1\Delta/2$, *p=0.0205; One-way ANOVA with Kruskal-Wallis Test) (B) RT-qPCR of *Pax7* expression following injury in wt and $\Delta 1\Delta/2$ mice showed significant up-regulation in injured muscles post-BaCl₂ injection in both genotypes, but the differences between genotypes are not significant. Uninjured: n=4 (wt), n=4 $\Delta 1\Delta/2$, BaCl₂ injured: n=6 wt, n=6 $\Delta 1\Delta/2$. (wt, *p=0.0362; $\Delta 1\Delta/2$, *p=0.0476) One-way ANOVA with Tukey's Multiple Comparisons Test) (C) RT-qPCR of *CEBPb* expression following injury in wt and $\Delta 1\Delta/2$ mice showed significant down-regulation in injured muscles post-BaCl₂ injection in both genotypes, but the differences between genotypes are not significant. Uninjured: n=6 wt, n=6 $\Delta 1\Delta/2$, BaCl₂ injured: n=7 wt, n=7 $\Delta 1\Delta/2$ (**** p<0.0001, One-way ANOVA with Tukey's Multiple Comparison Test) (D) Representative examples of WGA stained muscle sections comparing BaCl₂-injured wt vs $\Delta 1\Delta/2$ muscles at 14 days show no

obvious differences in myofiber shape or size. n=8 wt, n=8 $\Delta 1 / \Delta 2$ (E) Cross sectional area measurements of injured wt vs $\Delta 1 / \Delta 2$ muscles show no difference between genotypes. n=8 wt, n=8 $\Delta 1 / \Delta 2$ (F) Violin plots showing the range of CSAs displayed by myofibers when only those that display centrally located nuclei are quantified. The range of measured CSA values and median (red line) are very similar between wt and $\Delta 1 / \Delta 2$ animals. wt = 2827 myofibers from n=5 animals, $\Delta 1 / \Delta 2$ = 1759 myofibers from n=4 animals (G) Picrosirius Red staining of wt vs $\Delta 1 / \Delta 2$ muscles at 14 days post-injury shows no obvious differences in the intensity of staining and collagen deposition in the muscle. Two representative examples of each genotype are shown. n=7 wt n=8 $\Delta 1 / \Delta 2$ (H-N) Responses of wt and $\Delta 1 / \Delta 2$ muscles to BaCl₂ injury in uninjured mice. (H, I) RT-qPCR of *PDGFR α* (H) and *Pax7* (I) expression in uninjured muscles to compare basal levels between wt and $\Delta 1 / \Delta 2$ mice show no significant difference between the genotypes. n=5 wt, n=6 $\Delta 1 / \Delta 2$ (*PDGFR α* , Unpaired t-test; *Pax7* Welch's t-test) (J) Representative examples of WGA-stained muscle sections of uninjured wt vs $\Delta 1 / \Delta 2$ tissues show no obvious differences in myofiber shape or size. n=4 wt, n=5 $\Delta 1 / \Delta 2$ (K) Cross sectional area measurements of uninjured wt vs $\Delta 1 / \Delta 2$ muscles show no difference between genotypes. n= wt, n=8 $\Delta 1 / \Delta 2$ (L) RT-qPCR of *Myh1*, 2 and 4 expression in uninjured muscles to compare basal levels between wt and $\Delta 1 / \Delta 2$ mice showed no significant difference between the genotypes. n=5 wt, n=6 $\Delta 1 / \Delta 2$ (*Myh1*, *Myh2* Unpaired t-test, *Myh4* Mann-Whitney test) (M) Picrosirius Red staining of wt vs $\Delta 1 / \Delta 2$ muscles show no obvious differences in the intensity of staining and collagen deposition in the muscle prior to injury. (N) Quantification of Picrosirius Red staining uninjured and injured wt vs $\Delta 1 / \Delta 2$ muscles shows no difference in the deposition of collagen between genotypes in the uninjured state, or at 14 days post-injury, although collagen deposition is increased in injured muscles compared to uninjured samples. Uninjured: n=6 wt, n=5 $\Delta 1 / \Delta 2$, BaCl₂ injured: n=7 wt, n=8 $\Delta 1 / \Delta 2$ (wt uninjured vs. wt BaCl₂ *P=0.0014; wt uninjureds. $\Delta 1 / \Delta 2$ BaCl₂ *P=0.0068; One-way ANOVA with Sidak's Multiple Comparison's Test) Error bars represent mean + s.d. (RT-qPCR); mean + s.e.m (histograms) (Scale bars: 20 μ m)

Figure 2

Third decision letter

MS ID#: dev.204933R2

MS TITLE: Deletion of an enhancer that controls Wnt gene expression following tissue injury produces increased adipogenesis in regenerated muscle

AUTHORS: Catriona Logan, Xinhong Lim, Matt P. Fish, Makiko Mizutani, Brooke Swain and Roel Nusse

Dear Catriona,

Thanks a lot for submitting this revised version of your manuscript.

I am happy to tell you that your manuscript has been accepted for publication in Development, pending our standard publication integrity checks.